# VISTA drives pancreatic tumor progression through modulation of the tumor-associated macrophage polarity

Suk-Kyung Shin [1,2,3,4,10], Gwanghun Kim[1,5,10], Su Min Park [1,2,3,4], Eun-Bi Seo[1,6], Sang-Kyu Ye[1,2,4,6], Gyeong Hoon Kang [7], Keehoon Jung [1,4], Hyun Mu Shin [1,2,4,5] ✉, Hang-Rae Kim[8,9] ✉ & Dong-Sup Lee [1,3,4] ✉

Pancreatic ductal adenocarcinoma (PDAC) remains one of the deadliest malignancies due to its highly immunosuppressive tumor microenvironment (TME), which limits effective therapeutic interventions. Here, we demonstrate that V-domain immunoglobulin suppressor of T cell activation (VISTA) plays a crucial role in orchestrating macrophage polarity within the PDAC TME. Using murine PDAC models, we show that VISTA deficiency markedly impairs tumor growth, leading to prolonged survival. Functionally, VISTA deficiency is linked to a shift in tumor-associated macrophages (TAMs) from an immunosuppressive phenotype marked by secreted phosphoprotein 1 (SPP1), to one enriched for C-X-C motif chemokine ligand 9 (CXCL9), indicative of a pro-inflammatory state. This shift is accompanied by enhanced recruitment of CXCR3+ CD8+ T cells with sustained cytotoxic potential, among which terminal exhaustion-like CD8+ T cell states are less prevalent. Additionally, VISTA-deficient TAMs exhibit increased antigen cross-presentation, further amplifying CD8+ T cell response against tumors. These findings are corroborated by human PDAC data, which reflect similar immune reprogramming trends. By defining the role of VISTA in controlling *Cxcl9:Spp1* ratio and modulating CD8+ T cell dynamics, this study positions VISTA inhibition as a promising strategy to reshape the TME and potentiate anti-tumor immunity in PDAC.

Pancreatic ductal adenocarcinoma (PDAC) is among the deadliest malignancies, with a five-year survival rate below 5% and a rising incidence projected to become the second leading cause of cancer-related deaths in the United States by 2030[1,2]. Current therapeutic options remain largely ineffective, with only 15–20% of patients eligible for curative surgery[2,3]. Most patients present with unresectable or metastatic disease, and even those who undergo surgery often experience recurrence[3]. Gemcitabine, the standard chemotherapeutic agent for

[1]Department of Biomedical Sciences, Cancer Research Institute, Seoul National University College of Medicine, Seoul, Republic of Korea. [2]Wide River Institute of Immunology, Seoul National University, Gangwon, Republic of Korea. [3]Convergence Research Center for Dementia, Seoul National University Medical Research Center, Seoul, Republic of Korea. [4]BK21 FOUR Biomedical Science Project, Seoul National University College of Medicine, Seoul, Republic of Korea. [5]Medical Research Center, Seoul National University College of Medicine, Seoul, Republic of Korea. [6]Department of Pharmacology, Ischemic/Hypoxic Disease Institute, Seoul National University College of Medicine, Seoul, Republic of Korea. [7]Department of Pathology, Seoul National University College of Medicine, Seoul, Republic of Korea. [8]Samsung Precision Genome Medicine Institute, Research Institute for Future Medicine, Samsung Medical Center, Seoul, Republic of Korea. [9]Department of Health Sciences and Technology, Samsung Advanced Institute for Health Sciences & Technology (SAIHST), Sungkyunkwan University, Seoul, Republic of Korea. [10]These authors contributed equally: Suk-Kyung Shin, Gwanghun Kim.
✉e-mail: hyunmu.shin@snu.ac.kr; hangrae.kim@skku.edu; dlee5522@snu.ac.kr

PDAC, provides only marginal benefits due to rapid resistance, leaving patients with limited options and poor prognosis[4].

Recent breakthroughs in immunotherapy, particularly immune checkpoint blockade (ICB) therapies targeting Programmed cell Death protein 1 (PD-1), PD-Ligand 1 (PD-L1), and Cytotoxic T-Lymphocyte-Associated protein 4 (CTLA-4), have transformed the treatment landscape for several cancers[5–7]. However, PDAC remains resistant to these approaches due to its unique tumor microenvironment (TME)[8–10]. The TME in PDAC is characterized by dense stroma, low tumor mutational burden, and a highly immunosuppressive milieu, which together limit ICB effectiveness[5,11,12]. Thus, there is an urgent need to explore novel immune targets and mechanisms to overcome resistance in PDAC and foster tumor regression.

V-domain immunoglobulin suppressor of T cell activation (VISTA), also known as *Vsir*, c10orf54, GI 24, Dies-1, PD-1H, and DD1α, has emerged as a promising target in this context[13–16]. Unlike PD-1 and CTLA-4, VISTA regulates immune homeostasis during early T-cell activation[17]. VISTA is predominantly expressed on myeloid-derived suppressor cells (MDSCs) and tumor-infiltrating leukocytes, playing a pivotal role in promoting immune tolerance and dampening anti-tumor immunity[18].

While tumor-associated macrophages (TAMs) are abundant in many cancers, most studies on inhibitory receptors have focused on T cells[18]. This oversight is significant, as the high presence of TAMs has the potential to amplify inhibitory receptor signaling. Recent studies have shown that elevated VISTA expression in TAMs correlates with poor prognosis in multiple cancers, including PDAC[19,20]. Despite extensive research on VISTA in other malignancies, its role in regulating macrophage activity and shaping cancer immune responses in PDAC remains poorly understood.

In this study, we define the role of VISTA in regulating anti-tumor immunity by controlling macrophage polarization in PDAC. We show that VISTA deficiency suppresses tumor growth by shifting tumor-associated macrophages toward a pro-inflammatory state and enhancing CD8+ T cell recruitment and function. Consistent patterns observed in human PDAC datasets support the relevance of this immune reprogramming. Collectively, our findings identify VISTA as a key regulator of the PDAC immune microenvironment and highlight its potential as a therapeutic target to improve responses to immunotherapy.

## Results

### VISTA deficiency attenuates pancreatic tumor growth by enhancing CD8+ T cell and macrophage infiltration

To investigate how VISTA deficiency impacts the immune-oncologic landscape of PDAC, we utilized syngeneic orthotopic models to compare tumor growth between wild-type (WT) and VISTA-deficient (*Vsir*−/−) mice. Pan02 and KPC tumor cells ($5 \times 10^5$) were orthotopically inoculated into the pancreatic tail of each group. VISTA deficiency significantly impaired tumor growth in both models, resulting in a 2.5-fold reduction in tumor size in the immunogenic Pan02 model and a 1.8-fold reduction in the less immunogenic KPC model at endpoint (Fig. 1a and Supplementary Fig. 1a). These findings were further supported by pharmacological blockade of VISTA in WT mice, where treatment with an anti-VISTA antibody (αVISTA) similarly suppressed tumor growth in the Pan02 model (Fig. 1b and Supplementary Fig. 2).

VISTA deficiency significantly prolonged survival, with *Vsir*−/− mice surviving approximately 20 days longer than WT controls (Fig. 1c). In human PDAC, lower *VSIR* expression was similarly associated with improved patient survival by Kaplan–Meier analysis of The Cancer Genome Atlas Pancreatic Adenocarcinoma (TCGA-PAAD) cohort (Supplementary Fig. 3a). *VSIR* expression was significantly elevated in pancreatic cancer tissues ($n = 179$) compared to non-malignant pancreatic tissues ($n = 171$) (Supplementary Fig. 3b). Interestingly, *VSIR* expression increased with advancing histological grade (Supplementary Fig. 3c), a trend that was independently validated at the protein level by immunohistochemistry (IHC) and immunofluorescence (IF) analysis of a PDAC tissue microarray (Supplementary Fig. 3d–f). Together, these clinical observations support the relevance of VISTA as a therapeutic target in PDAC.

To characterize the immunological consequences of VISTA deficiency, we conducted flow cytometry and IHC analyses in orthotopic tumor models. *Vsir*−/− mice exhibited a 1.5-fold increase in F4/80+CD11b+ TAMs and a 1.8-fold (Pan02) to 2.0-fold (KPC) increase in CD8+ T cells compared to WT controls (Fig. 1d, e and Supplementary Figs. 1b, c and 4a). IF staining corroborated these findings, revealing increased infiltration of CD8+ T cells (*green/red*) and F4/80+ (*green*) or CD11b+ myeloid cells (*red*) in both *Vsir*−/− and αVISTA-treated tumors compared to WT controls (Supplementary Fig. 5a, b). In contrast, no significant changes were observed in Ly6G+ neutrophils, CD11c+MHCII+ dendritic cells, CD19+ B cells, or FoxP3+CD25+CD4+ Treg cells, underscoring the selective immunomodulatory effects of VISTA (Supplementary Fig. 6a).

Critically, TAMs in *Vsir*−/− tumors acquired a pro-inflammatory phenotype, characterized by increased iNOS+ macrophages (Pan02) and I-A/I-E+CD206− macrophages (KPC) (Fig. 1f, Supplementary Fig. 1d), accompanied by reduced Arginase-1+ macrophages (Fig. 1g). Concurrently, *Vsir*−/− tumors exhibited increased CD8+ T cell infiltration and a shift towards effector or effector memory phenotypes (CD44+CD62L−) (Fig. 1h), along with increased production of IFN-γ and higher frequencies of TNF-α−IFN-γ+ and TNF-α+IFN-γ+ CD8+ T cells (Supplementary Fig. 6b).

To explore clinical relevance of these findings, we performed multivariate logistic regression on the TCGA-PAAD dataset. Elevated *CD8B* expression was associated with 71.6% lower odds of high *VSIR* expression (Supplementary Table 1). Similarly, high expression of *ADGRE1* and *CCR2*–markers of myeloid lineage–were associated with 60.66% and 81.70% reduced odds of elevated *VSIR* expression, respectively. These patterns recapitulate those observed in murine models.

Collectively, these findings position VISTA as a key immunoregulatory checkpoint associated with immunosuppressive macrophage states and impaired CD8+ T cell function in PDAC. Targeting VISTA represents a promising immunotherapeutic strategy to overcome the immunosuppressive TME and enhance anti-tumor immunity in PDAC.

### VISTA^high^F4/80^high^CD11b^high^ tumor-associated macrophages modulate CD8+ T cell dynamics

To delineate the role of VISTA in shaping the PDAC TME, we focused on TAMs and CD8+ T cells. In *Vsir*−/− tumors, we observed a pronounced reduction in the F4/80^high^CD11b^high^ TAM subset, indicating that VISTA is required to maintain this tissue-resident, immunosuppressive macrophage population (Fig. 2a)[21]. Consistent with this, VISTA expression was higher in F4/80^high^CD11b^high^ TAMs than F4/80^int^CD11b^int^ TAMs in WT tumors (Fig. 2b). To test whether selectively depleting these VISTA^high^ TAMs could recapitulate the *Vsir*−/− phenotype, we blocked CSF1R–a key survival signal for tissue-resident macrophages[21]. Mice received 1 mg of anti-CSF1R antibody on day 1, followed by 0.5 mg intraperitoneally for four consecutive days (Supplementary Fig. 7a). By day 28, CSF1R blockade reduced tumor mass significantly (Fig. 2c and Supplementary Fig. 7b), despite leaving the overall frequency of F4/80+CD11b+ TAMs unchanged (Supplementary Fig. 7c). Crucially, the proportion of the F4/80^high^CD11b^high^ TAM subpopulation within the TAM compartment fell sharply (Fig. 2d), mirroring the shift seen in *Vsir*−/− mice. VISTA expression was highest in these F4/80^high^CD11b^high^ TAMs (Fig. 2e), underscoring their functional convergence.

Depletion of VISTA^high^F4/80^high^CD11b^high^ TAMs promoted a marked influx of CD8+ T cells (Fig. 2f) and skewed them toward an effector/effector memory phenotype (CD44+CD62L−; Fig. 2g), while reducing the proportion of exhausted PD-1+TIM-3+ CD8+ T cells (Supplementary Fig. 7d), which are typically associated with impaired cytotoxicity[22,23]. Collectively, these data show that VISTA^high^

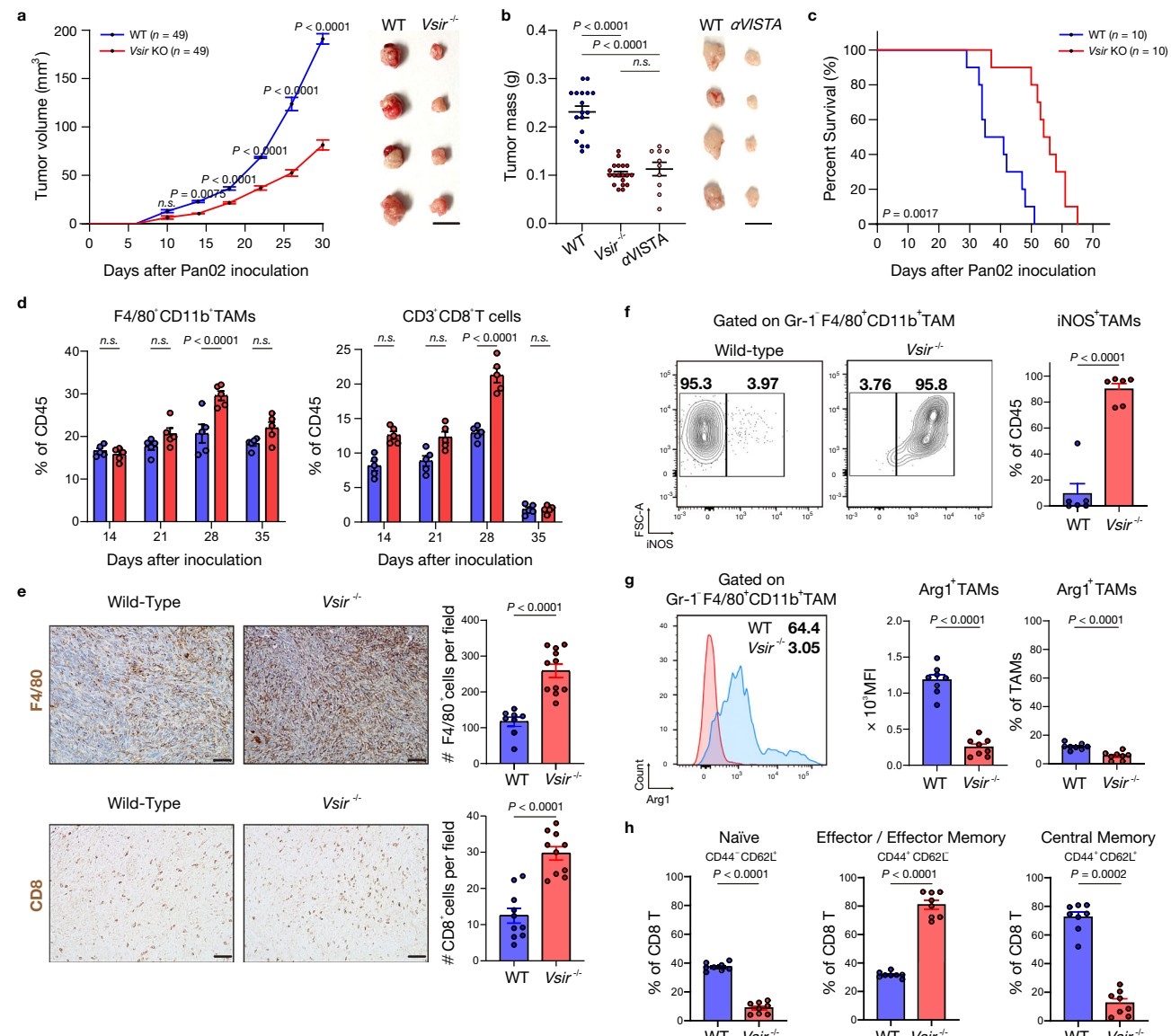

**Fig. 1 | VISTA deficiency attenuates pancreatic tumor growth in *Vsir⁻/⁻* mice.**
**a** Tumor growth curves of wild-type (WT) and *Vsir⁻/⁻* mice orthotopically implanted with Pan02 cells (*n* = 49 per genotype; *n* = 7 per group per time point). Statistical significance at each time point was determined using two-way ANOVA followed by Sidak's multiple-comparisons test. **b** Tumor mass at day 28 in WT (*n* = 17), *Vsir⁻/⁻* (*n* = 19), and αVISTA-treated mice (*n* = 10) (left). Representative ex vivo tumor images are shown (right). Statistical significance was determined using one-way ANOVA with Sidak's multiple-comparisons test (two-sided). **c** Kaplan–Meier survival analysis of WT and *Vsir⁻/⁻* mice (*n* = 10 per group). Statistical significance was determined using two-sided log-rank (Mantel–Cox) test. **d** Flow cytometric quantification of F4/80⁺CD11b⁺ TAMs (left) and CD8⁺ T cells (right) in WT and *Vsir⁻/⁻* mice at days 14, 21, 28, and 35, shown as a percentage of CD45⁺ cells (*n* = 20 biologically independent mice per group). **e** Immunohistochemical images of F4/80⁺ macrophages (top) and CD8⁺ T cells (bottom) in tumors at day 28; cells were quantified per field (*n* = 3 biologically independent mice per group). Scale bar, 50 μm. **f, g** Flow cytometric analysis of iNOS⁺ (*n* = 6 per group) and Arg1⁺ (*n* = 8 per group) TAMs, shown as percentage of CD45⁺ cells or mean fluorescence intensity. **h** Proportions of subsets as a percentage of CD8⁺ T cells (*n* = 8 per group). For (**d**–**h**), statistical significance was determined using unpaired two-sided Student's *t*-tests. All data are presented as mean ± SEM. Exact *P* values are shown in the figures. Experiments were independently repeated at least three times with similar results. αVISTA anti-VISTA; *n.s.* not significant.

macrophages actively suppress CD8⁺ T cell activation in PDAC. Targeting either VISTA or CSF1R disrupts this immunosuppressive axis, leading to depletion of VISTA^high^F4/80^high^CD11b^high^ TAMs and reinvigoration of anti-tumor CD8⁺ T cell responses.

## Tumor-associated macrophage characteristics in VISTA deficiency: chemokine signaling and pro-inflammatory shifts

To uncover the molecular mechanisms by which VISTA regulates tumor immune microenvironment, we performed single-cell RNA sequencing (scRNA-seq) on CD45⁺ immune cells isolated from WT and *Vsir⁻/⁻* Pan02 tumors 28 days post-inoculation. Following quality filtering, we applied uniform manifold approximation and projection

(UMAP) to analyze 20,415 single-cell transcriptome profiles, revealing 23 clusters (Supplementary Fig. 8a) corresponding to nine major immune cell types, including tumor-associated neutrophils (TAN), monocytes/macrophages (Mono/MΦ), dendritic cells (DC), CD8⁺ T cells, proliferating CD8⁺ T cells, CD4⁺ T cells, natural killer (NK) cells, B cells, and mast cells (Supplementary Fig. 8b, c). To ensure balanced comparison between WT and *Vsir⁻/⁻* conditions, we downsampled macrophages to 1365 cells per sample.

Sub-clustering revealed eight macrophage subsets (Fig. 3a, b and Supplementary Fig. 8d): (i) *Ccr2⁺Arg1⁺* Mac (MΦ_cluster0), (ii) *Vegfa⁺Spp1⁺* Mac (MΦ_cluster1), (iii) *Cxcl9*^high^*Pfn*^lo^ Mac (MΦ_cluster2), (iv) *Cxcl9*^low^*Cxcl10⁺Ch25h⁺* Mac (MΦ_cluster3), (v) *Sell⁺Ly6c2⁺* Mono

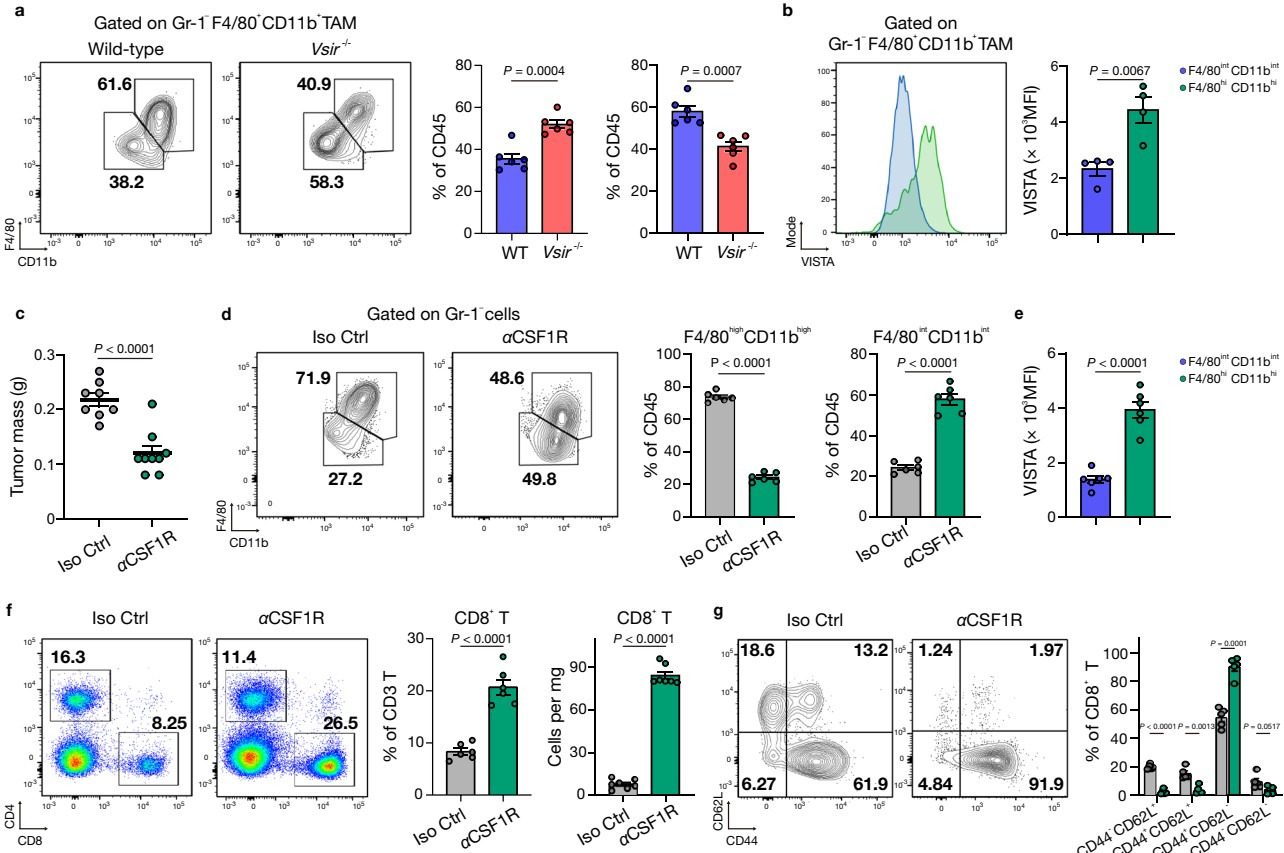

**Fig. 2 | Depletion of VISTA^high^F4/80^high^CD11b^high^ TAMs enhances CD8^+^ T cell responses in _Vsir^-/-^_ mice. a** Flow cytometric analysis of Gr-1^-^F4/80^+^CD11b^+^ TAMs in wild-type (WT) and _Vsir^-/-^_ tumors. Quantification of F4/80^high^CD11b^high^ and F4/80^int^CD11b^int^ TAM subsets shown as a percentage of CD45^+^ cells (_n_ = 6 per group). **b** Representative flow cytometry plots (left) and MFI of VISTA expression (right) on F4/80^high^CD11b^high^ (_green_) and F4/80^int^CD11b^+^ (_blue_) TAMs (_n_ = 5 per group) in WT mice. **c** Tumor mass at day 28 in WT (_n_ = 8) and αCSF1R-treated (_n_ = 9) mice. **d** Flow cytometric analysis of Gr-1^-^F4/80^high^CD11b^high^ and Gr-1^-^F4/80^int^CD11b^int^ TAMs in isotype control (Iso Ctrl) and αCSF1R-treated WT mice (_n_ = 6 biologically independent mice per group). **e** Representative flow cytometry plots (left) and MFI of VISTA expression (right) on F4/80^high^CD11b^high^ (_green_) and F4/80^int^CD11b^+^ (_blue_) TAMs in αCSF1R-treated WT mice (_n_ = 6 biologically independent mice per group). **f** Flow cytometric analysis

of CD4^+^ and CD8^+^ T cells in isotype control and αCSF1R-treated WT mice. Quantification of CD8^+^ T cells shown as a percentage of CD3^+^ T cells and normalized to tumor mass (_n_ = 6 biologically independent mice per group). **g** Flow cytometric analysis of effector memory (CD44^+^CD62L^-^) and central memory (CD44^+^CD62L^+^) CD8^+^ T cells in isotype control and αCSF1R-treated WT mice. Quantification shown as a percentage of CD8^+^ T cells (_n_ = 5 per group). All data are presented as mean ± SEM. Statistical significance was determined using unpaired two-sided Student's _t_-tests for (**a–f**) and one-way ANOVA followed by Sidak's multiple-comparisons test (two-sided) for (**g**). Data for (**a–f**) were obtained from at least three independent experiments; data for (**g**) were obtained from two independent experiments. TAM tumor-associated macrophages, MFI mean fluorescence intensity, αCSF1R- anti-CSF1R; _n.s._ not significant.

(Mono4), (vi) _Tox^+^Ikzf3^+_ Mac (MΦ_cluster5), (vii) _Lyve1^+^Hes^+_ Mac (MΦ_cluster6), and (viii) _Cxcr2^+^Mmp2^+_ Mac (MΦ_cluster7). Notably, _Vsir_ was enriched in MΦ clusters 1, 3, 5, 6, 7, and Mono4, with the highest levels in MΦ_cluster1 (Supplementary Fig. 9a, left). This cluster also showed elevated _Adgre1_ expression, identifying MΦ_cluster1 as the previously described VISTA^high^F4/80^high^CD11b^high^ TAMs (Supplementary Fig. 9a, right).

MΦ_cluster1 exhibited transcriptional signatures associated with immunosuppressive function, including elevated expression of _Vegfa_, _Spp1_, _Havcr2_, _Apoe_, _Trem2_, _Abca1_, and _Mertk_ (Supplementary Figs. 8d and 9b). In contrast, _Vsir^-/-_ macrophages in MΦ_cluster2 showed prominent upregulation of IFN signaling genes (_Stat1_, _Stat2_, _Irf7_, _Irf8_, _Socs1_, _Pim1_, and _Cxcl9_) and chemokines (_Ccl5_, _Ccl7_, _Cxcl9_, and _Cxcl10_), suggesting a shift toward a pro-inflammatory phenotype (Supplementary Fig. 9c). Downregulation of _Nfkb1_ and _Socs3_, key regulators of tumor-promoting and anti-inflammatory signaling respectively[24,25], further supports this phenotype in _Vsir^-/-_ TAM.

Macrophage cluster distribution differed between WT and _Vsir^-/-_ tumors, with significant transcriptional distinctions between _Vsir^-/-_ (_red_) and WT (_blue_) macrophages. MΦ_cluster 2 was almost exclusively present in _Vsir^-/-_ tumor, whereas MΦ_clusters 4, 0, 1, and 3 were prominent

in WT control, with MΦ_cluster1 and MΦ_cluster3 as dominant populations (Fig. 3b, c). _Ccr2_ expression was highest in Mono4 and MΦ_cluster0, while _Cx3cr1_ was elevated in MΦ_clusters 1, 2, and 3 (Supplementary Fig. 8d), suggesting that TAMs in both WT and _Vsir^-/-_ tumors originated from circulating monocytes. However, WT TAMs differentiated predominantly into MΦ_clusters 1 and 3, while _Vsir^-/-_ TAMs favored differentiation into MΦ_cluster2.

We next examined the balance of _Cxcl9_ and _Spp1_ expression across clusters (Fig. 3d). CXCL9 recruits and activates CD8^+^ T cells, while SPP1 (osteopontin) is associated with immunosuppressive function[26,27]. The _Cxcl9:Spp1_ ratio has recently emerged as a predictor of favorable immune responses[28,29]. In our data, _Cxcl9_ expression was significantly upregulated in _Vsir^-/-_ TAMs, likely contributing to enhanced CD8^+^ T cell infiltration. Specifically, _Cxcl9^+^Spp1^+_ TAMs were more prevalent in WT tumor (35.5%) compared to _Vsir^-/-_ tumors (19.0%), while _Cxcl9^+^Spp1^-_ cells were 2.45-fold more frequent in _Vsir^-/-_ TAMs (Fig. 3e). The _Cxcl9:Spp1_ ratio was 2.51 in MΦ_cluster2 _versus_ 0.69 in MΦ_cluster1 (Fig. 3f and Supplementary Fig. 9d), indicating a shift toward a more chemotactic, immunostimulatory TAM phenotype in _Vsir^-/-_ tumors.

These results aligned with human PDAC scRNA-seq data from Bill et al.[28], showing similarity between MΦ_cluster1 and

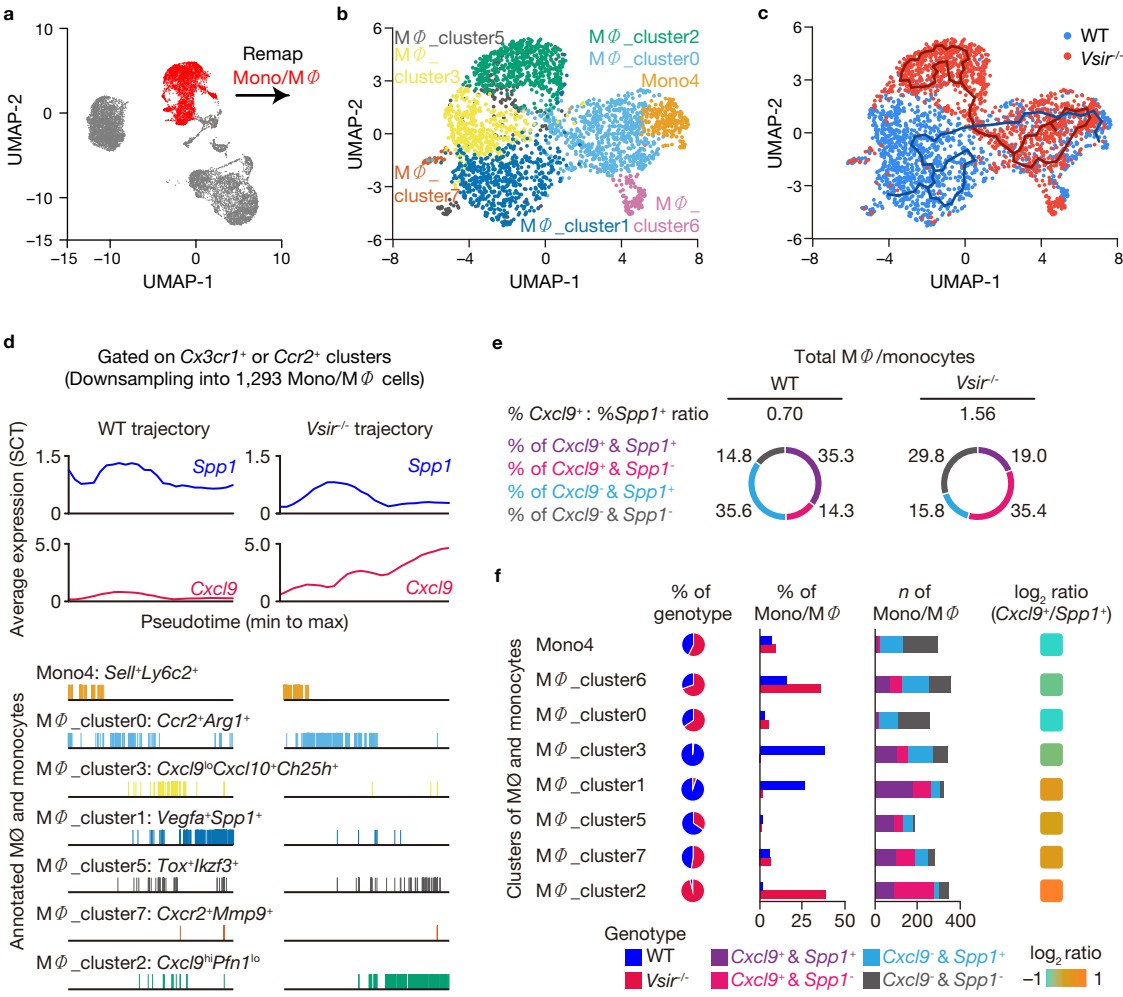

**Fig. 3 | *Cxcl9⁺:Spp1⁺* ratio determines anti-tumor effect between WT and *Vsir⁻/⁻* tumors. a** Uniform manifold approximation and projection (UMAP) visualization of single-cell transcriptomic profiles combined from wild-type (WT) and *Vsir⁻/⁻* tumor samples. Major immune cell lineages are annotated. The Mono/Macrophage cluster used for downstream subclustering is highlighted (as indicated in the figure). **b** UMAP projection of macrophage subclusters in WT and *Vsir⁻/⁻* tumors, annotated by distinct transcriptional states. **c** UMAPs split by genotype showing macrophage subset distribution in WT and *Vsir⁻/⁻* tumors. **d** Pseudotime trajectory analysis (top) of macrophage showing expression dynamics of *Cxcl9* and *Spp1* in WT and *Vsir⁻/⁻*

conditions. Cell density (bottom) is represented by tick marks along the trajectory. **e** Pie charts displaying the proportions of macrophage subsets classified by *Cxcl9* and *Spp1* expression status: (i) *Cxcl9⁺Spp1⁺*, (ii) *Cxcl9⁺Spp1⁻*, (iii) *Cxcl9⁻Spp1⁺*, and (iv) *Cxcl9⁻Spp1⁻*, compared between WT and *Vsir⁻/⁻* tumors. **f** Frequencies and cell numbers of each macrophage subset in WT and *Vsir⁻/⁻* tumors. The *Cxcl9:Spp1* expression ratio is calculated for each subcluster; clusters with the highest and lowest ratios are indicated in *red* and *blue*, respectively. The cumulative proportion of macrophages across all subsets is also compared between WT and *Vsir⁻/⁻* tumors.

immunosuppressive *Spp1⁺* TAMs, and between MΦ_cluster2 and *Cxcl9⁺* TAMs. Gene Set Enrichment Analysis (GSEA) confirmed that *Spp1⁺* TAMs were enriched in MΦ_cluster1 (NES = 1.84, $P < 0.001$), whereas *Cxcl9⁺* TAMs were significantly enriched in MΦ_cluster2 (NES = −2.51, $P < 0.001$) (Supplementary Fig. 9e, top). KEGG pathway analysis further showed that MΦ_cluster2 was enriched for cytokine–cytokine receptor interaction and antigen processing and presentation pathways (Supplementary Fig. 9e, bottom), supporting these TAMs may promote CD8⁺ T cell activation and function.

To validate these findings in human samples, we analyzed the HTAN WUSTL cohort. *VSIR^high* tumors were enriched for anti-inflammatory TAMs (hMΦ.0; *CCL2⁺MAF⁺* and hMΦ.4; *CSTB⁺SPP1⁺*), while *VSIR^low* tumors were dominated by pro-inflammatory monocyte-like TAMs (hMΦ.1; *S100A8/A9*) (Supplementary Fig. 10a–d). These findings suggest that *VSIR* expression in human PDAC is associated with macrophage states enriched for immunosuppressive gene programs, consistent with the immune regulatory patterns observed in murine models.

## Functional enhancement of TAMs in VISTA deficiency: antigen cross-presentation and CD8⁺ T cell activation

To assess the functional impact of VISTA deficiency on macrophages, we analyzed several characteristics. While WT and *Vsir⁻/⁻* BMDMs showed comparable phagocytic ability (Supplementary Fig. 11a), transcriptomic and pathway analyses revealed notable changes in immunomodulatory capacity. Consistent with these transcriptional differences, phospho-protein profiling of WT and *Vsir⁻/⁻* BMDMs under matched stimulation conditions revealed distinct phosphorylation patterns across multiple signaling nodes (Supplementary Fig. 12a, b), indicating altered intracellular signaling states in the absence of VISTA. Principal Component Analysis (PCA) highlighted distinct transcriptional profiles among MΦ_clusters 6, 0, and 4, while MΦ_clusters 1, 2, and 3 were closely related, with MΦ_clusters 2 and 3 particularly similar (Supplementary Fig. 13a). The largest variance (PC1) separated Mono4 and MΦ_cluster6 from other clusters. Given the distinct profiles between MΦ_cluster2 and MΦ_cluster3 in *Vsir⁻/⁻* tumor *versus* WT control, we focused on their functional characteristics and spatial proximity to CD8⁺ T cells.

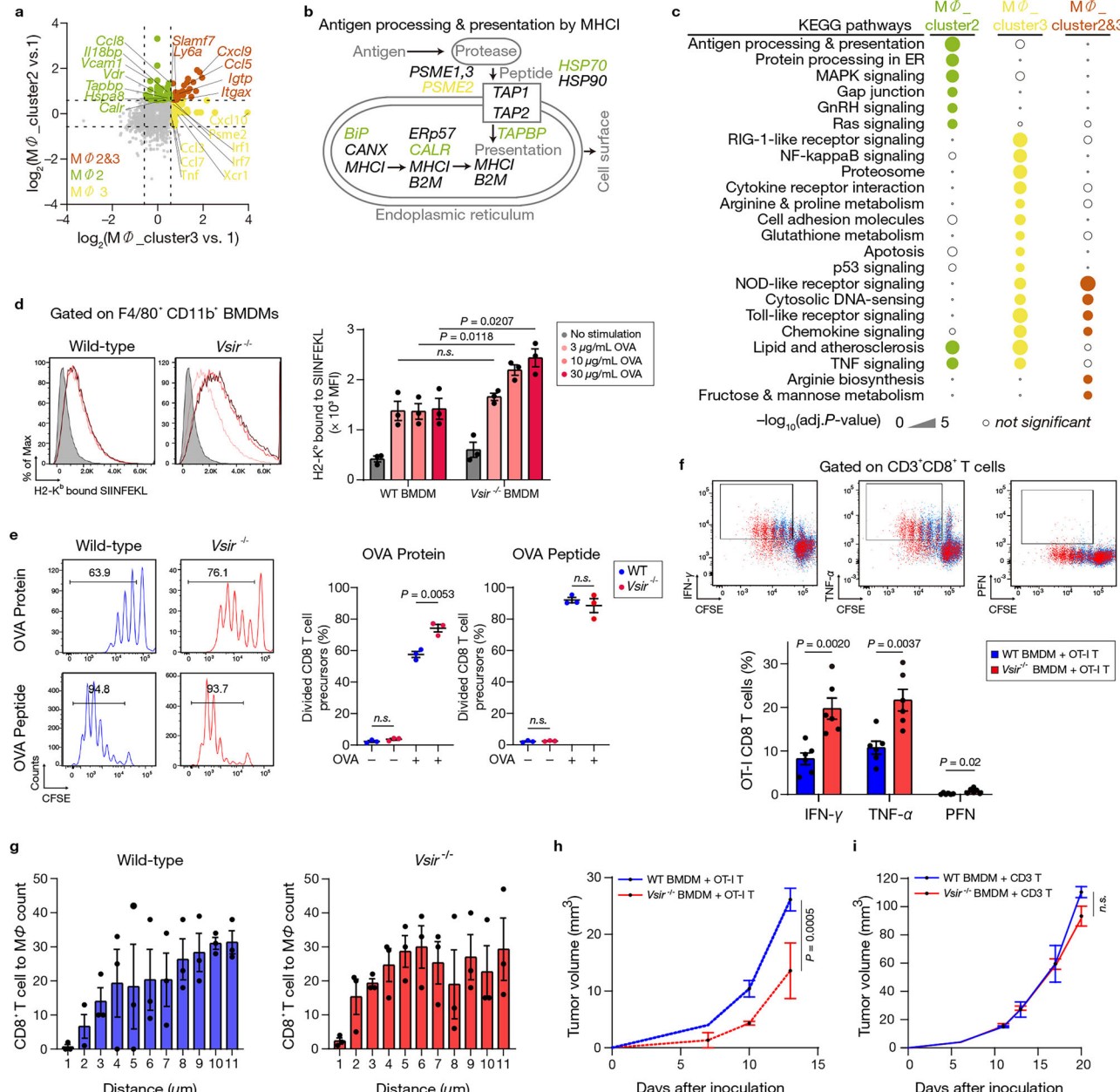

**Fig. 4 | *Vsir* deficiency enhances antigen presentation and stimulates CD8⁺ T cell responses. a** Differential gene expression analysis of macrophage subsets. Volcano plot comparing MΦ_clusters 2 and 3 with MΦ_cluster1. **b** Schematic of the MHC class I antigen processing and presentation pathway. **c** KEGG pathway analysis of MΦ_cluster2, MΦ_cluster3, and combined MΦ_clusters 2 and 3 ($p$_adj <0.05). **d** Flow cytometric analysis of SIINFEKL–H2-K$^b$ complex on wild-type (WT) and *Vsir*$^{-/-}$ bone marrow-derived macrophages (BMDM) stimulated with ovalbumin (OVA) protein ($n = 3$ independent BMDM differentiations). MFI quantification shown (right). **e** Proliferation of OT-I CD8⁺ T cells co-cultured with WT or *Vsir*$^{-/-}$ BMDMs loaded with OVA protein (top) or SIINFEKL peptide (bottom) ($n = 3$ biologically independent BMDM differentiations per group). **f** Flow cytometric analysis of IFN-γ, TNF-α, and perforin (PFN) production in OT-I CD8⁺ T cells co-cultured with WT or *Vsir*$^{-/-}$ BMDMs loaded with OVA protein ($n = 3$ independent biological replicates). **g** Quantification of CD8⁺ T cells in proximity to macrophages across varying distances (μm). $n = 3$ biologically independent mice per group. **h** Tumor growth curves following co-injection Pan02 cells with OT-I CD8⁺ T cells and WT or *Vsir*−/− BMDMs ($n = 5$ per group). **i** Tumor growth curves following co-injection of BMDMs and CD3 T cells ($n = 6$ per group). All data are presented as mean ± SEM. Statistical significance for (**d**–**f**) was determined using unpaired two-sided Student's $t$-tests. Statistical significance for (**h** and **i**) was determined using two-sided Student's $t$-tests at the endpoint. Exact $P$ values are shown in the figures. *n.s.* not significant. Experiments were independently repeated at least three times with similar results. MFI mean fluorescence intensity, CFSE carboxyfluorescein succinimidyl ester.

MΦ_cluster2 showed high expression of *Ccl8, Il18bp, Vcam1, Vdr, Tapbp, Hspa8,* and *Calr*, while MΦ_cluster3 expressed *Ccl3, Ccl7, Tnf, Irf1, Irf7,* and *Cxcl10*. Both clusters shared expression of *Slamf7, Cxcl9, Ccl5,* and *Itgax* (Fig. 4a and Supplementary Fig. 13b). KEGG pathway analysis revealed that MΦ_cluster2 was enriched in pathways related to antigen processing and presentation, protein processing in the endoplasmic reticulum, and MAPK and Ras signaling. In contrast, MΦ_cluster3 was enriched in NF-κB signaling, cytokine-cytokine receptor interactions, arginine and proline metabolism, apoptosis, and TNF signaling (Fig. 4b, c). These findings suggest that MΦ_cluster2 is more specialized for antigen presentation and may play a key role in activating CD8⁺ T cells in VISTA-deficient tumors.

Module score analysis confirmed that MΦ_cluster2 had significantly higher expression of antigen presentation-related genes

compared to MΦ_clusters 1 and 3 (Supplementary Fig. 13c). Transcriptomic profiling further showed increased expression of MHC class I pathway components, including *Tapbp, Tap1, Tap2*, and *Hsp70*, in MΦ_cluster2 from *Vsir*[-/-] TAMs (Fig. 4b). These findings were validated by RT-qPCR, which showed upregulated *Tapbp* and *Tap1* in *Vsir*[-/-] BMDMs following OVA and LPS stimulation (Supplementary Fig. 11b, c).

To directly test antigen presentation capacity, we performed an antigen cross-presentation assay by measuring SIINFEKL–H2-K[b] complex on WT and *Vsir*[-/-] BMDMs. Upon SIINFEKL peptide loading, no difference in MFI was observed (Supplementary Fig. 11d). However, *Vsir*[-/-] BMDMs exposed to full-length OVA protein displayed significantly higher SIINFEKL–H-2K[b] levels, especially at higher antigen concentrations (Fig. 4d), indicating enhanced antigen processing and cross-presentation in the absence of VISTA. Notably, this occurred without changes in CD80, CD86, or I-A/I-E expression (Supplementary Fig. 14), suggesting that VISTA loss selectively enhances antigen-presenting function without broadly activating macrophages.

In functional co-culture assays, *Vsir*[-/-] BMDMs stimulated robust proliferation of OT-I CD8[+] T cells by day 3 when loaded with full-length OVA, underscoring their superior antigen-presenting function in physiologically relevant conditions (Fig. 4e, top). These co-cultures also showed higher IFN-γ, TNF-α, and perforin production in OT-I T cells (Fig. 4f). Notably, enhanced IFN-γ production in the *Vsir*[-/-] BMDM co-cultures was accompanied by increased CXCL9 output (Supplementary Fig. 15), consistent with activation of an IFN-γ−CXCL9 chemokine axis in this setting. In contrast, SIINFEKL peptide stimulation produced similar CD8[+] T cell proliferation regardless of macrophage genotype (Fig. 4e, bottom), emphasizing the advantage of *Vsir*[-/-] macrophages in antigen processing rather than presentation alone. Importantly, no differences were observed with WT *versus Vsir*[-/-] splenic DCs (Supplementary Fig. 11e), indicating VISTA's modulatory role is specific to macrophages.

Spatial analysis revealed CD8[+] T cells were closer proximity to TAMs in *Vsir*[-/-] tumors (median distance: 3–4 μm) compared to WT tumors (>10 μm), suggesting enhanced macrophage−T cell crosstalk (Fig. 4g). This was further supported by increased CD3[+] T cell−CD11b[+] myeloid interactions in *Vsir*[-/-] tumor cores. To functionally validate these findings in vivo, we used a co-reconstitution model in *Rag1*[-/-] mice. Co-injection of *Vsir*[-/-] BMDMs and OT-I CD8[+] T cells with Pan02-OVA cells significantly reduced tumor growth compared to WT BMDMs (Fig. 4h). No difference was observed with polyclonal CD3[+] T cells (Fig. 4i), underscoring the antigen-specific nature of this response. Collectively, VISTA deficiency enhances TAM antigen processing/presentation, driving CD8[+] T cell activation and potent anti-tumor immunity.

### Reduced exhaustion and enhanced effector function of CD8[+] T cells in VISTA-deficient tumors

To explore how VISTA deficiency influences CD8[+] T cell function within the TME, we examined the balance between effector function and exhaustion in CD8[+] T cells. This balance is crucial for effective anti-tumor immunity, and understanding it may offer insights into enhancing immune responses. Using unsupervised network-based clustering, we identified six distinct CD8[+] T cell subclusters (Fig. 5a–c): (i) *Cx3cr1*[high]*Klrc1*[lo] CD8[+] T cells (Exhausted Intermediate, CD8[+] T_cluster0), (ii) *Cx3cr1*[lo]*Klrc1*[high] CD8[+] T cells (Exhausted Terminal, CD8[+] T_cluster1), (iii) *Cx3cr1*[high]*Klrc1*[high] CD8[+] T (Exhausted-KLR, CD8[+] T_cluster2), (iv) *Cxcr3*[+]*Slamf7*[+] CD8[+] T cells (CD8[+] T_cluster3), (v) *Fos*[+]*Gab2*[+] CD8[+] T cells (CD8[+] T_cluster4), and (vi) *Maf*[+]*Il2ra*[+] CD8[+] T cells (CD8[+] T_cluster5). By day 28 post-tumor cell inoculation, no naïve CD8[+] T cells (identified by *Lef1, Ccr7, Tcf7*, and *Sell*) were detected, suggesting extensive tumor-driven T cell differentiation. Most CD8[+] T cells exhibited a resident phenotype (*Cxcr6, Cd69*, and *Runx3*). Moreover, CD8[+] T_clusters 0, 1, and 2 expressed genes associated with cytotoxicity (*Gzmb, Ifng, Cst7, Prf1*, and *Nkg7*) as well as exhaustion markers (*Ctla4, Tigit, Pdcd1, Lag3*, and *Havcr2*) (Supplementary Fig. 16a).

CD8[+] T_clusters 0 and 3 were predominantly derived from *Vsir*[-/-] tumor, whereas clusters 1 and 2 were enriched in CD8[+] T cells from WT tumor (Fig. 5c, d). To further characterize their functional states, we quantified exhaustion and effector phenotypes using established gene signatures. CD8[+] T_clusters 1 and 2 contained higher proportions of *Cd38*[+]*Eomes*[+] cells (22.5% and 44.0%) and *Cd226*[-]*Pdcd1*[+] cells[30] (41.0% and 44.3%), respectively, compared to CD8[+] T_clusters 0 and 3 (*Cd38*[+]*Eomes*[+]: 14.0% and 5.0%; *Cd226*[-]*Pdcd1*[+]: 31.5% and 25.3%) (Fig. 5e and Supplementary Fig. 17a–c). Flow cytometric analysis demonstrated that WT tumors contained a higher cumulative burden of exhaustion-associated inhibitory receptors on CD8[+] T cells, as assessed by the co-expression of PD-1, TIM-3, LAG-3, and TIGIT (Supplementary Fig. 17d). In particular, CD8[+] T cells expressing three or more exhaustion markers were enriched in WT tumors compared with *Vsir*[-/-] tumors, consistent with a greater prevalence of terminal exhaustion−like CD8[+] T cell states. In contrast, *Vsir*[-/-] tumors exhibited a relative reduction in these highly exhausted populations, accompanied by increased TNF-α and IFN-γ production (Supplementary Fig. 6b).

Consistently, *Vsir* expression was significantly correlated with key exhaustion markers (*Havcr2, Tigit, Lag3, Pdcd1, Cxcl13*, and *Layn*) in both TCGA-PAAD tumor/normal samples and GTEx normal pancreas samples (Supplementary Fig. 16b), suggesting a potential association between VISTA and T cell exhaustion within the TME. GSEA showed enrichment of oxidative phosphorylation and[30] cytokine-cytokine interaction pathways in CD8[+] T_cluster3 (Supplementary Fig. 18). Given that oxidative phosphorylation is often reduced in chronically exhausted T cells, this suggests that CD8[+] T cells in *Vsir*[-/-] tumor exhibit a unique metabolic adaptation that supports their effector function[31,32]. In contrast, exhausted CD8[+] T cells in WT tumor likely suffer from impaired mitochondrial function and reduced oxidative phosphorylation, further contributing to their exhausted phenotype.

Flow cytometric analysis further characterized the altered state of CD8[+] T cells in *Vsir*[-/-] tumor. *Vsir*[-/-] tumor had significantly increased expression of CXCR3, with 28.3% of CD8[+] T cells co-expressing CXCR3 and CX₃CR1 compared to 8.07% in WT tumor (Fig. 5f). Additionally, *Vsir*[-/-] tumor had a higher proportion of TCF-1[+]PD-1[+] CD8[+] T cells and fewer terminally exhausted CXCR3[+]CX₃CR1[-] CD8[+] (Tex[term]) cells compared to WT control (Fig. 5g, h). These findings suggest that VISTA deficiency promotes a CD8[+] T cell phenotype with reduced exhaustion and improved effector function within the TME.

To extend these findings to human disease, multiplex IF analysis of a limited number of PDAC specimens revealed that, although overall CD8[+] T cell infiltration showed an increasing trend with advancing tumor grade, the fraction of CXCR3[+] cells within the CD8[+] T cell compartment was significantly reduced in grade 2 and grade 3 tumors compared with grade 1 tumors (Supplementary Fig. 19a, b), demonstrating an inverse relationship between tumor grade and CXCR3 expression. Notably, spatial analysis further showed a decreased nearest-neighbor distance between VISTA[+]CD68[+] macrophages and CD8[+] T cells in grade 2 and grade 3 tumors, indicating increased spatial proximity in higher-grade disease (Supplementary Fig. 19c).

Finally, we validated these observations in the HTAN WUSTL. The *VSIR*[low] group exhibited increased frequencies of effector (hCD8.1; CCL4[high]) and effector-like (hCD8.2; KCNQ1OT1[+]) CD8[+] T cells, while the *VSIR*[high] group was enriched for pre-effector (hCD8.0; GZMK[+]GPR183[+]) and exhausted (hCD8.7; CXCL13[high]) CD8[+] T cell subsets (Supplementary Figs. 10a and 20a–c). Together, these results confirm that low VISTA expression is associated with enhanced CD8[+] T cell effector function and reduced exhaustion in human pancreatic tumors.

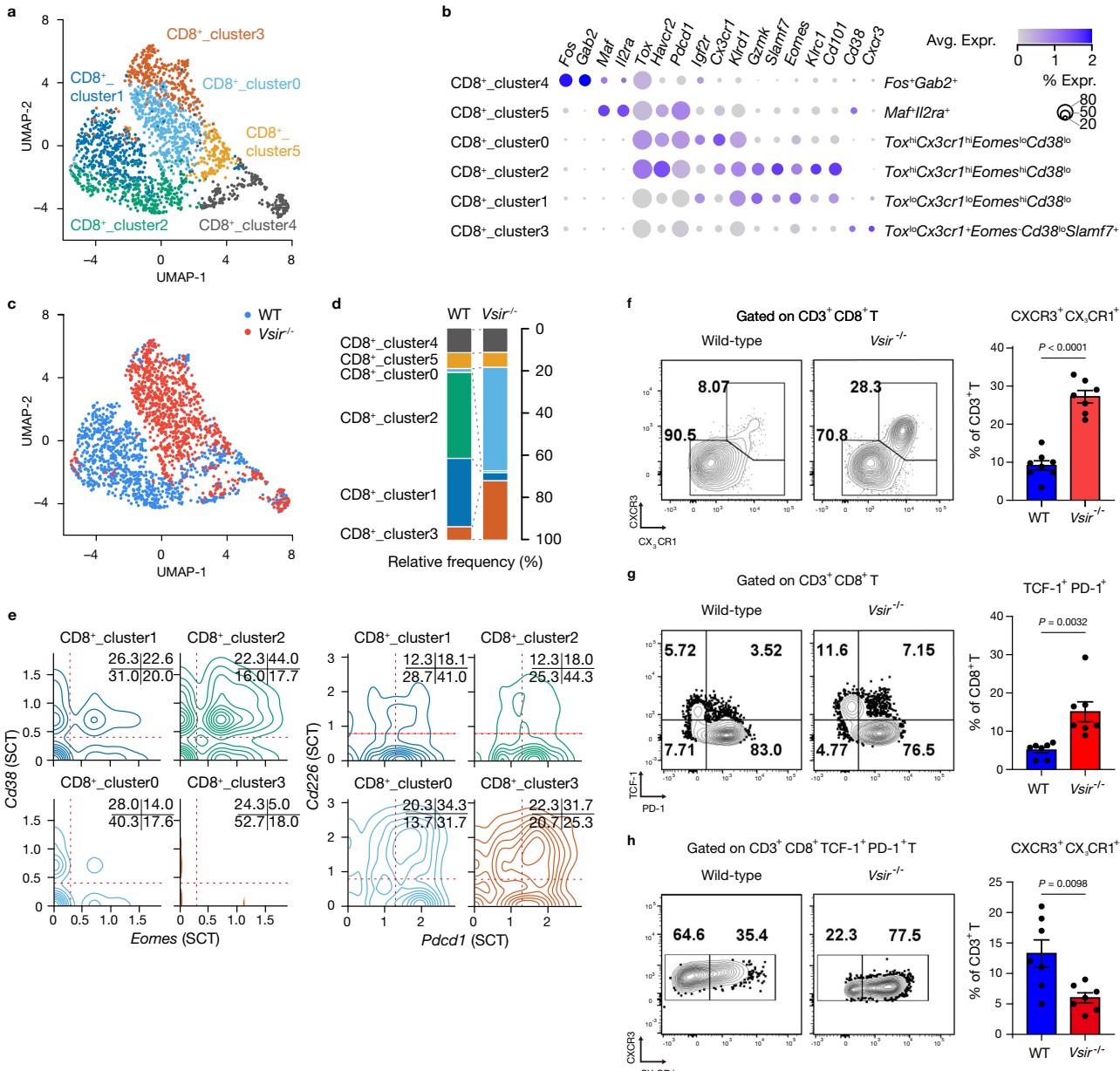

**Fig. 5 | *Vsir* deficiency delays CD8⁺ T cell exhaustion with increased CXCR3 expression. a** Uniform manifold approximation and projection (UMAP) of single-cell RNA-seq data showing CD8⁺ T cell subsets in wild-type (WT) and *Vsir⁻/⁻* tumors; subpopulations are color-coded. **b** Dot plot of differentially expressed genes across six CD8⁺ T cell clusters, with color and dot size representing expression level and cell proportion, respectively. **c** Genotype-split UMAP depicting CD8⁺ T cells from WT (*blue*) and *Vsir⁻/⁻* (*red*) tumors. **d** Relative abundance of each CD8⁺ T cell subsets in WT and *Vsir⁻/⁻* tumors. **e** Scatter plot of log₂ fold changes: CD8⁺_cluster1 (*x*-axis) *versus* CD8⁺_cluster2 (*y*-axis), each compared with *Vsir*-deficient clusters (CD8⁺_cluster0 and cluster3). Genes upregulated only in CD8⁺_cluster1 (*gold*), only

in CD8⁺_cluster2 (*green*), or in both WT-specific clusters (*blue*) are highlighted. Data were normalized using SCTransform. **f** Flow cytometry plots (left) and quantification (right) of CXCR3⁺CX₃CR1⁺ CD8⁺ T cells in WT ($n = 8$) and *Vsir⁻/⁻* ($n = 7$) tumors. **g** Flow cytometry plots (left) and quantification (right) of TCF-1⁺PD-1⁻ CD8⁺ T cells in WT and *Vsir⁻/⁻* tumors ($n = 7$ per group). **h** Flow cytometry plots (left) and quantification (right) of terminally exhausted TCF-1⁻PD-1⁺CXCR3⁺CX₃CR1⁻ CD8⁺ T cells (Texᵗᵉʳᵐ) in WT and *Vsir⁻/⁻* tumors ($n = 7$ per group). All data are presented as mean ± SEM. Statistical significance for (**f–h**) was determined using unpaired two-sided Student's *t*-tests. *n.s.* not significant. Experiments were independently repeated at least three times with similar results.

## *Cxcl9-Cxcr3* axis in VISTA deficiency: enhanced CD8⁺ T cell interaction with TAMs

To explore how VISTA-deficient macrophages and CD8⁺ T cells communicate within the TME, we conducted an intercellular communication network analysis using CellChat[33]. Mono/Mac played a central role, especially in *Vsir⁻/⁻* tumor, where they demonstrated enhanced interactions with CD8⁺ T cells (Fig. 6a). The increased node size representing Mono/Mac highlighted their importance in driving anti-tumor immune responses in pancreatic tumors.

Using CellChatDB, we analyzed ligand-receptor interactions, focusing on Mono/Mac as the primary signal sender. The out-degree magnitude represented the cumulative probability for signals sent by Mono/Mac, while the in-degree represented signals received by CD8⁺ T cells. Unique ligand-receptor interactions were observed in *Vsir⁻/⁻* tumor. The MΦ_cluster2 macrophage subtype, exclusive to VISTA deficiency, showed increased expression of *Cxcl9* and *Cxcl4 (Pf4)*, which are likely to interact with *Cxcr3* on CD8⁺ T_cluster3, predominantly found in *Vsir⁻/⁻* condition (Fig. 6b, left and d). This interaction suggests that MΦ_cluster2 macrophages in *Vsir⁻/⁻* tumor create

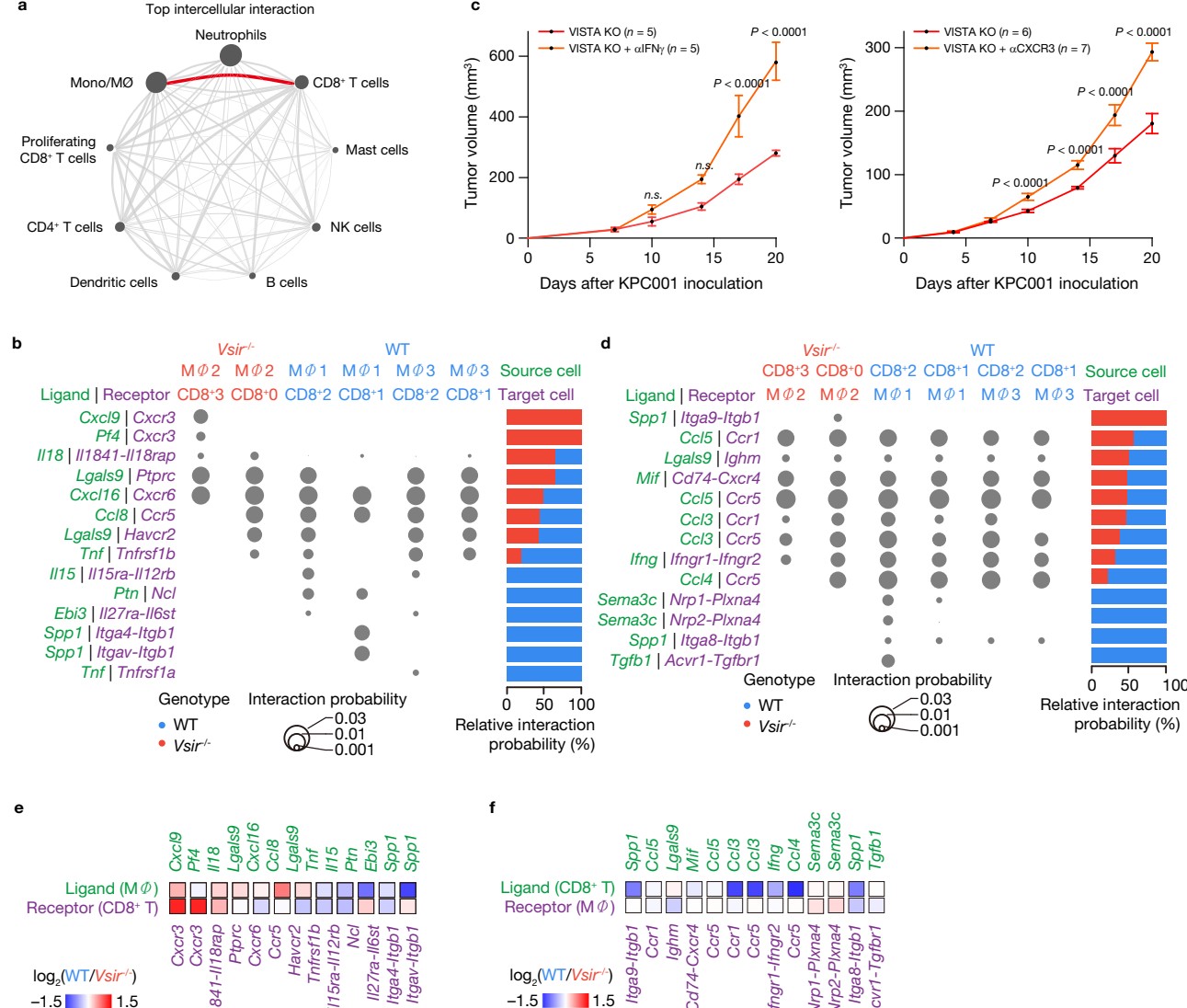

**Fig. 6 | Intercellular communication between macrophages and CD8+ T cells in VISTA-deficient tumors. a** Network diagram showing top predicted intercellular interactions between cell types in wild-type (WT) and *Vsir⁻/⁻* tumors using NicheNet analysis. The most prominent interaction between Mono/Mac and CD8+ T cells is highlighted in *red*. **b** Dot plot showing predicted ligand-receptor interactions from macrophages (source) to CD8+ T cells (target) in WT and *Vsir⁻/⁻* tumors. Dot size reflects interaction probability, and color intensity reflects ligand activity. **c** Longitudinal tumor growth curves following anti-IFN-γ antibody (left; *n* = 5 per group) or anti-CXCR3 (right; *n* = 6 control *Vsir⁻/⁻* mice and *n* = 7 anti-CXCR3-treated mice) treatment in *Vsir⁻/⁻* tumor-bearing mice. Data are presented as mean ± SEM.

Statistical significance at each time point was determined using two-way ANOVA with Sidak's multiple-comparisons test. Exact *P* values are shown in the figure. **d** Dot plot showing predicted ligand-receptor interactions from CD8+ T cells (source) to macrophages (target) in WT and *Vsir⁻/⁻* tumors. **e, f** Heatmaps displaying relative expression levels of ligand-receptor pairs across clusters with **e** corresponding to macrophage-to-CD8+ T cell interactions and **f** to CD8+ T cell-to-macrophage interactions. Log₂ fold change values are color-coded, with red indicating enrichment in *Vsir⁻/⁻* and *blue* in WT. Mono/MΦ monocytes and macrophages, αIFNγ anti-IFN-gamma, αCXCR3 anti-CXCR3.

an inflammatory environment, enhancing CD8+ T cell recruitment and activation, which strengthens the anti-tumor response. Specifically, the *Cxcl9-Cxcr3* axis facilitates CD8+ T cell infiltration and activation, supporting efficient tumor clearance[34].

We evaluated whether the anti-tumor effect of *Vsir* deletion was mediated by IFN-γ–CXCL9–CXCR3 axis by administering a neutralizing anti−IFN-γ antibody or anti-CXCR3 antibody to VISTA-KO mice. Anti-IFN-γ treatment abolished the reduced tumor growth seen in *Vsir⁻/⁻* mice. Likewise, blockade of CXCR3 also reversed the enhanced tumor control in *Vsir⁻/⁻* hosts. These results indicate that the protective effect of VISTA loss depends on the IFN-γ−CXCL9−CXCR3 pathway (Fig. 6c and Supplementary Fig. 21). In addition, both WT and *Vsir⁻/⁻* tumors exhibited *Cxcl16-Cxcr6* and *Ccl8-Ccr5* interactions, supporting tissue repair and immune regulation, respectively (Fig. 6b)[35,36].

We quantified the interaction probability from macrophages to CD8+ T cells by converting the total sum of interaction probabilities between WT and *Vsir⁻/⁻* tumors to 100%. In *Vsir⁻/⁻* tumor, Mono/Mac exhibited 100% relative interaction probability via the *Cxcl9-Cxcr3* and *Pf4-Cxcr3* ligand-receptor pairs, and over 50% for *Il18-Il18r1-Il18rap* and *Lgals9-Ptprc* interactions (Fig. 6b, right and e). These interactions are critical for immune activation and chemokine-mediated recruitment of effector T cells[34,37–39]. In contrast, WT Mono/Mac were predicted to interact through ligand-receptor pairs such as *Il15-Il15ra-Il2rb*, *Ptn-Ncl*, *Ebi3-Il27ra-Il6st*, *Spp1-Itga4-Itgb1*, *Spp1-Itgave-Itgb1*, and *Tnf-Tnfrsf1a*, suggesting a more immuno-suppressive role[40]. Both WT and *Vsir⁻/⁻* tumors also exhibited predicted interactions through *Cxcl16-Cxcr6*, *Ccl8-Ccr5*, and *Lglas9-Havcr2*.

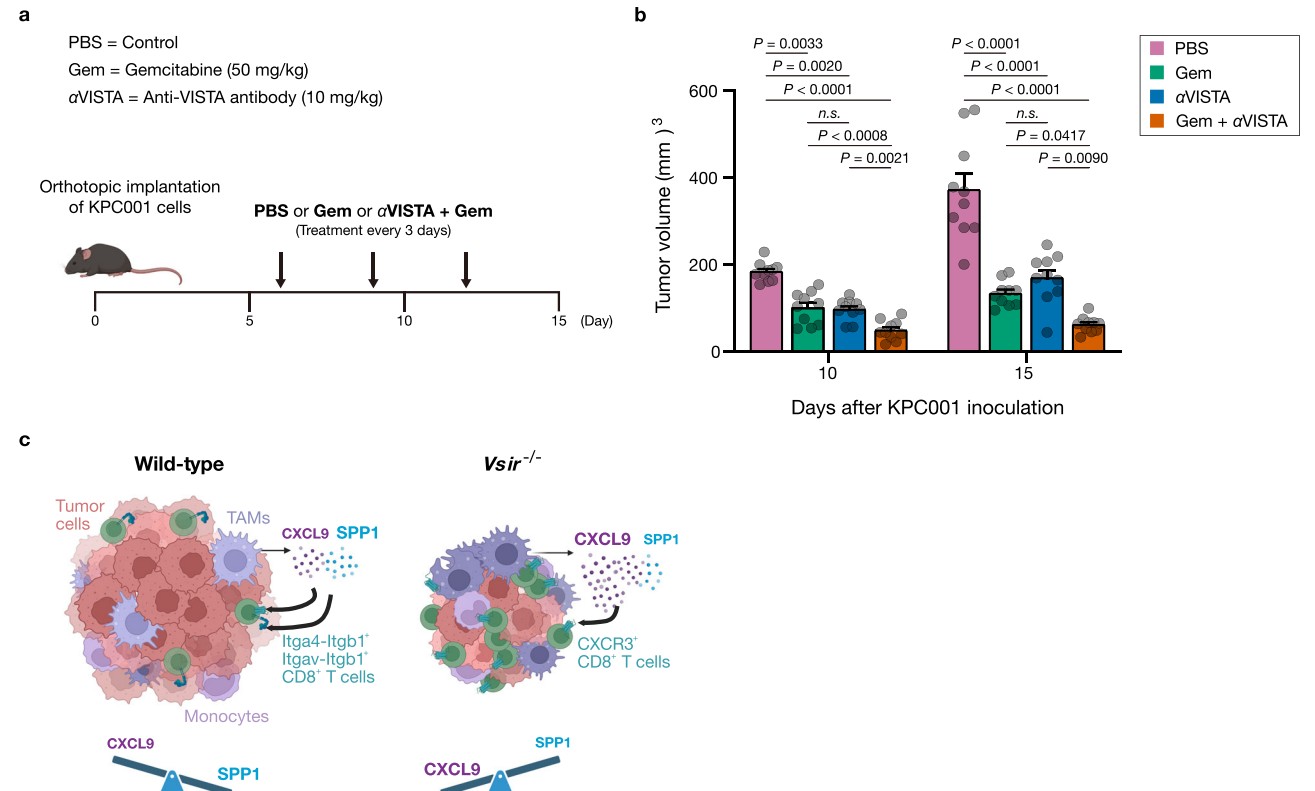

**Fig. 7 | Effect of anti-VISTA antibody in combination with gemcitabine therapy.** **a** Schematic of the treatment regimen for combination therapy with anti-VISTA antibody and gemcitabine (Gem). Created in BioRender. Kim, H. (2026) https://BioRender.com/ng9t0q2. **b** Tumor volumes in mice treated with phosphate-buffered saline (PBS) control, anti-VISTA antibody, gemcitabine, or combination therapy. (n = 10 per group). Statistical significance was determined using one-way ANOVA followed by Sidak's multiple comparisons test (two-sided). Exact P values are shown in the figure. P values less than the detection limit (0.0001) are indicated as P < 0.0001. Data are presented as mean ± SEM. Data are representative of at least three independent experiments. **c** Schematic summary of key ligand-receptor interactions and immune cell communication pathways in wild-type (WT) *versus* VISTA-deficient tumors, highlighting the enhanced crosstalk in the absence of VISTA. Created in BioRender. Shin, G. (2026) https://BioRender.com/x38j745. TAM tumor-associated macrophages, *n.s.* not significant.

---

Interestingly, unique interactions were observed in WT tumor, where MΦ_cluster1 was predicted to interact with CD8+ T_cluster1 through *Spp1* and its potential partners, *Itga4-Itgb1* or *Itgav-Itgb1* on CD8+ T cells. This interaction, absent in MΦ_cluster3, highlights the role of *Spp1* in maintaining an immunosuppressive TME.

We also identified reciprocal interactions, with CD8+ T cells acting as the source and Mono/Mac as the target (Fig. 6d). In both WT and *Vsir−/−* tumors, CD8+ T cells expressed *Ccl3* and *Ccl5*, which potentially interact with *Ccr5* on macrophages, forming a feedback loop that amplifies immune activation (Fig. 6d, left and f). Additionally, CD8+ T cells expressed *Ifng*, potentially interacting with *Ifngr1* and *Ifngr2* on macrophages, establishing a pro-inflammatory feedback loop that activates macrophages in both tumor types. However, WT CD8+ T cells uniquely expressed interactions involving *Sema3c-Nrp1-Plxna4*, *Sema3c-Nrp2-Plxna4*, and *Spp1-Itga8-Itgb1*, suggesting distinct suppressive signaling.

*Vsir−/−* CD8+ T cells demonstrated 100% interaction probability through the *Spp1-Itga9-Itgb1* pair, with strong interactions predicted via *Ccl5-Ccr1*, *Lgals9-Ighm*, *Mif-Cd74-Cxcr4*, and *Ccl5-Ccr5* (Fig. 6d, right and f). These interactions indicate a pro-inflammatory environment facilitated by VISTA deficiency. The shift toward pro-inflammatory signaling aligns with the enhanced immune response in *Vsir−/−* tumor, where CD8+ T cells exhibit robust anti-tumor activity.

### VISTA blockade synergizes with gemcitabine to remodel the pancreatic tumor immune landscape

To determine whether these VISTA-dependent signaling network we identified is therapeutically exploitable, we turned to the KPC orthotopic model and compared anti-VISTA monotherapy, gemcitabine, and their combination (Fig. 7a). Both monotherapies produced a modest but significant reduction in tumor burden; however, the combination elicited a markedly greater suppression, indicating at least an additive—and potentially synergistic—anti-tumor effect (Fig. 7b).

Immune profiling revealed that anti-VISTA monotherapy alone increased total TAMs and increased CD8+ T cells (Supplementary Fig. 22a). Importantly, the antibody skewed TAMs toward an iNOS+ phenotype while producing only minor changes in Arg1+ TAMs (Supplementary Fig. 22b). Consistent with our CellChat predictions, CXCL9+/SPP1+ TAMs were enriched in the anti-VISTA treatment group (Supplementary Fig. 22c), both anti-VISTA and the combination regimen boosted infiltration of CXCR3+ CD8+ T cells (Supplementary Fig. 22d)—hallmarks of the CXCL9–CXCR3 chemotactic axis we mapped in *Vsir−/−* tumors. These findings validate our mechanistic model and demonstrate that pharmacologic VISTA blockade can reprogram the myeloid-T-cell circuit even in the presence of cytotoxic chemotherapy.

Taken together, our data position VISTA as a central architect of immunosuppressive niches in pancreatic tumors. In WT tumors, TAMs co-express *Cxcl9* and *Spp1*: engagement of SPP1 with integrins *Itga4/Itgb1* or *Itgav/Itgb1* on CD8+ T cells likely dampens T cell recruitment and activation. By contrast, VISTA deficiency—or therapeutic blockade—tilts this balance toward CXCL9 dominance, amplifying the CXCL9–CXCR3 axis and fostering a pro-inflammatory TME that supports robust CD8+ T cell activity (Fig. 7c). Thus, combining VISTA inhibition with standard chemotherapy offers a rational strategy to overcome immune suppression and improve anti-tumor effect.

## Discussion

TAMs are key components of the TME that contribute significantly to cancer progression, including PDAC[41,42]. Their plasticity allows them to polarize into either a tumor-suppressive or tumor-promoting state depending on environmental cues[43–45]. In PDAC, most TAMs adopt an immunosuppressive phenotype that fosters tumor growth and sustains a tolerogenic TME[41]. Understanding factors that control TAM polarization is therefore crucial for developing therapeutic strategies that target this influential cell population[46].

Our study highlights a close association between VISTA expression and macrophage state composition in PDAC; how this relationship is established within the tumor microenvironment warrants further investigation[47,48]. We found that VISTA deficiency or inhibition in orthotopic PDAC mouse models significantly reduced tumor growth. This reduction was associated with increased infiltration of TAMs and CD8[+] T cells, accompanied by a pro-inflammatory shift in the TME, evidenced by an elevated $Cxcl9{:}Spp1$ ratio in VISTA-deficient TAMs.

CXCL9/10-engineered DCs have previously been shown to suppress tumor growth in NSCLC murine models, emphasizing a crucial role of CXCL9 in anti-tumor immune responses[49]. In PDAC, however, VISTA's predominant expression in TAMs creates distinct microenvironment dynamics. Our murine model revealed VISTA deficiency drives TAM polarization towards a $Cxcl9^+$ state while lowering $Spp1^+$ state. Increased $Cxcl9$ levels and reduced $Spp1$ expression collectively pointed to the reprogramming of TAMs in the absence of VISTA, disrupting the immunosuppressive microenvironment typically seen in PDAC.

The $Cxcl9{:}Spp1$ ratio was significantly elevated in TAMs from pancreatic tumor-bearing $Vsir^{-/-}$ mice, correlating with a shift towards a pro-inflammatory phenotype. Cxcl9, a chemokine that recruits CXCR3-expressing CD8[+] T cells, is known to enhance anti-tumor activities, while SPP1 is linked to immunosuppressive TAMs and tumor progression[50,51]. Recent research has indicated that the $Cxcl9{:}Spp1$ ratio may more effectively define macrophage polarity in vivo compared to traditional M1/M2 markers, correlating better with patient responsiveness to immune checkpoint inhibitors[28]. This suggests that VISTA serves as a regulatory factor influencing the $Cxcl9{:}Spp1$ balance within the TME.

Beyond TAM regulation, VISTA impacts CD8[+] T cells in the pancreatic TME. Our findings demonstrated that TAMs in VISTA-deficient mice produced increased levels of CXCL9, which in turn facilitated the recruitment of CXCR3[+] CD8[+] T cells. These CXCR3[+] T cells have a stem-like phenotype critical for sustaining an effector T cell pool and resisting exhaustion- an issue that limits immune response effectiveness in the TME[52–54]. Macrophage secreting CXCL9/10 are thought to recruit stem-like CD8[+] T cells, which are critical for effective responses to checkpoint inhibitors such as anti-PD-1 and anti-CTLA-4 antibodies[55–57]. The elevated levels of IFN-γ-producing CD8[+] T cells observed in VISTA-deficient mice indicate enhanced immune activation. While these findings indicate that VISTA may influence CD8[+] T cell recruitment and activation via the CXCL9/CXCR3 axis, further studies are required to establish whether this pathway is essential for these observed immune effects.

P-selectin Glycoprotein Ligand-1 (PSGL-1) has been identified as a potential binding partner for VISTA under acidic conditions-common in TME- suggesting that VISTA-PSGL-1 interactions may be significant in regulating immune cell function under these conditions[58,59]. Recent studies have also identified LRIG1, though their specific roles in regulating TAM function and T cell responses in PDAC remain to be explored[60]. Future studies should address how these interactions modulate TAM and T cell activity within the TME[60–62].

Our previous research also demonstrated that VISTA plays a role in preventing IFN-γ release from macrophages in kidney model, reducing fibrosis after injury[63]. This aligns with the current findings where VISTA[+] TAMs limited IFN-γ release from CD8[+] T cells, further supporting VISTA's function in dampening anti-tumor immune responses. Our single-cell analysis of the PDAC TME suggests that modulating the $Cxcl9{:}Spp1$ ratio via VISTA could enhance anti-tumor responses by shifting the balance toward pro-inflammatory immune responses. The precise mechanism by which VISTA drives this shift in macrophages remains to be determined.

ICB therapies targeting PD-1 and CTLA-4 have demonstrated limited efficacy in PDAC, underscoring the need for alternative strategies that address multiple layers of immune suppression. In this context, VISTA inhibition offers a promising therapeutic avenue. Notably, anti-VISTA monotherapy combined with gemcitabine, current standard chemotherapy, elicited a synergistic anti-tumor response, characterized by an increased CXCL9/SPP1 ratio in TAMs and enhanced infiltration of CXCR3[+] CD8[+] T cells. This additive effect is particularly relevant for immune-excluded tumors like PDAC, where conventional ICB therapies have failed to yield substantial clinical benefit. Moreover, considering the well-documented toxicity profile of gemcitabine, our findings suggest that VISTA blockade could enhance therapeutic efficacy while potentially reducing dependence on cytotoxic agents.

## Methods

### In vivo animal studies

All experiments and procedures were performed according to protocols approved by the Institutional Animal Care and Use Committee (IACUC No. 220222-8-11) at Seoul National University animal facilities. C57BL/6J (Stock No. 000664), B6.Rag1em10Lutzy/J (Stock No. 034159), B6.Tg (TcraTrcb)1100Mjb/J (OT-I; Stock No. 003831) mice were purchased from the Jackson Laboratory. B6.$Vsir^{-/-}$ mice were established and bred using a cryopreserved sperm (KOMP Repository, UCD, Davis, CA, USA). Mice were 7–9 weeks old at the start of experiments. Only male mice were used to minimize potential variability associated with sex hormones. All mice were housed in a specific pathogen-free (SPF) barrier facility at Seoul National University under controlled conditions (12-h reverse light/dark cycle, $23 \pm 2\,°C$, and 40–60% humidity). Experimental and control mice were co-housed whenever possible. Mice were euthanized by $CO_2$ inhalation followed by cervical dislocation.

### Cell lines

The Pan02 cell line from National Cancer Institute (NCI, 0509770), and KPC001 cell line isolated from the tumor of a 5- to 6-month-old genetically engineered mouse model of Kras^G12D^p53^R172H/+^Pdx-1-Cre mice were used[64,65]. Pan02-OVA variant was generated by cloning chicken ovalbumin (OVA) into the pCDH-CMV-EF1-RFP-T2A-puro plasmid, followed by transfection into the HEK293T cell line using PromoFectin. Lentiviral particles were harvested and transduced into the Pan02 cell line. The transduced cells were selected by positive expression level of RFP validated by flow cytometry and maintained with puromycin (4 μg/mL). Cell lines were cultured in RPMI-1640 medium (Gibco, Waltham, MA, USA) supplemented with 10% FBS (Gibco) and 10 μM ciprofloxacin. Cells were maintained at 37 °C in a humidified 5% $CO_2$ atmosphere. All cell lines were regularly tested for mycoplasma contamination and were authenticated prior to use.

### Orthotopic tumor models

To establish orthotopic models, 7- to 9-week-old male mice were anaesthetized with rompun®-ketamine mixture and subjected to surgical procedure. After upper left abdominal incision, pancreatic tails were exposed and injected with either $5 \times 10^5$ Pan02 or KPC001 cells resuspended in cold complete RPMI medium mixed at 1:1 dilution with Matrigel (18 mg/mL, Corning Incorporated, Corning, NY, USA) in a final volume of 20 μL. After injection, the pancreas was returned to the abdomen, and the incision was closed using 3-0 Vicryl™ (Ethicon, Inc.,

Somerville, NJ, USA) sutures. Experiments were terminated when tumor reached a size of 1000 mm³, as per the IACUC limit.

### In vivo treatments

To validate the VISTA model, 400 µg per mouse of anti-VISTA antibody (Clone 13F3, *InVivo*Mab, BioXCell, Lebanon, NH, USA) or an Armenian Hamster IgG isotype control (Polyclonal, *InVivo*Mab, BioXCell) was administered via intraperitoneal injection on days 5, 8, and 11 after cancer cell inoculation. For the macrophage depletion model, mice were intraperitoneally injected with 1 mg per mouse of anti-CS1FR antibody (Clone AFS98, *InVivo*Mab, BioXCell) or Rat IgG2α,κ isotype control (Clone 2A3, *InVivo*Mab, BioXCell) 13 days prior to tumor inoculation, followed by three consecutive daily injections of 0.5 mg. For IFN-γ and CXCR3 neutralization, 200 µg of anti-IFN-γ (Clone XMG1.2, *InVivo*Mab, BioXCell) or anti-CXCR3 neutralizing antibody (Clone CXCR3-173, *InVivo*Mab, BioXCell) were administered intraperitoneally 5 days post KPC001 cell inoculation every 3 days until tumor harvest. For combination therapy, 50 mg/kg gemcitabine (Selleckchem, Houston, TX, USA) and/or anti-VISTA antibody (Clone 13F3, *InVivo*Mab, BioXCell, Lebanon, NH, USA) were administered intraperitoneally, starting from day 6 after cancer cell injection. Treatments were given every 3 days.

### Tissue dissociation

Mouse pancreatic tumors were manually minced in small pieces. Tissues were suspended in RPMI-1640 supplemented with DNase I (1 mg/mL, Roche, Basel, Switzerland) and Collagenase IV (1 mg/mL, Roche) and incubated at 37 °C for 20 min, with frequent agitation. The digested cell suspensions were then filtered through a 70-µm nylon cell strainer and treated with 1 × RBC lysis buffer (BioLegend, San Diego, CA, USA) for 15 min on ice and resuspended in the appropriate buffer for cell counting and downstream application.

### Flow cytometry

Single-cell suspensions (1 × 10⁶ cells) from the tumor were incubated with mouse FcγIII/II receptor (CD16/CD32) blocking antibody for 15 min on ice and pelleted by centrifugation. Surface staining was then performed with fluorophore-conjugated primary antibodies for 30 min at 4 °C to detect different populations of tumor-associated leukocytes (gated as CD45⁺) that included TAMs (gated as CD45⁺CD11b⁺Ly6G⁻F4/80⁺), monocytes (CD45⁺CD11b⁺Ly6C⁺), granulocytes/PMN (CD45⁺CD11b⁺Gr-1⁺), dendritic cells (DC, CD45⁺CD11c⁺F4/80⁻), CD4⁺ helper T cells (CD4⁺ Th cells, CD45⁺CD3⁺CD4⁺), CD8⁺ T cells (CD45⁺CD3⁺CD8⁺), and B cells (CD45⁺CD19⁺). iNOS⁺ TAMs and Arg1⁺ TAMs were gated from F4/80⁺CD11b⁺ TAMs; Effector memory T cells within the CD8⁺ T cell population were gated as CD44⁺CD62L⁻; Exhausted CD8⁺ T cell populations were gated as TIM3⁺PD-1⁺; CXCR3⁺CX3CR1⁻ T cells, TCF-1⁺PD-1⁻ T cells were gated on CD3⁺CD8⁺ T cells; CXCR3⁺CX3CR1⁻ CD3⁺ T cells were gated on TCF-1⁻PD-1⁺ T cells. Dead cells were excluded by either FVS780 (BD Horizon) or FVS510 (BD Horizon) staining.

The antibodies against CD3 (BV711, 17A2, BioLegend; 1:500), CD4 (BV605, GK1.5, BioLegend; 1:500), CD4 (FITC, RM45, BD Pharmingen; 1:1000), CD8 (BV650, 53-6.7, BioLegend; 1:500), CD8 (APC, 53-6.7, BioLegend; 1:1000), CD45.2 (PerCPCy5.5, 104, BioLegend; 1:500), I-A/I-E (BV605, M5/11, BD Horizon; 1:500), F4/80 (BV711, T45-2342, BD Horizon; 1:500), CD11b (BV785, M1/70, BioLegend; 1:3000), CD11c (APC-Cy7, N418, BioLegend; 1:1000), Ly6G (APC, 1A8-Ly6g, Invitrogen; 1:500), Ly6C (PE-Cy7, HK1.4, BioLegend; 1:500), Gr-1 (BV510, RB6-8C5, BD Horizon; 1:1000), CD62L (BV605, MEL-14, BioLegend; 1:1000), CD44 (BV510, IM7, BioLegend; 1:500), TIM3 (PE, B8.2C12, BioLegend; 1:500), PD-1 (FITC, 29 F.1A12, BioLegend; 1:100), VISTA (APC, MIH63, BioLegend; 1:200), CX₃CR1 (Pacific Blue, SA011F11, BioLegend; 1:500), CXCR3 (BV421, CXCR3-173, BioLegend; 1:200), CCR2 (BV421, SA203G11, BioLegend; 1:400), CD206 (BV650, C068C2, BioLegend;

1:500), CD206 (BV605, C068C2, BioLegend; 1:500), CD80 (PE, 16-10A1, BioLegend; 1:500), and CD86 (FITC, GL-1, BioLegend; 1:200) were used for surface staining.

For intracellular staining, samples were fixed with 4% paraformaldehyde (PFA) after surface staining, followed by intracellular staining of CXCL9 (PE, MIG-2F5.5, BioLegend; 1:200), OPN (Alexa-Fluor647, LFMb-14, SantaCruz; 1:200), IFN-γ (APC, XMG1.2, BioLegend; 1:500), TNF-α (PE, Mab11, BioLegend; 1:500), Granzyme B (FITC, GB11, BioLegend; 1:500), and Perforin (PE, S16009A, BioLegend; 1:500) in Triton buffer. For intracellular cytokine detection, immune cells were stimulated with Phorbol 12-myristate 13-acetate (PMA, 50 ng/mL, Sigma-Aldrich), ionomycin (1 µg/mL, Sigma-Aldrich), and Golgi-Stop (2 µM, BD Biosciences) for 4 h.

Cells were washed with staining buffer and analyzed by flow cytometry on LSR Fortessa X-20 flow cytometer (BD Biosciences, Franklin Lakes, NJ, USA). Data were analyzed using FlowJo v10 software (BD Biosciences). Gating strategies of flow cytometry analysis are reported in Supplementary Fig. 4.

### Immunohistochemistry (IHC) and Immunofluorescence (IF) staining

Tumor tissues were fixed in 4% PFA, embedded in paraffin, and sectioned to 4 µm of thickness. For IHC, slides were first deparaffinized in xylene and rehydrated through a graded ethanol series. Antigen retrieval was performed by immersing the slides in pH 6 Citrate Antigen Retrieval Solution (Sigma-Aldrich, St. Louis, MO, USA) and heating for 10 min. Endogenous peroxidase activity was blocked by incubating the slides with 10% hydrogen peroxide for 10 min. Slides were then blocked for 1 h at RT in a solution containing 0.1 M Tris, 0.1% Triton ×-100, 10% normal goat/donkey serum, and 1% bovine serum albumin to reduce nonspecific binding.

Primary antibodies, including CD3 (SP162, Abcam, Cambridge, UK; 1:300), CD8 (EPR21916, Abcam; 1:1000), CD11b (EPR1344, Abcam; 1:4000), and F4/80 (EPR26545-166, Abcam; 1:5000), were diluted as per the manufacturer's instructions and incubated with the slides overnight at 4 °C. The slides were then washed with PBS and incubated with biotinylated secondary antibodies for 1 h at RT, followed by detection using a DAB chromogen system. Slides were counterstained with hematoxylin, dehydrated, and mounted for imaging. Images were acquired using a Leica DMi8 microscope (Leica Microsystems, Wetzlar, Germany). IHC was performed to visualize immune cell infiltration and tumor architecture.

For IF staining, the following primary antibodies were used: CD8 (4SM15, Novus Biologicals, Centennial, Co, USA; 1:300), F4/80 (A3-1, Abcam; 5 µg/mL) and CD11b (EPR1344, Abcam; 1:2000). The secondary antibodies included Goat anti-rat IgG H&L AF488 (Thermo Fisher Scientific, Waltham, MA, USA; 1:500), goat anti-rabbit IgG H&L AF647 (Thermo Fisher Scientific; 1:500), goat anti-rat IgG H&L AF488 (Thermo Fisher Scientific; 1:500), and goat anti-rabbit IgG H&L AF647 (Thermo Fisher Scientific; 1:500). Nuclei were counterstained with DAPI and images were captured using a laser scanning confocal NIKON Eclipse Ti microscope (Nikon Instruments Inc., Tokyo, Japan). Contact analysis between macrophages and infiltrated T cells was conducted using the Imaris program (version 9.3.1; Bitplane).

For human tissue microarray, PA1921c (TissueArray, Derwood, MD, USA) was stained with anti-VISTA (D1L2G, Cell Signaling Technology, Danvers, MA, USA) antibody diluted as per manufacturer's instructions and incubated. The image was scanned using the Aperio AT2 digital whole slide image scanner (Leica Biosystems, Wetzlar, Germany), and tumor areas (tumor and stroma) were annotated by pathologist. Tumor cells were annotated for the measurement of staining intensity of the tumor cells, and artifact, pigment, or necrotic area were excluded. Counting of positive cells was performed using an image analyzer, and density was calculated by pathologist (Density = Number of positive cells/Annotated are in square mm).

Opal multiplex IF staining was performed on pancreatic cancer tissue microarrays (PA2082b, PA2081-L64, PA961-L87, and PA483-L97). Sections were stained with antibodies against CD68, VISTA, CXCR3, and CD8α, which were visualized using Opal 620, Opal 690, Opal 520, and Opal 480 fluorophores, respectively.

Human pancreatic ductal adenocarcinoma tissue microarrays were purchased from TissueArray. According to the vendor, all specimens were collected with written informed consent under Institutional Review Board approval and were fully de-identified prior to purchase. The use of de-identified human tissue specimens in this study does not involve interaction with human subjects.

## BMDM cell culture and treatment
Bone marrow-derived macrophages (BMDMs) were generated from femurs and tibias of either WT or *Vsir*$^{-/-}$ mice, filtered, and red blood cells were lysed. Cells were washed and plated at $2 \times 10^6$ cells/mL in non-treated six-well plates containing DMEM (Gibco) supplemented with 10% FBS, 1% Antibiotic/Antimycotic, and 10% L929 supernatant. On day 3, cells were refed with fresh media. At day 6, the differentiated BMDMs were detached, counted, and reseeded to the cell culture plate for subsequent experiments.

To conduct CD8$^+$ T cell proliferation assay, BMDMs were cultured overnight in media containing 100 ng/mL lipopolysaccharide (LPS) (Sigma-Aldrich) and OVA peptide or protein according to the experiment. Maturation was confirmed via flow cytometry using anti-mouse CD11b (PE, M1/70, BioLegend) and anti-mouse F4/80 (BV711, BM8, BioLegend) antibodies.

## Splenic dendritic cell culture
Spleens were harvested from WT or *Vsir*$^{-/-}$ mice, minced into small pieces, and agitated in Hank's Balanced Salt Solution (HBSS) containing Collagenase IV (1 mg/mL, Roche), DNase I (50 µg/mL, Roche), and 1% FBS for 20 min at 37 °C. The enzymatic reaction was stopped by adding 1 mM EDTA for 5 min at RT. The cell suspension was filtered through a 70-µm cell strainer. CD11c$^+$ DCs were enriched using magnetic-activated cell sorting (MACS Microbeads and Separation Unit, Miltenyi Biotec, Bergisch Gladbach, Germany).

## In vitro Macrophage phagocytosis assay
BMDMs from WT or *Vsir*$^{-/-}$ mice cultured in 24-well plates ($2 \times 10^5$/well) were co-incubated with fluorescently stained $1 \times 10^6$ apoptotic thymocytes. Thymocytes were added to the BMDMs at 1:4 ratio. Phagocytosis was allowed to proceed for 2 h at 37 °C. After coculture, thymocytes were washed away extensively and macrophages were detached by trypsinization. Percentage of macrophages engulfing dead cells was determined on a flow cytometer.

## In vitro CD8$^+$ T cell proliferation assay
BMDMs from WT or *Vsir*$^{-/-}$ mice were differentiated and on day 6, BMDMs were treated with different concentrations of OVA (Sigma-Aldrich): 3 µg/mL, 10 µg/mL, 30 µg/mL, 100 µg/mL, 300 µg/mL, and 100 ng/mL LPS (Sigma-Aldrich) were added to the culture. CD8$^+$ T cells were isolated from the lymph nodes of OT-I transgenic mice by negative selection using MACS (Miltenyi Biotec). To test T cell proliferation and cytotoxicity, cultured BMDMs were co-incubated with freshly prepared CD8$^+$ T cells (> 95% viability) at a ratio of 1:4. At 48 h and 72 h post-co-culture, IFN-γ, TNF-α, and perforin-producing CD8$^+$ T cells were analyzed after stimulation with PMA (50 ng/mL) and ionomycin (1 µg/mL) and Golgi-Stop (2 µM) for 4 h. T cell proliferation and cytotoxicity were verified by flow cytometry.

## Real-time qPCR
Total cellular RNA was extracted using Trizol reagent (Thermo Fisher Scientific, Waltham, MA, USA). RNA was reversely transcribed using the PrimeScript RT Reagent Kit (TaKaRa Bio Inc., Shiga, Japan). Gene expression was evaluated by real-time reverse-transcriptase PCR using iQ SYBR Green Supermix (Bio-Rad Laboratories, Hercules, CA, USA) on a PCR amplification and detection instrument (CFX Connect Real-Time PCR Detection System; Bio-Rad). Gene expression was normalized to *Gapdh* and the mean relative gene expression was calculated using the $2^{-\Delta\Delta Ct}$ method. The primers used are listed in Supplementary Table 2.

## Single-cell RNA sequencing
Samples were collected 28 days after cancer cell inoculation and dissociated as described above. Live CD45$^+$ cells were enriched using FACS on an AriaIII sorter (BD Biosciences). ScRNA-seq libraries were generated using the Chromium Single Cell 3′ Reagent Kit v3.1 (10× Genomics, Pleasanton, CA, USA) following the manufacturer's instructions. Briefly, single cells were encapsulated in gel beads in emulsion (GEMs), lysed, and subjected to RNA barcoding, reverse transcription, and PCR amplification. Library quality control was performed with an Agilent 2100 Bioanalyzer (Agilent Technologies, Santa Clara, CA, USA) at the Genomic Medicine Institute Research Service Center, and sequencing was conducted on an Illumina NovaSeq 6000 instrument.

## Single-cell curation and high-dimensional data reduction
FASTQ files were aligned to mouse genome reference (mm10-2020-A) and converted into gene counts with Cell Ranger (v 4.0.0) with default parameters. Gene counts were imported into the R environment (v 4.4.0) and processed with Seurat (v 5.1.0). During Seurat object creation, cells and genes satisfying any of the criteria were removed: (1) genes expressed in fewer than three cells, (2) cells with more than 100,000 RNA counts, fewer than 200 genes, or more than 9000 genes, or (3) cells with a mitochondrial gene expression exceeding 10%.

Gene counts for each cell were normalized using the SCTransfrom (SCT) normalization method and scaled with the "ScaleData" function. The top 3000 highly variable genes were identified using the "FindVariableFeatures" function using the "vst" method in the Seurat R-package. PCA was performed using the "RunPCA" function with default parameters in Seurat R-package.

## Cluster annotation and detection of differential genes between genotypes
A shared nearest neighbor (SNN) graph was computed using the "FindNeighbors" function, a Seurat R-package, based on the first 20 principal components to calculate the distance between cells. Cell clusters were defined using the Louvain algorithm with the "FindCluster" function in Seurat R-package. For visualizing high dimensional components into two dimensions, UMAP was adopted using the "RunUMAP" function in Seurat R-package. Cluster-specific genes were identified using the "FindAllMarkers" and "FindMarker" functions in Seurat R-package with following options, including "pos = TRUE and min.pct = 0.1" and "log fold-change (logfc.threshold) cut-off of <0.25". The 'FindConservedMarkers' function in Seurat R-package was used for each cluster to identify conserved markers, and top features were selected based on a minimum cutoff of 10% of cells with the lowest expression of the gene to ensure adequate cell cluster annotation. We downsampled Mono/Macrophages to 1365 cells, and CD8$^+$ T cells to 966 cells so that the number of cells were identical between WT and *Vsir*$^{-/-}$ genotypes. For all downstream analyses, CD45$^+$ cancer-associated fibroblasts (CAF) were neglected and downsampled files were used to study macrophages and CD8$^+$ T cells.

## Trajectory analysis in macrophages
To assess monocytes/macrophages trajectory, we used the R package Monocle 3[66]. The macrophage subset was randomly downsampled to 1365 cells and imported into Monocle 3. Non-linear dimensionality reduction was performed using UMAP. Cell trajectory was assessed using the simplePPT method, which learns tree-like trajectories and

further reduces dimensionality, implemented by the "learn_graph" function. The resolution for clustering was set to 1e-2. Data was visualized with UMAP embeddings and trajectories derived within Monocle and overlaid with clusters annotated from the Seurat analysis.

### Gene set enrichment analysis

To analyze the potential biological pathways involved in the pathogenesis of pancreatic cancer, we performed gene set enrichment analysis (GSEA, v4.3.2), comparing Macrophage and CD8+ T cell subclusters between WT and *Vsir*−/− samples. Three gene set databases were used: Hallmark gene sets (h.all.v2024.1.Hs.symbols.gmt), Kyoto Encyclopedia of Genes and Genomes (KEGG, c2.cp.kegg_medicus.v2024.1.Hs.symbols.gmt), and the Immunologic Signature gene sets (c7.immunesigdb.v2024.1.Hs.symbols.gmt). The number of permutations was set to 1000, and a *p*-value less than 0.05 was considered statistically significant. Significant pathways differentially expressed between WT and *Vsir*−/− samples were visualized using the ggplot2 based on their normalized enrichment scores (NES) and $-\log_{10}$ (*P*-value).

### Cell-cell interaction

Cell-cell interactions were analyzed using CellChat v.2.1[33] to explore interaction patterns among clusters in pancreatic tumor. Overall interactions between cell type pairs were first calculated, followed by an analysis of communication probabilities between macrophages and CD8+ T cell clusters in WT and *Vsir*−/− mice. A minimum of 10 cells per cluster was required to infer communication. Communication probabilities were computed using the triMean to determine average gene expression per group. Differences in interaction strength and ligand-receptor (L-R) pairs between the two cell populations were subsequently identified.

### Overall survival analysis on the *VSIR* gene signature

TCGA patient data were stratified into "high" (top 25%) and "low" (bottom 25%) VISTA expression groups based on quartile cutoffs. Survival differences between the two groups were analyzed using the Kaplan–Meier estimator and compared with a log-rank test. Hazard ratios (HR) were calculated using Cox proportional hazards regression. Visualization was performed using the R packages "survival" and "survminer" function.

### Logistic regression of dichotomized gene expression profiles in TCGA PAAD

The STAR counts from the TCGA PAAD cohort for the following genes, or the ratio of STAR counts between genes, were dichotomized into LOW and HIGH groups based on their median values: *VSIR, CD8B, ADGRE1, CD68, CCR2, CX3CR1,* and *CXCL9/SPP1*. The associations between the LOW and HIGH groups for *VSIR* and those for other genes were evaluated using multiple logistic regression. Groups based on genes other than *VSIR* were included as independent variables, with the *VSIR* group as the dependent variable, and analyses were adjusted for age and sex. Missing values were handled by listwise deletion. The significance threshold was set at 0.05.

### Processing and analysis of human PDAC single-cell RNA-Seq data

Single-cell RNA-seq data from 25 pancreatic samples in the HTAN WUSTL cohort (pancreatobiliary-type carcinoma, 10× Genomics) were accessed via the HTAN data portal (https://humantumoratlas.org) through Synapse[67] and processed using Seurat R-package. Raw 10× matrices were imported using "Read10X()" and converted into Seurat objects with "CreateSeuratObject()", retaining only *PTPRC*-expressing cells.

Samples were merged using "merge()" and integrated via "JoinLayers()". Mitochondrial content was calculated with "PercentageFeatureSet()" (pattern "^MT-"), and cells were filtered based using strict QC criteria: <50% mitochondrial reads, 100–5000 genes, and 100–20,000 UMIs. Data were normalized using "SCTransform()", with regression on transcripts and gene counts. Dimensionality reduction was performed using PCA (top 30 components), followed by UMAP "RunUMAP()". Clusters were identified using Louvain algorithm with a resolution parameter set to 0.8 for the TIL population and 0.5 for macrophage and CD8+ T cell subsets.

Cluster markers were identified using FindAllMarkers() (expressed in > 25% of cells, log2 FC > 1.5, and adjusted-*P* < 0.05). To stratify patients by immune checkpoint expression, aggregate *VSIR* expression within the TIL compartment was computed. Based on ranked expression values, 21 patients were stratified into $VSIR^{\text{high}}$ (top 10; *n* = 10) and $VSIR^{\text{low}}$ (bottom 10; *n* = 10) groups. Group comparisons were conducted using the one-sided Mann–Whitney *U* test.

### Quantification of gene co-expression and differential expression analysis

To assess gene co-expression patterns at single-cell resolution, two-dimensional kernel density contour plots were generated using the geom_density_2d() function from the ggplot2 R package[68]. Normalized expression values were extracted from Seurat objects using the FetchData() function. To mitigate issues arising from zero expression values that can cause fragmented contours, small random noise (i.e., jitter) was added to each expression value, resulting in smoother and more interpretable density estimates.

Gene pairs were selected based on relevant biological context, and for each gene, the mean expression level across all cells was used as a threshold to define vertical and horizontal cutoff lines, dividing the expression space into four quadrants. Cells were classified as positive if their expression exceeded the gene-specific mean, and the proportion of cells in each quadrant was quantified to evaluate co-expression dynamics across conditions.

### Differential gene expression analysis between clusters and genotypes

For differential expression analysis between selected clusters irrespective of genotype, we used the full set of cells without downsampling to preserve native cell distributions. DEGs were identified using the FindMarkers() function in Seurat R-package, with consistent filtering criteria applied: > 25% expression frequency, fold change > $\log_2(1.5)$, and adjusted *p*-value < 0.05. In a separate analysis focusing on genotype-specific differences, we selected clusters enriched in either wild-type or *Vsir*-deficient conditions. To control for unequal cell numbers across clusters, 300 cells were randomly sampled per cluster prior to DEG analysis. Resulting gene sets were intersected or unioned to identify genotype-specific or shared transcriptional signatures.

### Quantification and statistical analyses

Results were expressed as mean ± standard error of the mean (SEM). Differences between groups were evaluated using appropriate statistical tests. Repeated Measures (RM) ANOVA with Geisser-Greenhouse correction was used for tumor growth curves. One-way ANOVA with Sidak's *post hoc* test was used for immune cell frequency comparisons over time. Pairwise group comparisons were performed using unpaired two-sided Student's *t*-test or Mann–Whitney *U* test, as appropriate. Survival data were analyzed using the log-rank test. For scRNA-seq data, adjusted *p*-values (*p*_adj) for multiple comparisons were calculated using the FindAllMarkers function, with all DEGs meeting the criterion *p*_adj <0.05. For the analysis of human scRNA-seq data, which was conducted to validate findings from murine models, the one-sided Mann–Whitney *U* test was employed.

Statistical analyses were conducted using GraphPad Prism v9.0 (GraphPad Software) or R v3.4.1 (R project). Details regarding

statistical tests, exact *n* values, and their definitions are provided in the figure legends. Exact *P* values are reported in the figures wherever possible.

## Reporting summary

Further information on research design is available in the Nature Portfolio Reporting Summary linked to this article.

## Data availability

The single-cell RNA sequencing data generated in this study have been deposited in NCBI Gene Expression Omnibus (GEO) under the accession code GSE282101. The publicly available TCGA-PAAD dataset used in this study can be accessed via the GDC Data Portal [https://portal.gdc.cancer.gov/projects/TCGA-PAAD]. Source data are provided with this paper.

## Code availability

All custom-written code can be accessed at [https://github.com/gracesshin/VISTA_PDAC].

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

## Acknowledgements

Figures 7a and 7c were created with BioRender.com. This research was supported by a grant of the Korea Health Technology R&D Project through the Korea Health Industry Development Institute (KHIDI), funded by the Ministry of Health & Welfare, Republic of Korea (No. RS-2024-00406325 to D.-S.L.). This research was also supported by the National Research Foundation of Korea (NRF) grant, funded by the Korea government (MSIT) (No. RS-2024-00345658 and RS-2025-25403101 to D.-S.L.) and by the Education and Research Encouragement Fund of Seoul National University Hospital. Suk-Kyung Shin was supported by BK21 Four Biomedical Science Program, Seoul National University College of Medicine.

## Author contributions

S.-K.S. conceived and designed the study, performed experiments, analyzed murine single-cell data, and wrote the manuscript. G.K. performed and led the analysis of murine and human single-cell data and contributed to manuscript revision. G.H.K. guided human tissue staining and performed human IHC scoring. K.J. provided key reagents and methodological support. S.M.P. assisted with experiments. E.-B.S., S.-K.Y., and H.M.S. contributed to data discussion and manuscript revision. H.-R.K. and D.-S.L. supervised the study and edited the manuscript. All authors discussed the results and commented on the manuscript.

## Competing interests

The authors declare no competing interests.
