## [Transparent Peer Review file · Nature Communications]

VISTA Drives Pancreatic Tumor Progression Through Modulation of the Tumor-Associated Macrophage Polarity

Corresponding Author: Professor Dong-Sup Lee

Version 0:

Reviewer comments:

Reviewer #1

(Remarks to the Author)

Shin et al. explore the role of VISTA in shaping the TME of PDAC, a malignancy characterized by poor prognosis and resistance to immunotherapy. Using murine models and corroborated with human datasets, the authors demonstrate that VISTA deficiency shifts TAMs from an immunosuppressive phenotype to a pro-inflammatory state, enhancing CD8+ T cell recruitment and activation. Specifically, it identifies a novel axis where VISTA deficiency shifts TAMs from an immunosuppressive (SPP1-high) phenotype to a pro-inflammatory (CXCL9-high) state, enhancing CD8+ T cell recruitment, activation, and cytotoxicity. This reprogramming reduces tumor growth, improves survival, and is a reshaped TME more conducive to anti-tumor immunity. The findings align with the growing understanding that macrophage-driven reprogramming can synergize with adaptive immune responses to overcome the resistance of PDAC to immune checkpoint blockade therapies. This manuscript highlights the therapeutic potential of targeting VISTA to overcome the immunosuppressive barriers in PDAC.

The study lacks sufficient human experimental validation to establish translational relevance. Furthermore, the study would benefit from a deeper exploration of the molecular pathways regulating VISTA expression and its upstream and downstream effects since others have already established CXCL9-SPP1 (ex. Bill et al. Science 2023). Despite these limitations, the research offers significant contributions to understanding VISTA's role in the PDAC TME and lays a strong foundation for considering VISTA inhibition as a promising therapeutic strategy.

Major Weaknesses:

1. While human scRNA-seq datasets are used, more direct experimental validation in human PDAC samples is required to strengthen translational relevance.
2. The role of upstream regulators of VISTA expression and downstream signaling pathways influencing TAM polarization and mechanism remains insufficiently addressed.
3. The study does not compare the efficacy of VISTA inhibition with standard treatments (ex., gemcitabine plus paclitaxel or FOLFIRINOX) or other immune checkpoint inhibitors, limiting its positioning within the broader therapeutic landscape.

Minor Weaknesses:

1. Functional validation of macrophage-T cell interactions beyond proximity measures (e.g., cytokine secretion profiles) would be beneficial.
2. Sentences in the introduction and discussion sections occasionally overstate claims (e.g., significantly reprogram TAMs).
3. Figure 2: The role of CSF1R inhibition in TAM modulation is compelling but needs additional experimental replication to confirm its rigor.
4. Figure 3: Comparative gene expression heatmaps could better illustrate shifts in Cxcl9:Spp1 ratios.
5. Clarify ambiguities in the discussion, such as "VISTA inhibition could synergize with ICB" (specify conditions).
6. The depletion of F4/80^{high}CD11b^{high} TAMs via CSF1R inhibition in Figure 2 is an important mechanistic result, but the text focuses more on TAM dynamics than on connecting these changes to downstream CD8+ T cell activation (Figure 2g-h).
7. The importance of macrophage cluster2's higher Cxcl9:Spp1 ratio (Figure 3f) is briefly stated in the text but not adequately contextualized regarding CD8+ T cell activation and tumor regression.
8. The results text emphasizes the reduced exhaustion and enhanced effector function of CD8+ T cells in VISTA-deficient tumors (Figure 5), but the exhaustion marker analysis (e.g., module scores) lacks sufficient explanation to clarify the distinction between effector and terminally exhausted phenotypes.

Supplementary figures:

1. S3: Immunofluorescence staining of CD8+ T cells and CD11b+ macrophages is crucial to visualize immune infiltration but is only superficially discussed in the context of Figure 1.
2. Supplementary Figure S5 contains important flow cytometric data on CSF1R-mediated macrophage depletion and VISTA expression, but its relationship to Figures 2 and 3 is not explicitly stated. For instance, the text does not sufficiently highlight how the functional phenotypes of TAMs (e.g., changes in iNOS and Arginase-1 expression in S5a–c) connect to their pro-inflammatory or immunosuppressive states.
3. S7 supports Figure 3 but is insufficiently explained in the results, particularly regarding the functional differences between Spp1+ and Cxcl9+ macrophages.
4. Supplementary Figure S8 discusses the lack of difference in phagocytosis capacity between WT and VISTA-deficient macrophages (S8a), which could provide a contrasting point to emphasize functional gains in antigen presentation.

Reviewer #2

(Remarks to the Author)

This manuscript by Shin et al. investigates the role of the inhibitory immune receptor VISTA in macrophage polarization and T cell differentiation in a mouse model of pancreatic cancer. The authors find a clear tumor growth phenotype when VISTA is genetically or therapeutically targeted, and ultimately use single cell RNA sequencing to associate this effect with transcriptional changes in tumor macrophages associated with suppressive-to-inflammatory macrophage conversion (Spp1: CXCL9 axis) and a shift in CD8 T cell differentiation away from terminal exhaustion. The phenotypes are convincing and the data is novel, relevant, and clearly presented. Key weaknesses to be addressed include its heavy reliance on an established, but genetically irrelevant and immunologically receptive PDAC model, potential concerns related to the number of experimental replicates across in vivo experiments, and additional validation experiments needed to confirm mechanistic insights that are established only through correlation. In summary, the manuscript provides valuable new insights into the role of VISTA in PDAC and has room for improvement through validation experiments that seem feasible and have a high chance of success. Please see my specific comments below:

Major comments:

1. Most in vivo experiments seem to utilize 3-5 mice per group with no statement of replicates, unless I have missed a statement of experiment replication somewhere in the manuscript. Please clarify in methods and/or figure legends how many samples per condition were used, whether any samples were removed from consideration and by what criteria, and how many independent experiments were performed for each experiment.
2. The findings of this paper are highly reliant on a relatively immunogenic and genetically irrelevant PancO2 PDAC model. Numerous, more genetically relevant and poorly immunogenic PDAC models exist at this time; can the key results of this study be replicated in a more rigorous KPC-type model?
3. It remains unclear whether (i) VISTA deficiency primarily affects macrophage polarization leading to secondary effects on T cell differentiation; (ii) VISTA deficiency primarily affects T cell differentiation/exhaustion leading to secondary effects on macrophage polarization; or (iii) these effects occur independently. Can the authors provide any data that indicates whether VISTA deficiency on macrophages or T cells alone is necessary or sufficient for their phenotypes?
4. Can the authors explain why VISTA-KO BMDMs show a functional phenotype as compared to VISTA WT BMDMs in vitro presumably in the absence of VISTA ligand? What is acting as the VISTA ligand in the WT BMDMs?
5. Key points of the proposed mechanism could benefit from in vivo validation. Is the CXCL9/CXCR3 axis required for the robust antitumor phenotype in the VISTA-KO mice? Are macrophages derived from tumors grown in VISTA-KO versus WT mice differentially able to mediate T cell chemotaxis, priming, and/or function? The validation only in BMDMs does not recapitulate the key, complex biology of the TME.

Minor comments:

1. Fig 2B: What is the rationale for your anti-CSF1R regimen starting 2 weeks before tumor implantation? Why not apply the antibody during/after implantation, in context of tumor progression? What macrophages are you depleting with the D-14 anti-CSF1R regimen?
2. I would recommend removing the artificial "gates" on the scRNA UMAP plots; let the clusters speak for themselves. Lines drawn in UMAP space have no inherent meaning
3. Figure 3F: specifically what do you mean by cumulative frequency at the far right? Cluster 2 shows 100% frequency in both genotype conditions? What does this mean? Also, how did you determine the cutoff for CXCL9 positivity versus Spp1 positivity at the gene expression level?
4. Figure 4i: the % positive gate seems to bias the results because it excludes non-proliferating T cells, of which there are more in the WT BMDM group. If you only assess only the proliferating cells in both groups, is there a difference in cytokine production btw groups?
5. Figure 4H: Is there any difference in costimulatory ligand expression (CD80/CD86, etc.) in BMDMs from WT versus VISTA-KO mice? Or is the difference in cross-presentation capacity solely due to higher levels of pMHC?
6. Figure 6: CXCR3 and CX3CR1 expression are likely insufficient to conclusively infer levels of T cell exhaustion. Could you show a difference in expression of canonical markers such as PD1/Tim-3, PD-1/CD39, or PD-1/TOX?

Reviewer #3

(Remarks to the Author)

The article investigates the role of VISTA in PDAC and its regulatory effects on TAM polarization and the immune microenvironment. This topic is highly relevant, given the advancement of immunotherapy. The study's use of single-cell RNA sequencing and the novel discovery of TAM polarity (using the Cxcl9/Spp1 ratio) hold substantial scientific value. VISTA is proposed as a potential immunotherapeutic target, complementary to existing PD-1/PD-L1 or CTLA-4 immune checkpoint inhibitors, underscoring the clear clinical relevance of this research.

The study employs a broad array of modern techniques, including single-cell RNA sequencing, immunohistochemistry, flow cytometry, and in vivo/in vitro functional assays. The methodology is comprehensive and robust, with the syngeneic orthotopic model providing a realistic simulation of the immune microenvironment and strong data support. However, certain areas of the study warrant further refinement to maximize its scientific and translational impact.

1. Deeper Mechanistic Research Needed

- When the study reveals the regulatory effects of VISTA deficiency on TAM polarization, it lacks a deeper exploration of upstream signaling pathways or regulators. Including more mechanistic studies could enhance its scientific impact.
- The mechanisms underlying the enhanced CD8⁺ T cell function (e.g., metabolic states or epigenetic regulation) are underexplored. Please adding more data on these aspects would strengthen this part.

2. Optimizing Data Integration and Presentation

- The article relies heavily on supplementary materials (e.g., many critical findings are presented in supplementary figures). This might hinder reviewers' ability to grasp the study's significance quickly.
- Although the single-cell RNA sequencing analysis is comprehensive, the data could be better integrated to simplify conclusions and improve readability.

3. Strengthening the Discussion on Clinical Relevance

- The discussion on the clinical feasibility of VISTA-targeted therapy is brief. For example, more information on the development and challenges of existing VISTA inhibitors would be valuable. High-impact journals often emphasize translational relevance.

4. Addressing the Lack of Comparative Analysis

- The study does not compare VISTA's function and utility with other immune checkpoint molecules (e.g., PD-1, TIM-3). Such comparisons could better highlight VISTA's uniqueness.

In general, the manuscript requires major revisions to achieve its full potential.

Version 1:

Reviewer comments:

Reviewer #1

(Remarks to the Author)

The authors have significantly improved the manuscript by addressing previous critiques and enhancing both translational and mechanistic depth. The inclusion of human pancreatic ductal adenocarcinoma (PDAC) tissue and single cell RNA sequencing data from the HTAN cohort connects findings from mouse models with human data. This demonstrates a consistent relationship between VSIR expression, the phenotype of tumor-associated macrophages (TAMs), and the functionality of CD8⁺ T cells. The addition of rigorous multivariate analyses, single-cell stratification, and other co-expression profiling provides strong support for the manuscript's central hypothesis: that VISTA orchestrates immunosuppression in PDAC by modulating both the TAM and T-cell compartments.

Finally, expanded the discussion of upstream and downstream VISTA signaling, and strengthened the experimental data by including studies on CSF1R inhibition and combination therapy with gemcitabine. The revised manuscript is comprehensive, data-driven, and well-integrated across both murine and human systems.

Reviewer #2

(Remarks to the Author)

Thank you for your detailed responses to my concerns, which have been sufficiently addressed.

Reviewer #3

(Remarks to the Author)

The revised manuscript by authors investigates the immunomodulatory role of VISTA in pancreatic ductal adenocarcinoma (PDAC), with a particular focus on tumor-associated macrophage (TAM) polarization and the downstream effects on CD8⁺ T cell function. The study addresses an important and timely topic, given the poor responsiveness of PDAC to current immunotherapies and the increasing interest in checkpoint blockade.

Major Concerns

1. Mechanistic Evidence for VISTA Signaling Remains Incomplete:

Although the authors state that phospho-protein array experiments were performed to explore downstream signaling pathways, they have not presented any of these data. As a result, the claim that VISTA regulates TAM polarization through specific pathways remains speculative.

2. Lack of Functional Validation for CXCL9/CXCR3 Axis:

While the manuscript shows increased CXCL9 expression and CXCR3⁺ CD8⁺ T cell infiltration in VISTA-deficient tumors, it

does not directly test whether this axis is necessary for the anti-tumor effects. Blocking experiments would be critical to establish causality. If not feasible, this limitation should be explicitly stated in the Discussion.

3. CD8⁺ T Cell Exhaustion Analysis Requires Further Support:

The conclusions regarding reduced T cell exhaustion rely mainly on module scoring from scRNA-seq. Without additional validation—such as flow cytometric co-expression of exhaustion markers or functional assays. The current interpretation may overstate the data. The language should be moderated to reflect the correlative nature of the evidence.

4. Insufficient Human Tissue Validation:

Although the authors perform database analyses, they have not provided direct experimental data on VISTA expression or macrophage phenotype in human PDAC tissue. Immunohistochemistry or multiplex immunofluorescence staining on a small patient cohort could help validate the translational significance of the findings.

Version 3:

Reviewer comments:

Reviewer #3

(Remarks to the Author)

The revised manuscript incorporates substantial improvements in response to peer review, transforming the study from a phenotypic analysis into a rigorous, mechanistically detailed investigation with clear clinical relevance. Using a multi-tiered experimental approach, the authors establish a comprehensive evidence chain and confirm VISTA as a pivotal regulator of immune evasion in pancreatic ductal adenocarcinoma (PDAC). Thank you for your detailed responses to my concerns, which have been sufficiently addressed.

Responses to Reviewers' Comments

We are deeply grateful for your thoughtful comments and invaluable suggestions regarding our manuscript. Your feedback has provided us with an excellent opportunity to improve the quality and clarity of our work. We have carefully revised the manuscript in accordance with the reviewers' recommendations and your editorial guidance. Below, we respectfully addressed each of the reviewers' comments point-by-point and described the corresponding revisions made to the manuscript. All changes are clearly **marked in red** for your convenience. Please note that, as we have reorganized several figures and added new ones, we made minor adjustments to a few words in the text. These changes were made solely for consistency and clarity, and do not affect the meaning of the manuscript. Thank you once again for your time and consideration.

Reviewer #1's Comments and Responses

Major Weaknesses:

*1. While human scRNA-seq datasets are used, more direct experimental validation **in human PDAC samples** is required to strengthen **translational relevance**.*

(Author's Response/Action)

We sincerely appreciate your suggestion to strengthen the translational relevance of our findings. In response, we performed multi-layered analyses using publicly available human PDAC datasets and primary tumor tissue samples to assess whether key observations from our murine studies extend to human PDAC. Specifically, we sought to determine whether:

- (i) *V SIR* expression increases with tumor progression,
- (ii) CD8⁺ T cell- macrophage-associated gene signatures (*e.g.*, *CD8B*, *ADGRE1*, and *CCR2*) are enriched in pancreatic cancer patients with lower *V SIR* expression in bulk RNA-seq, and
- (iii) patterns of macrophage polarization and T cell exhaustion differ between *V SIR*^{high} and *V SIR*^{low} pancreatic cancer patients, consistent with our observation that VISTA-

deficient mouse tumors exhibit fewer *SPPI*⁺ TAMs and a greater abundance of functional effector-like CD8⁺ T cells.

To address the first question, we analyzed *VSIR* expression in the TCGA-PAAD cohort. After adjusting the effects of age and sex, Spearman rank partial correlation analysis revealed a positive correlation between *VSIR* transcript levels and advancing tumor grade ($\rho = 0.1507$, $p = 0.0041$), and analysis-of-covariance analysis revealed the overall mean difference in *VSIR* transcript levels across tumor grades ($F = 20.1097$, $p < 0.0001$; *post-hoc* tests were used, suggesting a role in PDAC progression (Supplementary Fig. S3c). This trend was mirrored at the protein level: immunohistochemistry of pancreatic tissue microarray demonstrated elevated VISTA expression in grade 2 and grade 3 tumors compared to grade 1 tumors (Supplementary Fig. S3d and e). While these samples lacked co-staining for VISTA and pan-macrophage marker CD68, the observed pattern aligns with prior reports of VISTA⁺CD68⁺ macrophage enrichment in PDAC stroma relative to melanoma (Blando *et al.*, *PNAS*, 2019).

New Supplementary Figure S3

Supplementary Figure S3. High *VSIR* (*c10orf54*) expression level correlates with poor overall survival in PAAD patients. a Kaplan-Meier curve for the *VSIR* signature in The

Cancer Genome Atlas (TCGA) pancreatic adenocarcinoma (PAAD) cohort. Statistical significance was determined by the log-rank (Mantel-Cox) test. **b** *V SIR* expression levels in tumor ($n = 179$) versus normal pancreas ($n = 171$) samples from the TCGA PAAD cohort, transformed as $-\log(\text{TPM}+1)$. **c** *V SIR* expression levels in tumor stratified by tumor grade from the TCGA PAAD cohort. **d** Quantification of $VISTA^+$ cells in pancreatic tumors stratified by tumor grade. **e** Representative immunohistochemical images from a tumor microarray are shown. *n.s.*, not significant.

In **Results** (Page 5, Lines 113-119)

Original:

“This finding was further supported by clinical data from the TCGA dataset, where higher *VISTA* expression correlated with poorer overall survival across multiple cancer types ($n = 33$), including pancreatic adenocarcinoma (Supplementary Fig. S1a). In the PAAD cohort, elevated *VISTA* expression was observed in both basal ($n = 65$) and classical ($n = 85$) subtypes compared to normal tissues (Supplementary Fig. S1b). These clinical correlations underscore the potential of targeting *VISTA* as a therapeutic strategy for PDAC.”

Revised:

→ “In human PDAC, lower *V SIR* expression was similarly associated with improved patient survival by Kaplan-Meier analysis of The Cancer Genome Atlas Pancreatic Adenocarcinoma (TCGA-PAAD) cohort (Supplementary Fig. S3a). *V SIR* expression was significantly elevated in pancreatic cancer tissues ($n = 179$) compared to non-malignant pancreatic tissues ($n = 171$) (Supplementary Fig. S3b). Interestingly, *V SIR* expression increased with advancing histological grade (Supplementary Fig. S3c), a trend also confirmed at the protein level by immunohistochemistry using a PDAC tissue microarray (Supplementary Fig. S3d and e). Together, these clinical observations highlight the potential relevance of *VISTA* as a therapeutic target in PDAC.”

Second, we stratified TCGA-PAAD patients into $V SIR^{\text{high}}$ and $V SIR^{\text{low}}$ groups. Tumors in the *V SIR* low group exhibited relatively higher expression of *CD8B*, *ADGRE1*, and *CCR2*

consistent with a more immune-infiltrated phenotype similar to that observed in VISTA KO mouse tumors. Although associations with *CXCL9* and *SPPI* did not reach statistical significance, they showed directionally consistent trends, supporting the hypothesis that *V SIR* expression may influence myeloid chemokine signaling.

New Supplementary Table S1

Supplementary Table S1. Multivariate Logistic Regression Analysis of Immune-Related Gene Associations with *V SIR* Expression (High versus Low), Adjusted for Age and Sex

	Odds ratio	95% Confidence Interval		P-value
		Lower bound	Upper bound	
CD8B	0.0803	0.0343	0.1878	< 0.0001
ADGRE1	0.1254	0.0567	0.2773	< 0.0001
CD68	0.4612	0.2240	0.9495	0.0340
CCR2	0.0610	0.0257	0.1447	< 0.0001
CX3CR1	0.2622	0.1265	0.5433	0.0003
CXCL9/SPPI	0.3893	0.1892	0.8009	0.0097

In **Results** (Page 6, Lines 134-138)

Original:

“Beyond macrophage reprogramming, *Vsir*^{-/-} tumor exhibited a significant increase in CD8⁺ T cell infiltration, with a shift towards an effector or effector memory (CD44⁺CD62L⁻) phenotype (Fig. 1i). This was accompanied by elevated IFN- γ levels and increased frequency of TNF- α ⁺IFN- γ ⁺ and TNF- α ⁻IFN- γ ⁺ CD8⁺ T cells (Supplementary Fig. S4b), indicating enhanced immune activation and cytotoxic potential.

Taken together, our findings suggest that VISTA serves as a key regulator of immune evasion within the PDAC microenvironment, and targeting VISTA represents a promising strategy to overcome the immunosuppressive barriers in PDAC and enhance anti-tumor immunity.”

Revised:

→ “... TNF- α ⁺IFN- γ ⁺ and TNF- α ⁻IFN- γ ⁺ CD8⁺ T cells (Supplementary Fig. S6b), indicating enhanced immune activation and cytotoxic potential.

To explore clinical relevance of these findings, we performed multivariate logistic regression on the TCGA-PAAD dataset. Elevated *CD8B* expression was associated with 71.6% lower odds of high *VSIR* expression (Supplementary Table S1). Similarly, high expression of *ADGRE1* and *CCR2*—markers of myeloid lineage—were associated with 60.66% and 81.70% reduced odds of elevated *VSIR* expression, respectively. These patterns recapitulate those observed in murine models.

Collectively, these findings identify VISTA as a central regulator of immune suppression in PDAC, orchestrating TAM-mediated immunosuppression and dampening CD8⁺ T cell responses. Targeting VISTA represents a promising immunotherapeutic strategy to overcome the immunosuppressive TME and enhance anti-tumor immunity in PDAC.”

Third, we analyzed a publicly available single-cell RNA-seq dataset from the HTAN WUSTL cohort to examine immune cell composition and gene expression at higher resolution. Patients were stratified into *VSIR*^{high} and *VSIR*^{low} groups based on aggregate *VSIR* expression in tumor-infiltrating lymphocytes (TILs). In the myeloid compartment, *VSIR*^{high} tumors were enriched for anti-inflammatory, TAM-like subclusters (e.g., *CCL2*⁺*MAF*⁺ and *CSTB*⁺*SPP1*⁺), whereas *VSIR*^{low} tumors contained a greater proportion of monocyte-like macrophages expressing *S100A8/9*. In the CD8⁺ T-cell compartment, *VSIR*^{high} tumors were dominated by pre-effector (*GZMK*⁺*GPR183*⁺) and cycling-exhausted (*TOX*⁺*MKI67*⁺) subclusters with reduced expression of cytotoxic effector molecules. Conversely, *VSIR*^{low} tumors exhibited increased frequencies of effector-like CD8⁺ T cells, including *CCL4*^{high} subsets, consistent with the CD44⁺CD62L⁻ effector phenotypes seen in VISTA KO mice.

New Supplementary Figure S10

Supplementary Figure S10. Validation of *VSIR* expression in human scRNA-seq datasets.

a Schematic workflow of human scRNA-seq analysis from the HTAN WUSTL cohort of pancreatobiliary-type carcinoma patients. **b** UMAP visualization of macrophage subsets following subclustering. **c** Dot plot showing gene expression patterns across macrophage subsets. **d** Proportion of pro-inflammatory-like and anti-inflammatory-like genes in macrophages from *VSIR*^{high} ($n = 10$) versus *VSIR*^{low} ($n = 10$) patients. Patients were stratified by immune checkpoint expression: aggregate *VSIR* expression within the TIL compartment was calculated for each patient, and the top and bottom 10 ranked patients were assigned to

V SIR^{high} and *V SIR*^{low} groups, respectively. Group comparisons were performed using the one-sided Mann-Whitney *U* test.

In **Results** (Pages 10-11 , Lines 237-246)

Original:

“Our findings align with human scRNA-seq data on SPP1⁺ and CXCL9⁺ TAMs reported by Bill *et al.*²⁸, with similarities noted between MΦ_clusters 1 and 2. Gene set enrichment analysis (GSEA) showed that *Cxcl9*⁺ TAMs were consistently enriched in MΦ_cluster2 compared to MΦ_cluster1, whereas *Spp1*⁺ TAMs were predominantly in MΦ_cluster1 (Supplementary Fig. S7d, top). Kyoto Encyclopedia of Genes and Genomes (KEGG) pathway analysis further demonstrated that MΦ_cluster2 was enriched in cytokine-cytokine receptor interactions and antigen processing/presentation pathways (Supplementary Fig. S7d, bottom).

These findings suggest that targeting VISTA may modulate the immune microenvironment by shifting TAMs from an immunosuppressive to an immunostimulatory phenotype, thereby enhancing anti-tumor immunity.”

Revised:

→ “These results aligned with human PDAC scRNA-seq data from Bill *et al.*²⁸, showing similarity between MΦ_cluster1 and immunosuppressive *Spp1*⁺ TAMs, and between MΦ_cluster2 and *Cxcl9*⁺ TAMs. Gene set enrichment analysis (GSEA) confirmed that *Spp1*⁺ TAMs were enriched in MΦ_cluster1 (NES = 1.84, *P* < 0.001), whereas *Cxcl9*⁺ TAMs were significantly enriched in MΦ_cluster2 (NES = -2.51, *P* < 0.001) (Supplementary Fig. S9e, top). KEGG pathway analysis further showed that MΦ_cluster2 was enriched for cytokine–cytokine receptor interaction and antigen processing and presentation pathways (Supplementary Fig. S9e, bottom), supporting these TAMs may promote CD8⁺ T cell activation and function.

To validate these findings in human samples, we analyzed the HTAN WUSTL cohort. *V SIR*^{high} tumors were enriched for anti-inflammatory TAMs (hMΦ.0; *CCL2*⁺*MAF*⁺ and hMΦ.4; *CSTB*⁺*SPP1*⁺), while *V SIR*^{low} tumors were dominated by pro-inflammatory monocyte-like TAMs (hMΦ.1; *S100A8/A9*) (Supplementary Fig. S10a–d). These findings suggest that VISTA

shapes the macrophage landscape to favor immune suppression, and that its inhibition reprograms TAMs toward a phenotype that promotes anti-tumor immunity.”

New Supplementary Figure S17

Supplementary Figure S17. Validation of VSIR expression in human scRNA-seq data. a UMAP visualization of CD8⁺ T-cell subsets following subclustering. **b** Dot plot showing gene expression profiles across identified CD8⁺ T-cell subsets. **c** Proportion of effector-like genes in

V SIR^{high} versus *V SIR*^{low} patients. Patient groups were stratified using the same criteria as in Supplementary Fig. S10: aggregate *V SIR* expression within the TIL compartment was calculated for each patient from the HTAN WUSTL cohort, with the top and bottom 10 patients assigned to *V SIR*^{high} versus *V SIR*^{low} groups, respectively. Statistical significance was assessed using the one-sided Mann-Whitney *U* test.

In **Results** (Pages 14-15, Lines 336-342)

Original:

“Flow cytometric analysis further characterized the altered state of CD8⁺ T cells in *Vsir*^{-/-} tumor. *Vsir*^{-/-} tumor had significantly increased expression of CXCR3, with 28.3% of CD8⁺ T cells co-expressing CXCR3 and CX3CR1 compared to 8.07% in WT tumor (Fig. 5h). Additionally, *Vsir*^{-/-} tumor had a higher proportion of TCF-1⁺PD-1⁺ CD8⁺ T cells and fewer CXCR3⁺CX3CR1⁻ CD8⁺ Tex^{term} cells compared to WT control (Fig. 5i, j). These findings suggest that VISTA deficiency promotes a CD8⁺ T cell phenotype with reduced exhaustion and improved effector function within the TME.”

Revised:

→ “...improved effector function within the TME.

Finally, we validated these observations in the HTAN WUSTL cohort. The *V SIR*^{low} group exhibited increased frequencies of effector (hCD8.1; CCL4^{high}) and effector-like (hCD8.2; KCNQ1OT1⁺) CD8⁺ T cells, while the *V SIR*^{high} group was enriched for pre-effector (hCD8.0; GZMK⁺GPR183⁺) and exhausted (hCD8.7; CXCL13^{high}) CD8⁺ T-cell subsets (Supplementary Fig. S10a and Fig. S17a–c). Together, these results confirm that low VISTA expression is associated with enhanced CD8⁺ T cell effector function and reduced exhaustion in human pancreatic tumors.”

In **Methods**

→ “**Logistic Regression of Dichotomized Gene Expression Profiles in TCGA PAAD**

The STAR counts from the TCGA PAAD cohort for the following genes, or the ratio of STAR counts between genes, were dichotomized into LOW and HIGH groups based on their median

values: *VSIR*, *CD8B*, *ADGRE1*, *CD68*, *CCR2*, *CX3CR1*, and *CXCL9/SPPI*. The associations between the LOW and HIGH groups for *VSIR* and those for other genes were evaluated using multiple logistic regression. Groups based on genes other than *VSIR* were included as independent variables, with the *VSIR* group as the dependent variable, and analyses were adjusted for age and sex. Missing values were handled by listwise deletion. The significance threshold was set at 0.05.

Processing and analysis of human PDAC single-cell RNA-Seq data

Single-cell RNA-seq data from 25 pancreatic samples in the HTAN WUSTL cohort (pancreatobiliary-type carcinoma, 10x Genomics) were accessed via the HTAN data portal (<https://humantumoratlas.org>) through Synapse⁶⁹ and processed using Seurat R-package. Raw 10X matrices were imported using ‘Read10X()’ and converted into Seurat objects with ‘CreateSeuratObject()’, retaining only *PTPRC*-expressing cells.

Samples were merged using ‘merge()’ and integrated via ‘JoinLayers()’. Mitochondrial content was calculated with ‘PercentageFeatureSet()’ (pattern ‘^MT-’), and cells were filtered based using strict QC criteria: < 50% mitochondrial reads, 100–5,000 genes, and 100–20,000 UMIs. Data were normalized using ‘SCTransform()’, with regression on transcripts and gene counts. Dimensionality reduction was performed using PCA (top 30 components), followed by UMAP ‘RunUMAP()’. Clusters were identified using Louvain algorithm with a resolution parameter set to 0.8 for the TIL population and 0.5 for macrophage and CD8⁺ T-cell subsets.

Cluster markers were identified using FindAllMarkers() (expressed in > 25% of cells, log₂ FC > 1.5 and adjusted-*P* < 0.05). To stratify patients by immune checkpoint expression, aggregate *VSIR* expression within the TIL compartment was computed. Based on ranked expression values, 21 patients were stratified into *VSIR*^{high} (top 10; *n* = 10) and *VSIR*^{low} (bottom 10; *n* = 10) groups. Group comparisons were conducted using the one-sided Mann–Whitney *U* test.

Quantification of gene co-expression and differential expression analysis

To assess gene co-expression patterns at single-cell resolution, two-dimensional kernel density contour plots were generated using the geom_density_2d() function from the ggplot2

R package ⁷⁰. Normalized expression values were extracted from Seurat objects using the `FetchData()` function. To mitigate issues arising from zero expression values that can cause fragmented contours, small random noise (*i.e.* jitter) was added to each expression value, resulting in smoother and more interpretable density estimates.

Gene pairs were selected based on relevant biological context, and for each gene, the mean expression level across all cells was used as a threshold to define vertical and horizontal cutoff lines, dividing the expression space into four quadrants. Cells were classified as positive if their expression exceeded the gene-specific mean, and the proportion of cells in each quadrant was quantified to evaluate co-expression dynamics across conditions.

Differential gene expression analysis between clusters and genotypes

For differential expression analysis between selected clusters irrespective of genotype, we used the full set of cells without downsampling to preserve native cell distributions. DEGs were identified using the `FindMarkers()` function in Seurat R-package, with consistent filtering criteria applied: > 25% expression frequency, fold change > $\log_2(1.5)$, and adjusted p -value < 0.05. In a separate analysis focusing on genotype-specific differences, we selected clusters enriched in either wild-type or *Vsir*-deficient conditions. To control for unequal cell numbers across clusters, 300 cells were randomly sampled per cluster prior to DEG analysis. Resulting gene sets were intersected or unioned to identify genotype-specific or shared transcriptional signatures.”

While we recognize that human and mouse datasets differ in resolution, sampling depth, and immunological context, these cross-platforms and cross-species findings are directionally aligned and conceptually supportive of our model. Collectively, they suggest that *V SIR* expression is associated with an immunosuppressive landscape in human PDAC, and may represent a clinically relevant axis for immune stratification and therapeutic targeting.

2. The role of upstream regulators of VISTA expression and downstream signaling pathways influencing TAM polarization and mechanism remains insufficiently addressed.

(Author's Response/Action)

We thank you for this valuable suggestion. In light of your comment, we have (1) expanded the *Discussion* to incorporate relevant mechanistic insights to better contextualize our findings, and (2) conducted a phospho-protein kinase array analysis to explore signaling molecules potentially involved in the VISTA pathway.

First, upstream regulation of VISTA (*VSIR*) expression has been linked to various immunological and microenvironmental cues. Hypoxia-induced signaling through HIF-1 α has been shown to enhance VISTA expression in tumor-infiltrating myeloid cells, consistent with the role of metabolic stress in shaping the immune landscape (Deng *et al.*, *Cancer Immunol Res*, 2020). Although type I and II interferons are known to induce broad immune checkpoint programs, direct transcriptional regulation of VISTA by interferons remains incompletely characterized and requires further investigation.

Downstream of VISTA, several studies have described immunoregulatory functions that influence macrophage behavior. Zhang *et al.* (*Cell Rep*, 2024) reported that VISTA deficiency reduces phosphorylation of ERK and STAT3 in tumor-associated myeloid-derived suppressor cells, suggesting that VISTA positively regulates ERK/STAT3 signaling. Given that STAT3 is a known transcriptional activator of SPP1 (Yu *et al.*, *Dev Cell*, 2025), these findings support our findings in which VISTA promotes SPP1 expression. Whether this is through STAT3-dependent signaling will be necessary to delineate the full signaling cascade.

Second, our own experimental data support additional downstream signaling effects of VISTA in macrophages during early activation. Using LPS and IFN- γ -stimulated BMDMs, we found that VISTA deficiency leads to increased phosphorylation of GSK-3 α/β at Ser21/Ser9—an established inactivating modification. We also observed decreased phosphorylation of p53 (S15/S45) in VISTA KO BMDMs. As p53 activity has been associated with M2 polarization (Li *et al.*, *Cell Death Differ*, 2014), this result further supports a shift away from an immunosuppressive phenotype. Additionally, phosphorylation of STAT3 at S727, which

contributes to M2-like macrophage function (Zhang *et al.*, *Front Immunol*, 2023), was similarly reduced in VISTA KO BMDMs.

While further studies will be required to determine whether this regulation is direct or mediated via intermediate signaling nodes, these findings offer a mechanistic basis for the observed immunophenotypic shifts.

These findings are now contextualized in the revised *Discussion*. We have opted not to include the phospho-proteome array data in the main manuscript, as this analysis remains ongoing. We appreciate your suggestion, which has helped refine and deepen the mechanistic framework of our study. Based on our previous experience studying VISTA in kidney disease model, we plan to pursue these signaling pathways in future mechanistic investigations.

In **Discussion** (Page 18, Line 412)

Original:

“Our study highlights VISTA, a B7 checkpoint molecule, as a pivotal regulator of TAM polarization and the immune landscape in PDAC ^{46, 47}.”

Revised:

→ “Our study highlights VISTA as a modulator of macrophage polarization and immune dynamics in PDAC, **although further studies are needed to define its mechanistic role** ^{46, 47}.”

3. The study does not compare the efficacy of VISTA inhibition with standard treatments (ex., gemcitabine plus paclitaxel or FOLFIRINOX) or other immune checkpoint inhibitors, limiting its positioning within the broader therapeutic landscape.

(Author’s Response/Action)

We appreciate this important comment regarding the need to compare VISTA inhibition with standard PDAC therapies. In response, and in alignment with your concerns regarding model relevance, we selected the KPC001 syngeneic PDAC model for combination therapy studies. While our prior mechanistic work was performed in the more immunogenic Pan02 model, which robustly highlights differences between WT and *Vsir* KO tumors, the KPC001 model was chosen for its closer resemblance to the immune-excluded microenvironment of human PDAC.

Accordingly, we conducted *in vivo* experiments using the KPC001 model, comparing gemcitabine monotherapy, anti-VISTA monotherapy, and their combination (Fig. 7a). Both monotherapies moderately reduced tumor growth, while the combination therapy resulted in a significantly enhanced antitumor effect, suggesting additive or synergistic efficacy (Fig. 7b).

Mechanistically, anti-VISTA treatment did not lead to an overall increase in macrophage percentage in the KPC001 model, in contrast to findings in the Pan02 model, where VISTA deficiency increased macrophage abundance (Supplementary Fig. S18b). However, we observed a selective increase in iNOS⁺ macrophages, with only a modest change in Arg1⁺

macrophages, indicating a phenotypic shift toward inflammatory macrophages despite a stable overall frequency (Supplementary Fig. S18c). These results suggest that anti-VISTA modulates macrophage polarization rather than abundance in this PDAC context. The mechanism of action in the combination group appears to differ and remains to be elucidated.

We also observed an increased CXCL9/SPP1 ratio in the anti-VISTA monotherapy group, consistent with a shift toward a proinflammatory chemokine profile. Although this ratio was only modestly elevated in the combination group, the trend supports the notion that VISTA blockade alone can induce IFN-associated chemokine expression. The more limited effect observed in the combination group may reflect the immunosuppressive influence of gemcitabine, underscoring the importance of dosing and timing in combination strategies.

In addition, anti-VISTA monotherapy led to increased infiltration of CD8⁺ T cells. In the combination group, CD8⁺ T cell levels remained relatively stable, likely reflecting counterbalancing effects between gemcitabine-induced depletion and VISTA blockade-mediated recruitment. This finding further supports the need for optimized scheduling and dosing in combinatorial regimens, which we plan to evaluate in future studies.

Beyond CD8⁺ T cell dynamics, we found that both anti-VISTA monotherapy and the combination regimen increased infiltration of CXCR3⁺ immune cells while reducing CX₃CR1⁺ populations relative to control and gemcitabine-treated groups. This increase in CXCR3⁺ cells mirrors observations in the Pan02 tumor model following VISTA deficiency, suggesting a conserved immunologic response across PDAC models. These shifts are consistent with enhanced effector T cell recruitment and attenuation of immunosuppressive myeloid subsets, respectively, reinforcing the concept that VISTA inhibition reprograms the tumor immune microenvironment in a manner complementary to cytotoxic chemotherapy. We hope these results, now presented in the revised manuscript, enhance the positioning of VISTA-targeted therapy within the current PDAC treatment landscape.

New Figure 7

Figure 7. Effect of anti-VISTA antibody in combination with gemcitabine therapy. a Schematic of the treatment regimen for combination therapy with anti-VISTA antibody and gemcitabine. **b** Tumor volumes of mice treated with isotype control, anti-VISTA antibody, gemcitabine, or combination therapy ($n = 10$ per group). Data are presented as mean \pm SEM. Statistical significance was determined using multiple t -tests, with p -values corrected for multiple comparisons using the Sidak method. $n.s.$, not significant. Data are representative of at least three independent experiments. **c** Schematic summary of key ligand-receptor interactions and immune cell communication pathways in WT *versus* VISTA-deficient tumors, highlighting the enhanced crosstalk in the absence of VISTA.

New Supplementary Figure S18

Supplementary Figure S18. Immune cell dynamics following combination therapy. **a** Flow cytometric quantification of F4/80⁺CD11b⁺ TAMs (left) and CD8⁺ T cells (right) at days 10 and 15 post-treatment ($n = 5$), expressed as a percentage of CD45⁺ cells **b** Flow cytometric analysis of iNOS⁺ Arg-1⁻ (left) and iNOS⁻ Arg-1⁺ (right) TAM subsets in tumor, shown as a percentage of CD45⁺ cells ($n = 10$). **c** Representative flow cytometry plots (left) and quantification bar graphs (right) of CXCL9⁺/SPP1⁺ TAMs in tumor, expressed as a percentage of CD45⁺ cells ($n = 7$). **d** Representative flow cytometry plots (left) and quantification bar graphs (right) of CXCR3⁺CX₃CR1⁻ CD8⁺ T cells and CXCR3⁻CX₃CR1⁺ CD8⁺ T-cell subsets in tumor, shown as a percentage of CD45⁺ cells ($n = 5$). Data are mean \pm SEM. Statistical analysis for panels **a–d** was performed using one-way ANOVA with Sidak's multiple comparisons *post hoc* test. *n.s.*, not significant. Data represent at least three independent experiments.

In **Results** (Page 17, Lines 400-402)

Original:

“These findings reveal how VISTA deficiency reprograms the TME from an immunosuppressive to an immune-stimulatory state, highlighting the potential for VISTA as a therapeutic target to boost immune responses against pancreatic cancer.”

Revised:

→ **“VISTA Blockade Synergizes with Gemcitabine to Remodel the Pancreatic Tumor Immune Landscape**

To determine whether these VISTA-dependent signaling network we identified is therapeutically exploitable, we turned to the KPC orthotopic model and compared anti-VISTA monotherapy, gemcitabine, and their combination (Fig. 7a). Both monotherapies produced a modest but significant reduction in tumor burden; however, the combination elicited a markedly greater suppression, indicating at least an additive—and potentially synergistic—anti-tumor effect (Fig. 7b).

Immune profiling revealed that anti-VISTA monotherapy alone increased total TAMs and increased CD8⁺ T cells (Supplementary Fig. S18a). Importantly, the antibody skewed TAMs toward an iNOS⁺ phenotype while producing only minor changes in Arg1⁺ TAMs (Supplementary Fig. S18b). Consistent with our CellChat predictions, CXCL9⁺/SPP1⁺ TAMs were enriched in the anti-VISTA treatment group (Supplementary Fig. S18c), both anti-VISTA and the combination regimen boosted infiltration of CXCR3⁺ CD8⁺ T cells (Supplementary Fig. S18d)—hallmarks of the CXCL9–CXCR3 chemotactic axis we mapped in *Vsir*^{-/-} tumors. These findings validate our mechanistic model and demonstrate that pharmacologic VISTA blockade can re-program the myeloid–T-cell circuit even in the presence of cytotoxic chemotherapy.

Taken together, our data position VISTA as a central architect of immunosuppressive niches in pancreatic tumors. In WT tumors, TAMs co-express *Cxcl9* and *Spp1*: engagement of SPP1 with integrins *Itga4/Itgb1* or *Itgav/Itgb1* on CD8⁺ T cells likely dampens T cell recruitment and activation. By contrast, VISTA deficiency—or therapeutic blockade—tilts this balance toward CXCL9 dominance, amplifying the CXCL9–CXCR3 axis and fostering a

pro-inflammatory TME that supports robust CD8⁺ T cell activity (Fig. 7c). Thus, combining VISTA inhibition with standard chemotherapy offers a rational strategy to overcome immune suppression and improve anti-tumor effect.”

In **Methods** (Page 21, Line 477)

Original:

“The Pan02 cell line, derived from a methylcholanthrene-induced PDAC, was a generous gift from Dr. Keehoon Jung (Seoul National University) ⁶⁵.”

Revised:

→ “The Pan02 cell line from National Cancer Institute (NCI, 0509770), and KPC001 cell line isolated from the tumor of a 5– to 6–month-old genetically engineered mouse model of Kras^{G12D}p53^{R172H/+}Pdx-1-Cre mice were used ^{66, 67}.”

In **Methods** (Pages 21-22, Lines 489-490)

Original:

“After upper left abdominal incision, pancreatic tails were exposed and injected with 5×10^5 Pan02 cells resuspended in cold...”

Revised:

→ “After upper left abdominal incision, pancreatic tails were exposed and injected with either 5×10^5 Pan02 or KPC001 cells...”

In **Methods** (Page 22, Line 503-504)

Original:

“or Rat IgG2 α , κ isotype control (Clone 2A3, *InVivo*Mab, BioXCell) 13 days prior to tumor inoculation, followed by three consecutive daily injections of 0.5 mg.”

Revised:

→ “...0.5 mg. For combination therapy, 50 mg/kg gemcitabine (Selleckchem, Houston, TX, USA) and/or anti-VISTA antibody (Clone 13F3, *InVivoMab*, BioXCell, Lebanon, NH, USA) were administered intraperitoneally, starting from day 6 after cancer cell injection. Treatments were given every 3 days.”

Minor Weaknesses:

1. Functional validation of macrophage-T cell interactions beyond proximity measures (e.g., cytokine secretion profiles) would be beneficial.

(Author’s Response/Action)

We thank you for this valuable suggestion. We fully agree that proximity-based spatial analyses benefit from complementary functional validation. In our original submission, we addressed this by performing a co-culture assay of BMDMs and CD8⁺ T cells. As shown in Figure 4, VISTA KO macrophages significantly enhanced CD8⁺ T cell secretion of IFN- γ , granzyme B and perforin, indicating increased effector function. This experiment was designed precisely to validate the biological relevance of TAM-T cell proximity observed in the tumor microenvironment. We have now revised the *Results* to better highlight this connection by moving the proximity measure result after cytokine secretion data (Fig. 4f).

In Figure 4 (Page 44)

Original:

Revised:

Figure 4. *Vsir* deficiency enhances antigen presentation and stimulates CD8⁺ T cell responses. **a** Differential gene expression analysis of macrophage subsets. Volcano plot comparing MΦ_clusters 2 and 3 with MΦ_cluster1. **b** Schematic of the MHC class I antigen processing and presentation pathway. **c** KEGG pathway analysis of MΦ_cluster2 (light green), MΦ_cluster3 (green), and combined MΦ_clusters 2 and 3 (brown), showing pathways satisfying *p*_{adj} < 0.05. **d** Flow cytometric analysis of SIINFEKL-H2-K^b complex on WT and *Vsir*^{-/-} BMDMs stimulated with OVA protein at varying concentrations. MFI

quantification shown on the right. **e** Proliferation of OT-I CD8⁺ T cells co-cultured with WT or *Vsir*^{-/-} BMDMs loaded with OVA protein (top left) or SIINFEKL peptide (bottom right), shown as the percentage of divided CD8⁺ T cells. **f** Flow cytometric analysis of IFN- γ , TNF- α , and perforin (PFN) production in OT-I CD8⁺ T cells co-cultured with WT or *Vsir*^{-/-} BMDMs loaded with OVA protein. **g** Quantification of CD8⁺ T cells in proximity to macrophages across varying distances (μm). **h** Tumor growth curves following co-injection of BMDMs and CD3⁺ T cells ($n = 6$ per group). **i** Tumor growth curves following co-injection Pan02 cells with OT-I CD8⁺ T cells and WT or *Vsir*^{-/-} BMDMs ($n = 5$ per group). Data are presented as mean \pm SEM. Unpaired two-tailed Student's *t*-tests were applied for panels **d–f**. Multiple *t*-tests with Holm-Sidak correction were used for **h** and **i**. *n.s.*, not significant. All experiments were independently repeated at least twice; representative results are shown.

In **Results** (Page 13, Lines 295-299)

Original:

“...suggesting that VISTA uniquely modulates antigen cross-presentation in macrophages, even though DCs are typically more proficient in this function.

Collectively, our results indicate that VISTA deficiency enhances the functional capacity of TAMs to process and present antigens, leading to increased CD8⁺ T cell activation and a more potent anti-tumor immune response.”

Revised:

→ “Spatial analysis revealed CD8⁺ T cells were closer proximity to TAMs in *Vsir*^{-/-} tumors (median distance: 3–4 μm) compared to WT tumors (> 10 μm), suggesting enhanced macrophage–T cell crosstalk (Fig. 4g). This was further supported by increased CD3⁺ T cell–CD11b⁺ myeloid interactions in *Vsir*^{-/-} tumor cores.”

In addition, mouse cytokine array analysis of IFN- γ and LPS-stimulated BMDMs revealed elevated levels of Serpin E1 (Serp1), leptin, DKK-1, GAS6, C1q, TNF- α , Pref1, IL-27 p28, and IL-28A/B in VISTA KO BMDMs compared to WT BMDMs. The increase in Serpin E1

(Serpin 1) suggests enhanced M1 polarization and IFN- γ -mediated stress responses in VISTA-deficient macrophages. We anticipate conducting further studies in the future to investigate the functionality of VISTA; however, this will not be address in the present manuscript. Accordingly, the data below is not included in this submission.

2. Sentences in the introduction and discussion sections occasionally overstate claims (e.g., significantly reprogram TAMs).

(Author’s Response/Action)

We appreciate this observation and have carefully revised the *Introduction*, *Discussion*, and *Abstract* sections to ensure that all claims are appropriately moderated and substantiated by the data presented. In particular, potentially overstated phrases such as “significantly reprogram TAMs” have been revised to more evidence-aligned alternatives. Below, we provide edits made in the manuscript:

In **Abstract** (Page 2, Lines 38-43)

Original:

“Mechanistically, VISTA loss reprograms tumor-associated macrophages (TAMs) from an immunosuppressive, secreted phosphoprotein 1 (SPP1) expressing phenotype to a pro-inflammatory, C-X-C motif chemokine ligand 9 (CXCL9) expressing subtype. This switch enhances the recruitment of C-X-C motif chemokine receptor 3 (CXCR3) positive CD8⁺ T cells, reducing their exhaustion and sustaining their cytotoxic activity.”

Revised:

→ “Mechanistically, VISTA deficiency is linked to a shift in tumor-associated macrophages (TAMs) from an immunosuppressive phenotype marked by secreted phosphoprotein 1 (SPP1) to one enriched for C-X-C motif chemokine ligand 9 (CXCL9), indicative of a pro-inflammatory state. This shift was accompanied by enhanced recruitment of CXCR3⁺ CD8⁺ T cells, which showed reduced exhaustion and sustained cytotoxic potential. ”

In **Introduction** (Page 4, Line 87-89)

Original:

“Our findings demonstrate that VISTA is crucial for maintaining TAM polarization and immune suppression in the TME of PDAC.”

Revised:

→ “Our findings demonstrate that VISTA deficiency impairs tumor growth by disrupting its role in sustaining immunosuppressive TAM phenotypes and limiting CD8⁺ T cell activity within the PDAC microenvironment. ”

In **Discussion** (Page 18, Line 412)

Original:

“Our study highlights VISTA, a B7 checkpoint molecule, as a pivotal regulator of TAM polarization and the immune landscape in PDAC ^{46, 47}.”

Revised:

→ “Our study highlights VISTA as a modulator of macrophage polarization and immune dynamics in PDAC, although further studies are needed to define its mechanistic role ^{46, 47}.”

3. Figure 2: The role of CSF1R inhibition in TAM modulation is compelling but needs additional experimental replication to confirm its rigor.

(Author's Response/Action)

We thank you for highlighting the importance of experimental reproducibility. In response, we performed additional independent replicates of the CSF1R inhibition experiments across multiple cohorts. These experiments confirmed the robustness of our initial findings, demonstrating consistent TAM depletion and modulation of CD8⁺ T cell infiltration following CSF1R blockade. The results of these additional cohorts are now included in the revised manuscript (Figure 2 and Supplementary Figure S7) and confirm the rigor and reproducibility of our conclusions.

In Figure 2 (Page 41)

Original:

Figure 2. Depletion of VISTA^{high}F4/80^{high}CD11b^{high} TAMs enhances CD8⁺ T cell responses in *Vsir*^{-/-} mice. **a** Flow cytometric analysis of Gr-1⁻F4/80⁺CD11b⁺ TAMs in WT and *Vsir*^{-/-} tumors. Quantification of F4/80^{high}CD11b^{high} and F4/80^{int}CD11b^{int} TAMs as a percentage of CD45⁺ cells (*n* = 4 per group). **b** Anti-CSF1R antibody treatment scheme. **c** Tumor mass at day 28 in WT (*n* = 8) and α CSF1R-treated (*n* = 9) mice. **d** Flow cytometric analysis of Gr-1⁻F4/80⁺CD11b⁺ TAMs in isotype control and α CSF1R-treated WT mice. Quantification of F4/80⁺CD11b⁺ TAMs as a percentage of CD45⁺ cells and normalized to tumor mass (*n* = 3 per

group). **e** Flow cytometric analysis of Gr-1⁻F4/80^{high}CD11b^{high} TAMs and Gr-1⁻F4/80^{int}CD11b^{int} TAMs in isotype control and α CSF1R-treated WT mice. Quantification of Gr-1⁻F4/80^{high}CD11b^{high} and F4/80^{int}CD11b^{int} TAMs as a percentage of CD45⁺ cells ($n = 3$ per group). **f** Flow cytometric analysis of CD4⁺ and CD8⁺ T cells in isotype control and α CSF1R-treated WT mice. Quantification of CD8⁺ T cells as a percentage of CD3⁺ T cells and normalized to tumor mass ($n = 3$ per group). **g** Flow cytometric analysis of effector memory (CD44⁺CD62L⁻) and central memory (CD44⁺CD62L⁺) CD8⁺ T cells in isotype control and α CSF1R-treated tumors. Quantification is shown as a percentage of CD8⁺ T cells ($n = 4$ per group). **h** Flow cytometric analysis of PD-1 and TIM-3 expression on CD8⁺ T cells in isotype control and α CSF1R-treated tumors. Quantification is shown as a percentage of CD8⁺ T cells ($n = 4$ per group). **i** Quantification of CD8⁺ T cells in proximity to macrophages at varying distances (μ m). **j** Immunofluorescence analysis of CD3 and CD11b in the tumor center and margin in WT and *Vsir*^{-/-} tumors. **k** Tumor growth curves of WT BMDM + OT-I T cells and *Vsir*^{-/-} BMDM + OT-I T cells following subcutaneous injection of Pan02 cells ($n = 5$ per group). Data are presented as mean \pm SEM and unpaired two-tailed Student's *t*-test was used for panels **a**, **c–h** and **k**. *n.s.*, not significant.

Revised:

Figure 2. Depletion of VISTA^{high}F4/80^{high}CD11b^{high} TAMs enhances CD8⁺ T cell responses in *Vsir*^{-/-} mice. **a** Flow cytometric analysis of Gr-1⁻F4/80⁺CD11b⁺ TAMs in WT and *Vsir*^{-/-} tumors. Quantification of F4/80^{high}CD11b^{high} and F4/80^{int}CD11b^{int} TAM subsets shown as a percentage of CD45⁺ cells (*n* = 6 per group). **b** Representative flow cytometry plots (left) and MFI of VISTA expression (right) on F4/80^{high}CD11b^{high} (green) and F4/80^{int}CD11b⁺ (blue) TAMs in WT mice. **c** Tumor mass at day 28 in WT (*n* = 8) and α CSF1R-treated (*n* = 9) mice. **d** Flow cytometric analysis of Gr-1⁻F4/80^{high}CD11b^{high} and Gr-1⁻F4/80^{int}CD11b^{int} TAMs in isotype control and α CSF1R-treated WT mice (*n* = 6 per group). **e** Representative flow cytometry plots (left) and MFI of VISTA expression (right) on F4/80^{high}CD11b^{high} (green) and F4/80^{int}CD11b⁺ (blue) TAMs in α CSF1R-treated WT mice (*n* = 6 per group). **f** Flow cytometric analysis of CD4⁺ and CD8⁺ T cells in isotype control and α CSF1R-treated WT mice. Quantification of CD8⁺ T cells shown as a percentage of CD3⁺ T cells and normalized to tumor mass (*n* = 6 per group). **g** Flow cytometric analysis of effector

memory (CD44⁺CD62L⁻) and central memory (CD44⁺CD62L⁺) CD8⁺ T cells in isotype control and α CSF1R-treated WT mice. Quantification shown as a percentage of CD8⁺ T cells ($n = 5$ per group). Data are mean \pm SEM. Unpaired two-tailed Student's *t*-tests were used for statistical analysis in panels **a–g**. *n.s.*, not significant. Data for **a–f** are representative of at least three independent experiments; data for **g** are from two independent experiments with representative results.

Supplementary Figure S7. Effect of anti-CSF1R antibody treatment. **a** Anti-CSF1R antibody treatment scheme. **b** Representative flow cytometry plots showing effective depletion of F4/80^{high}CD11b⁺ tumor-associated macrophages (TAMs) in WT mice treated with anti-CSF1R antibody on day 0 and day 21 after cancer cell inoculation. **c** Flow cytometric analysis of Gr-1⁻F4/80⁺CD11b⁺ TAMs in isotype control *versus* α CSF1R-treated WT mice. Quantification is shown as the percentage of CD45⁺ cells and normalized to tumor

mass ($n = 6$ per group). **d** Flow cytometric analysis of PD-1 and TIM-3 expression on CD8⁺ T cells in isotype control *versus* α CSF1R-treated tumors. Quantification is shown as the percentage of CD8⁺ T cells ($n = 6$ per group). Data are presented as mean \pm SEM and unpaired two-tailed Student's *t*-test was used for panels **c** and **d**. *n.s.*, not significant. All experiments were independently repeated at least twice; representative results are shown.

4. Figure 3: Comparative gene expression heatmaps could better illustrate shifts in *Cxcl9*:*Spp1* ratios.

(Author's Response/Action)

We thank you for this insightful suggestion. Instead of simply presenting the frequencies ratio of *Cxcl9*⁺ to *Spp1*⁺ cells as independent values, we visualized the ratio of *Cxcl9*⁺ to *Spp1*⁺ cells within each macrophage cluster. This approach was adopted because *Cxcl9* exhibits globally higher expression than *Spp1* across macrophage clusters, which skews $\log_2(Cxcl9/Spp1)$ ratios toward the positive range and may obscure biologically meaningful variation in *Spp1*⁺ macrophage clusters. As these values illustrate, the log-ratio approach tends to amplify *Cxcl9*-dominant patterns and underrepresent the contributions of *Spp1*⁺ populations (**Table 1**).

Table 1. Relative expression of *Cxcl9* and *Spp1* across macrophage clusters

Cluster	Average Expression (SCT)			
	Cxcl9	Spp1	Cxcl9/Spp1	$\log_2(Cxcl9/Spp1)$
Mono_4	2.7	1.3	2.1	1.0
MΦ_cluster0	22.3	4.4	5.0	2.3
MΦ_cluster6	3.6	1.2	3.0	1.6
MΦ_cluster1	17.0	1.3	12.8	3.7
MΦ_cluster3	61.0	1.4	42.8	5.4
MΦ_cluster7	27.0	1.4	19.6	4.3
MΦ_cluster5	42.4	1.8	23.8	4.6
MΦ_cluster2	58.9	0.5	117.0	6.9

By focusing on the proportion of positive cells, our visualization minimizes distortion from expression amplitude and more accurately reflects the relative enrichment of each macrophage

clusters. To ensure comparability across clusters, contour thresholds for each gene were derived from total macrophage cells, respectively.

Additionally, to further support the reviewer’s suggestion, we incorporated a quadrant-based co-expression analysis of *Cxcl9* and *Spp1* for each cluster, which is now presented in Supplementary Figure S9d. We believe that this combined visualization strategy provides a more nuanced and interpretable depiction of macrophage heterogeneity, as demonstrated in Figure 3f.

New Supplementary Figure S9d

d Density contour plots of macrophage clusters based on *Cxcl9* (*x*-axis) and *Spp1* (*y*-axis). Each plot depicts a two-dimensional kernel density estimation of single macrophages within a cluster. Red dashed lines denote the average expression levels of each gene across all macrophages, defining quadrant thresholds. Cell frequencies within each quadrant are reported based on relative *Cxcl9* and *Spp1* expression.

In Figure 3f (Page 41)

Original:

Revised:

In Figure 3f (Page 41)

Original:

Revised:

In Results (Page 10, Lines 232-236)

Original:

“Specifically, $Cxcl9^+ Spp1^+$ TAMs were more prevalent in WT tumor (35.5%) than in $Vsir^{-/-}$ tumors (12.6%) (Fig. 3e), while $Cxcl9^+ Spp1^-$ cells were 2.45-fold higher in $Vsir^{-/-}$ TAMs. Further, MΦ_cluster2 had a $Cxcl9:Spp1$ ratio of 2.51, compared to a ratio of 0.69 in MΦ_cluster1, indicating a higher $Cxcl9:Spp1$ ratio in $Vsir^{-/-}$ TAMs could enhance anti-tumor effect (Fig. 3f).”

Revised:

→ “In our data, $Cxcl9$ expression was significantly upregulated in $Vsir^{-/-}$ TAMs, likely contributing to enhanced CD8⁺ T cell infiltration. Specifically, $Cxcl9^+ Spp1^+$ TAMs were more prevalent in WT tumor (35.5%) compared to $Vsir^{-/-}$ tumors (19.0%), while $Cxcl9^+ Spp1^-$ cells were 2.45-fold more frequent in $Vsir^{-/-}$ TAMs (Fig. 3e).”

5. Clarify ambiguities in the discussion, such as “VISTA inhibition could synergize with ICB” (specify conditions).

(Author’s Response/Action)

We thank you for pointing out this ambiguity. In the originally submitted manuscript, we refrained from specifying the conditions under which VISTA inhibition may synergize with immune checkpoint blockade (ICB), as we had not yet performed combination treatment experiments. As such, the statement was left intentionally broad. However, in the revised manuscript, we have included new experimental data (Figure 7) that address this question more directly.

New Figure 7

Figure 7. Effect of anti-VISTA antibody in combination with gemcitabine therapy. a Schematic of the treatment regimen for combination therapy with anti-VISTA antibody and gemcitabine. **b** Tumor volumes of mice treated with isotype control, anti-VISTA antibody, gemcitabine, or combination therapy ($n = 10$ per group). Data are presented as mean \pm SEM. Statistical significance was determined using multiple t -tests, with p -values corrected for multiple comparisons using the Sidak method. *n.s.*, not significant. Data are representative of at least three independent experiments. **c** Schematic summary of key ligand-receptor interactions and immune cell communication pathways in WT *versus* VISTA-deficient tumors, highlighting the enhanced crosstalk in the absence of VISTA.

New Supplementary Figure S18

Supplementary Figure S18. Immune cell dynamics following combination therapy. a

Flow cytometric quantification of F4/80⁺CD11b⁺ TAMs (left) and CD8⁺ T cells (right) at days 10 and 15 post-treatment ($n = 5$), expressed as a percentage of CD45⁺ cells **b** Flow cytometric analysis of iNOS⁺Arg-1⁻ (left) and iNOS⁻Arg-1⁺ (right) TAM subsets in tumor, shown as a percentage of CD45⁺ cells ($n = 10$). **c** Representative flow cytometry plots (left) and quantification bar graphs (right) of CXCL9⁺/SPP1⁺ TAMs in tumor, expressed as a percentage of CD45⁺ cells ($n = 7$). **d** Representative flow cytometry plots (left) and quantification bar graphs (right) of CXCR3⁺CX₃CR1⁻ CD8⁺ T cells and CXCR3⁻CX₃CR1⁺ CD8⁺ T-cell subsets in tumor, shown as a percentage of CD45⁺ cells ($n = 5$). Data are mean \pm SEM. Statistical analysis for panels **a–d** was performed using one-way ANOVA with Sidak's multiple comparisons *post hoc* test. *n.s.*, not significant. Data represent at least three independent experiments.

In Discussion (Page 20, Lines 458-464)

Original:

“The findings from our study provide a strong rationale for targeting VISTA in PDAC. Inhibiting VISTA has the potential to reprogram TAMs to adopt a pro-inflammatory, antitumor phenotype while enhancing recruitment and activation of stem-like CD8⁺ T cells. Such an approach could synergize with existing immune checkpoint inhibitors to improve their efficacy. By simultaneously reprogramming TAMs and augmenting T cell function, VISTA inhibition represents a promising strategy to overcome the immunosuppressive environment of PDAC and improve patient outcomes^{63, 64}.”

Revised:

→ “ICB therapies targeting PD-1 and CTLA-4 have demonstrated limited efficacy in PDAC, underscoring the need for alternative strategies that address multiple layers of immune suppression. In this context, VISTA inhibition offers a promising therapeutic avenue. Notably, anti-VISTA monotherapy combined with gemcitabine elicited a synergistic anti-tumor response, characterized by an increased CXCL9/SPP1 ratio in TAMs and enhanced infiltration of CXCR3⁺ CD8⁺ T cells. This additive effect is particularly relevant for immune-excluded tumors like PDAC, where conventional ICB therapies have failed to yield substantial clinical benefit. Moreover, considering the well-documented toxicity profile of gemcitabine, our findings suggest that VISTA blockade could enhance therapeutic efficacy while potentially reducing dependence on cytotoxic agents.”

6. The depletion of F4/80^{high}CD11b^{high} TAMs via CSF1R inhibition in Figure 2 is an important mechanistic result, but the text focuses more on TAM dynamics than on connecting these changes to downstream CD8⁺ T cell activation (Figure 2g–h).

(Author’s Response/Action)

We thank you for prompting this important point. To clarify the mechanistic link between F4/80^{high} CD11b^{high} TAM depletion and CD8⁺ T cell activation, we now emphasize the following:

(1) Functional significance of F4/80^{high} TAMs: F4/80^{high} TAMs are tissue-resident, embryonically derived populations with immunosuppressive properties, whereas F4/80^{int} macrophages are typically monocyte-derived and infiltrating (Cheng *et al*, Signal transduction and target therapy, 2025). Our data suggest that VISTA sustains the F4/80^{high} subset, which correlates with an immunosuppressive TME.

(2) Restructured manuscript for clarity: Revised Figure 2 now focuses on macrophage phenotypes in VISTA-proficient vs. VISTA-deficient settings, while Figure 3 provides transcriptional and functional characterization of macrophage subsets. We hope this reorganization clarifies the logic from TAM depletion to CD8⁺ T cell infiltration.

In Figures (Page 41)

Original:

Revised:

In **Results** (Page 7, Lines 147-149)

Original:

We found that F4/80^{high}CD11b^{high} TAM subpopulation was significantly decreased in *Vsir*^{-/-} mice compared to WT control (Fig. 2a), suggesting that VISTA supports a distinct F4/80^{high}CD11b^{high} TAM subpopulation that influences TME immune-cell composition.

Revised:

→ “In *Vsir*^{-/-} tumors, we observed a pronounced reduction in the F4/80^{high}CD11b^{high} TAM subset, indicating that VISTA is required to maintain this tissue-resident, immunosuppressive macrophage population (Fig. 2a)²¹. Consistent with this, VISTA expression was higher in F4/80^{high}CD11b^{high} TAMs than F4/80^{int}CD11b^{int} TAMs in WT tumors (Fig. 2b). To test whether selectively depleting these VISTA^{high} TAMs could recapitulate the *Vsir*^{-/-} phenotype, we blocked CSF1R – a key survival signal for tissue-resident macrophages²¹.”

7. The importance of macrophage cluster2's higher *Cxcl9:Spp1* ratio (Figure 3f) is briefly stated in the text but not adequately contextualized regarding CD8⁺ T cell activation and tumor regression.

(Author's Response/Action)

Thank you for commenting on this. In our original manuscript, we briefly stated that a higher *Cxcl9:Spp1* ratio in TAMs could enhance antitumor effects as we wanted to separate our topics on macrophages and CD8⁺ T cells for better delivery, but we agree this statement lacked sufficient context and could be interpreted as an overreach. In the revised manuscript, we have rephrased this section to avoid making a causal interference and instead provide mechanistic context grounded in our data.

In **Results** (Page 12, Lines 235-237)

Original:

“Further, MΦ_cluster2 had a *Cxcl9:Spp1* ratio of 2.51, compared to a ratio of 0.69 in MΦ_cluster1, indicating a higher *Cxcl9:Spp1* ratio in *Vsir*^{-/-} TAMs could enhance anti-tumor effect (Fig. 3f).”

Revised:

→ “The *Cxcl9:Spp1* ratio was 2.51 in MΦ_cluster2 versus 0.69 in MΦ_cluster1 (Fig. 3f, Supplementary Fig. S9d), indicating a shift toward a more chemotactic, immunostimulatory TAM phenotype in *Vsir*^{-/-} tumors.”

8. The results text emphasizes the reduced exhaustion and enhanced effector function of CD8⁺ T cells in *VISTA*-deficient tumors (Figure 5), but the exhaustion marker analysis (e.g., module scores) lacks sufficient explanation to clarify the distinction between effector and terminally exhausted phenotypes.

(Author's Response/Action)

We appreciate your insightful comment regarding the limitations of module score-based analysis in distinguishing between effector and exhausted CD8⁺ T cell phenotypes. As noted, while module scores are useful for summarizing gene set level expression trends, they may lack the resolution needed to capture co-expression patterns of key marker genes or to define discrete functional states.

To address this concern, we performed additional analyses beyond the initial module score approach. Specifically, we focused on CD8⁺ T-cell clusters exhibiting genotype-specific differences by conducting differential gene expression (DEG) analysis to identify key markers that distinguish CD8⁺ T-cell subsets between tumors from WT and *Vsir*-deficient hosts. The results are presented as scatter plots in the revised Supplementary Figure S15a, and cluster-specific marker distributions are further clarified in Figure 5c. To complement these findings, we also conducted a co-expression analysis, which replaced the original module score panel in Figure 5f. In this analysis, cell density was visualized using contour plots, and quadrant thresholds were defined based on the average expression level of each gene across the total CD8⁺ T-cell population. This approach was adopted to account for differences in baseline expression levels of a gene across clusters. We also quantified the proportion of cells within each quadrant to facilitate interpretation of the co-expression patterns.

In detail, we focused on clusters with genotype-specific frequency biases—specifically, WT-enriched clusters 1 and 2, and clusters 0 and 3 enriched in tumors from *Vsir*-deficient hosts—to characterize the functional identity of each CD8⁺ T-cell population. Each cluster was compared to a pooled set of clusters from the opposite genotype to identify DEGs. To minimize bias from unequal cell numbers, we downsampled each cluster to 300 cells prior to DEG analysis and applied thresholds of $\log_2(\text{fold change}) > 1.5$ and adjusted $P\text{-value} < 0.05$. This analysis revealed that clusters 1 and 2 from WT-specific clusters have consistently expressed markers that denote exhausted CD8⁺ T cells such as *Eomes* and associated markers *Cd38*, an NAD⁺-consuming ectoenzyme upregulated in dysfunctional CD8⁺ T cells, where it impairs mitochondrial fitness and mitophagy (Chen *et al.*, *Sci Adv*, 2022). In tumors, high *Cd38* expression marks terminally exhausted T cells with fixed chromatin states and limited reprogrammability to effector function (Philip *et al.*, *Nature*, 2017). In contrast, clusters 0 and 3 from *Vsir*-deficient host, uniquely expressed *Cd226* (Supplementary Figure S15). *Cd226*

functions as a co-stimulatory receptor required for cytotoxicity and ICB responsiveness in CD8⁺ T cells; its loss marks a more terminally exhausted state and correlates with impaired effector function in tumor-infiltrating lymphocytes (Pichler *et al.*, *Front Immunol*, 2022).

Co-expression contour plots further showed that *Eomes*⁺*Cd38*⁺ double-positive cells accounted for over 22.6% of cells in clusters 1 and 2 whereas less than 14.0% of cells in cluster 0 and 3, consistent with a terminally exhausted phenotype (revised Figure 5e left). In contrast, the proportion of *Pdcd1*⁻*Cd226*⁺ single-positive cells were exceeded 20.3% in clusters 0 and 3, whereas less than 12.3% of cells in cluster 1 and 2. Additionally, *Pdcd1*⁺*Cd226*⁺ double-positive cells were counted over 31.7%, whereas less than 18.0% of cells in cluster 1 and 2, indicative of a pre-exhausted phenotype (revised Figure 5e right). These findings are consistent with previous studies of CD8⁺ T cell exhaustion in HIV patients and LCMV clone13-infected mice (Cella *et al.*, *Eur J Immunol*, 2010)

Together, these analyses address the limitations of module score-based interpretation and establish a multi-layered framework for more precisely resolving CD8⁺ T cell differentiation states. These findings support the conclusion that CD8⁺ T cells from WT hosts are enriched for terminally exhausted phenotypes, whereas those from *Vsir*-deficient hosts predominantly exhibit less differentiated, partially exhausted, or effector-like states.

While we acknowledge that these data are preliminary and derived from transcriptomic inference rather than direct functional assays, we believe they provide a coherent basis for the observed phenotypic differences. Ongoing functional and epigenetic experiments will directly test these hypotheses in future studies.

In addition, while revising Figure 5f, we have identified a typographical error in Figure 5e that has now been corrected. We apologize for this oversight.

In Figure 5f (Page 46)

Original(left)/Revised(right):

e Scatter plot of log₂ fold changes: CD8⁺_cluster1 (*x*-axis) versus CD8⁺_cluster2 (*y*-axis), each compared with *Vsir*-deficient clusters (CD8⁺_cluster0 and cluster3). Genes upregulated only in CD8⁺_cluster1 (*gold*), only in CD8⁺_cluster2 (*green*) or in both WT-specific clusters (*blue*) are highlighted.

Supplementary Figure S15. Transcriptional differences between WT- and $Vsir^{-/-}$ -specific CD8⁺ T-cell subclusters in the tumor microenvironment. **a** Scatter plot visualizing log₂ fold changes in gene expression for CD8⁺_cluster1 (x-axis) and CD8⁺_cluster2 (y-axis), each compared to the $Vsir^{-/-}$ clusters (CD8⁺_cluster0 and cluster3), respectively. Each point represents a gene, color-coded by differential expression: *gold* for genes upregulated only in CD8⁺_cluster1, *green* for those upregulated only in CD8⁺_cluster2, and *blue* for genes upregulated in both WT-specific clusters. **b** Differential expression analysis of $Vsir^{-/-}$ CD8⁺ T-cell clusters relative to WT-specific subsets. The scatter plot shows log₂ fold changes for CD8⁺_cluster0 (x-axis) and CD8⁺_cluster3 (y-axis), each compared to WT clusters (CD8⁺_cluster1 and cluster2), respectively. Genes are color-coded as follows: *light red* for genes upregulated only in CD8⁺_cluster0, *cyan* for those upregulated only in CD8⁺_cluster3, and *red* for genes upregulated in both $Vsir^{-/-}$ clusters. **c** Violin plots of normalized expression levels of selected genes across CD8⁺ T-cell clusters. Clusters are grouped by genotype: WT-specific clusters [CD8⁺_cluster1 (*gold*) and cluster2 (*green*)] and $Vsir^{-/-}$ -specific [CD8⁺_cluster0 (*light red*) and cluster3 (*cyan*)].

In Figure 5e

Original:

Revised:

Supplementary figures:

1.S3: Immunofluorescence staining of CD8⁺ T cells and CD11b⁺ macrophages is crucial to visualize immune infiltration but is only superficially discussed in the context of Figure 1.

(Author's Response/Action)

We thank you for highlighting the importance of our immunofluorescence data in visualizing the spatial distribution of immune infiltrates. As we did not include F4/80 staining in our immunofluorescence analysis, we presented these data in the *Supplementary* section. In response to the comment, we have revised the *Results* section to more clearly integrate and interpret the results from Supplementary Fig. S3. Specifically, we now emphasize that IF analysis of Pan02 tumors demonstrated a marked increase in CD8⁺ T cells (red) and F4/80⁺ or CD11b⁺ myeloid cells (green/red) infiltration in *Vsir*^{-/-} and α VISTA-treated tumors compared to WT controls. These findings visually confirm and complement the quantitative flow cytometric data presented in Fig. 1d, reinforcing that VISTA deficiency promotes selective recruitment of CD8⁺ T cells and macrophages into the tumor microenvironment.

The revised text is now included in the *Results* to explicitly connect the IF data to the overall conclusion that VISTA regulates tumor immune infiltration. We thank you for encouraging a more integrated presentation of this dataset.

In **Results** (Page 6, Lines 124-126)

Original:

“Immunofluorescence staining of Pan02 tumors further revealed increased infiltration of CD8⁺ T cells (green) and CD11b⁺ myeloid cells (red) in α VISTA-treated tumors compared to control (Supplementary Fig. S3a).”

Revised:

→ “Immunofluorescence staining corroborated these findings, **revealing increased infiltration of CD8⁺ T cells (green/red) and F4/80⁺ (green) or CD11b⁺ myeloid cells (red) in both *Vsir*^{-/-}**

and α VISTA-treated tumors compared to WT controls (Fig. 1d, Supplementary Fig. S5a and b).”

In Supplementary Fig. S3a

Original:

Revised:

2. Supplementary Figure S5 contains important flow cytometric data on CSF1R-mediated macrophage depletion and VISTA expression, but its relationship to Figures 2 and 3 is not explicitly stated. For instance, the text does not sufficiently highlight how the functional

phenotypes of TAMs (e.g., changes in *iNOS* and *Arginase-1* expression in *S5a-c*) connect to their pro-inflammatory or immunosuppressive states.

(Author’s Response/Action)

We thank you for pointing this out. We agree that the connection between the flow cytometry findings in Supplementary Figure S5 and the transcriptional profiling in Figure 2 and 3 was not sufficiently explained. In response, we have edited the Result section to bridge this gap and clarify the relationship between surface marker-defined subsets and their functional states.

In Figure 3, we aimed to further characterize TAM heterogeneity using scRNA-seq. Although we initially attempted to stratify macrophages by F4/80 expression levels, this was not feasible due to limitations of transcript-level resolution for these surface markers. Therefore, we performed unsupervised clustering and identified eight distinct monocyte/macrophage subsets.

These clarifications are now reflected in the revised manuscript, and we hope this better integrates our data to support a coherent model of VISTA-mediated macrophage regulation.

In Supplementary Figure S5

Original:

Revised:

3.S7 supports Figure 3 but is insufficiently explained in the results, particularly regarding the functional differences between *Spp1*⁺ and *Cxcl9*⁺ macrophages.

(Author's Response/Action)

We thank you for this insightful comment. In the revised manuscript, we have expanded the *Results* to provide a more explicit interpretation of Supplementary Figure S7, which details the functional differences between *Spp1*⁺ and *Cxcl9*⁺ TAMs. We now describe the differential enrichment of gene signatures and pathways associated with immunosuppressive *versus* pro-inflammatory programs, thereby clarifying the functional polarization of these TAM subsets. We believe this addition directly addresses your concern and improves the interpretability of the result.

In **Results** (Page 10-11, Lines 237-246)

Original:

“Gene set enrichment analysis (GSEA) showed that *Cxcl9*⁺ TAMs were consistently enriched in MΦ_cluster2 compared to MΦ_cluster1, whereas *Spp1*⁺ TAMs were predominantly in MΦ_cluster1 (Supplementary Fig. S7d, top). Kyoto Encyclopedia of Genes and Genomes (KEGG) pathway analysis further demonstrated that MΦ_cluster2 was enriched in cytokine-cytokine receptor interactions and antigen processing/presentation pathways (Supplementary Fig. S7d, bottom).”

Revised:

→ “Gene set enrichment analysis (GSEA) confirmed that *Spp1*⁺ TAMs were enriched in MΦ_cluster1 (NES = 1.84, *P* < 0.001), whereas *Cxcl9*⁺ TAMs were significantly enriched in MΦ_cluster2 (NES = -2.51, *P* < 0.001) (Supplementary Fig. S9e, top). KEGG pathway analysis further showed that MΦ_cluster2 was enriched for cytokine-cytokine receptor interaction and antigen processing and presentation pathways (Supplementary Fig. S9e, bottom), supporting these TAMs may promote CD8⁺ T cell activation and function.”

4. Supplementary Figure S8 discusses the lack of difference in phagocytosis capacity between WT and VISTA-deficient macrophages (S8a), which could provide a contrasting point to emphasize functional gains in antigen presentation.

(Author’s Response/Action)

We thank you for this important point. To address this, we emphasize that VISTA deficiency does not result in a general upregulation of all macrophage functions. As shown in Supplementary Figure S8a, the phagocytosis capacity of VISTA-deficient macrophages remains comparable to that of WT controls, indicating that baseline innate macrophage functions are preserved. This lack of change in phagocytosis highlights that the observed increases in antigen presentation and T cell activation are likely function-specific gains, rather than a global enhancement of macrophage activity. We have added clarification to this point in the revised *Results*.

In **Results** (Page 11, Lines 251-253)

Original:

“While WT and *Vsir*^{-/-} BMDMs showed comparable phagocytic abilities (Supplementary Fig. S8a), transcriptomic and pathway analyses revealed notable changes in immunomodulatory capacity.”

Revised:

→ “To directly test antigen presentation capacity, we performed an antigen cross-presentation assay by measuring SIINFEKL–H2-K^b complex on WT and *Vsir*^{-/-} BMDMs. Upon SIINFEKL peptide loading, no difference in MFI were observed (Supplementary Fig. S11d). However, *Vsir*^{-/-} BMDMs exposed to full-length OVA protein displayed significantly higher SIINFEKL–H2-K^b levels, especially at higher antigen concentrations (Fig. 4d), indicating enhanced antigen processing and cross-presentation in the absence of VISTA. Notably, this occurred without changes in CD80, CD86, or I-A/I-E expression (Supplementary Fig. S13), suggesting that VISTA loss selectively enhances antigen-presenting function without broadly activating macrophages.”

Reviewer #2’s Comments and Responses

Major comments:

1. Most in vivo experiments seem to utilize 3-5 mice per group with no statement of replicates, unless I have missed a statement of experiment replication somewhere in the manuscript. Please clarify in methods and/or figure legends how many samples per condition were used, whether any samples were removed from consideration and by what criteria, and how many independent experiments were performed for each experiment.

(Author’s Response/Action)

We deeply regret this significant oversight and take full responsibility for the lack of clarity regarding sample sizes and experimental replication in the initial submission. This was entirely my mistake, as I failed to ensure that the necessary details were properly included in the manuscript prior to submission.

To address this issue, we have carefully revised the methods section and all figure legends to explicitly detail the exact number of mice used per condition and the number of independent experiments for each study are revised in the following sections:

Figure Legends

→ **Figure 1. VISTA deficiency attenuates pancreatic tumor growth in *Vsir^{-/-}* mice.** **a** Tumor growth curve of WT and *Vsir^{-/-}* mice orthotopically implanted with Pan02 cells (WT: $n = 35$; *Vsir^{-/-}*: $n = 35$). Statistical significance was determined using multiple *t*-tests with Holm-Sidak correction for multiple comparisons. **b** Tumor mass at day 28 in WT ($n = 17$), *Vsir^{-/-}* ($n = 19$), and α VISTA-treated mice ($n = 10$) (left). Representative *ex vivo* tumor images at day 28 are shown on the right (**a** and **b**). **c** Kaplan-Meier survival analysis of WT and *Vsir^{-/-}* mice ($n = 10$ per group). *P*-value calculated using the log-rank (Mantel-Cox) test. **d** Flow cytometric quantification of F4/80⁺CD11b⁺ TAMs (left) and CD8⁺ T cells (right) in WT and *Vsir^{-/-}* mice at days 14, 21, 28, and 35, shown as a percentage of CD45⁺ cells ($n = 10$ per group). **e** Immunohistochemical images of F4/80⁺ macrophages (top) and CD8⁺ T cells (bottom) in WT and *Vsir^{-/-}* tumors (day 28). Quantification of CD8⁺ T cells per field ($n = 3$ per group). **f** Flow cytometric analysis of iNOS⁺ TAMs, shown as a percentage of CD45⁺ cells and normalized to tumor mass ($n = 6$ per group). **g** Flow cytometric analysis of Arg1⁺ TAMs, shown as mean fluorescence intensity (MFI) and as a percentage of CD45⁺ cells ($n = 8$ per group). **h** Proportions of CD8⁺ T-cell subsets shown as a percentage of CD3⁺ T cells ($n = 8$ per group). Data are mean \pm SEM. Unpaired two-tailed Student's *t*-tests were applied to panels **d–h**. *n.s.*, not significant. Data represent at least three independent experiments.

Figure 2. Depletion of VISTA^{high}F4/80^{high}CD11b^{high} TAMs enhances CD8⁺ T cell responses in *Vsir^{-/-}* mice. **a** Flow cytometric analysis of Gr-1⁻F4/80⁺CD11b⁺ TAMs in WT and *Vsir^{-/-}* tumors. Quantification of F4/80^{high}CD11b^{high} and F4/80^{int}CD11b^{int} TAM subsets shown as a percentage of CD45⁺ cells ($n = 6$ per group). **b** Representative flow cytometry plots (left) and MFI of VISTA expression (right) on F4/80^{high}CD11b^{high} (green) and F4/80^{int}CD11b⁺ (blue) TAMs in WT mice. **c** Tumor mass at day 28 in WT ($n = 10$) and α CSF1R-treated ($n = 10$) mice. **d** Flow cytometric analysis of Gr-1⁻F4/80^{high}CD11b^{high} and Gr-1⁻F4/80^{int}CD11b^{int} TAMs in

isotype control and α CSF1R-treated WT mice ($n = 6$ per group). **e** Representative flow cytometry plots (left) and MFI of VISTA expression (right) on F4/80^{high}CD11b^{high} (green) and F4/80^{int}CD11b⁺ (blue) TAMs in α CSF1R-treated WT mice ($n = 6$ per group). **f** Flow cytometric analysis of CD4⁺ and CD8⁺ T cells in isotype control and α CSF1R-treated WT mice. Quantification of CD8⁺ T cells shown as a percentage of CD3⁺ T cells and normalized to tumor mass ($n = 6$ per group). **g** Flow cytometric analysis of effector memory (CD44⁺CD62L⁻) and central memory (CD44⁺CD62L⁺) CD8⁺ T cells in isotype control and α CSF1R-treated WT mice. Quantification shown as a percentage of CD8⁺ T cells ($n = 6$ per group). Data are mean \pm SEM. Unpaired two-tailed Student's *t*-tests were used for statistical analysis in panels **a–g**. *n.s.*, not significant. **Data for a–f are representative of at least three independent experiments; data for g are from two independent experiments with representative results.**

Figure 4. *Vsir* deficiency enhances antigen presentation and stimulates CD8⁺ T cell responses. **a** Differential gene expression analysis of macrophage subsets. Volcano plot comparing M Φ _clusters 2 and 3 with M Φ _cluster1. **b** Schematic of the MHC class I antigen processing and presentation pathway. **c** KEGG pathway analysis of M Φ _cluster2 (light green), M Φ _cluster3 (green), and combined M Φ _clusters 2 and 3 (brown), showing pathways satisfying $p_{adj} < 0.05$. **d** Flow cytometric analysis of H2-K^b-SIINFEKL complex on WT and *Vsir*^{-/-} BMDMs stimulated with OVA protein at varying concentrations. MFI quantification shown on the right. **e** Proliferation of OT-I CD8⁺ T cells co-cultured with WT or *Vsir*^{-/-} BMDMs loaded with OVA protein (top left) or SIINFEKL peptide (bottom right), shown as the percentage of divided CD8⁺ T cells. **f** Flow cytometric analysis of IFN- γ , TNF- α , and perforin (PFN) production in OT-I CD8⁺ T cells co-cultured with WT or *Vsir*^{-/-} BMDMs loaded with OVA protein. **g** Quantification of CD8⁺ T cells in proximity to macrophages across varying distances (μ m). **h** Tumor growth curves following co-injection of BMDMs and CD3⁺ T cells ($n = 6$ per group). **i** Tumor growth curves following co-injection Pan02 cells with OT-I CD8⁺ T cells and WT or *Vsir*^{-/-} BMDMs ($n = 5$ per group). Data are presented as mean \pm SEM. Unpaired two-tailed Student's *t*-tests were applied for panels **d–f**. **Multiple *t*-tests with Holm-Sidak correction were used for h and i.** *n.s.*, not significant. **All experiments were independently repeated at least twice; representative results are shown.**

Figure 5. *Vsir* deficiency delays CD8⁺ T cell exhaustion with increased CXCR3 expression. **a** UMAP of single-cell RNA-seq data showing CD8⁺ T-cell subsets in WT and *Vsir*^{-/-} tumors; subpopulations are color-coded. **b** Dot plot of differentially expressed genes across six CD8⁺ T-cell clusters, with color and dot size representing expression level and cell proportion, respectively. **c** Genotype-split UMAP depicting CD8⁺ T cells from WT (*blue*) and *Vsir*^{-/-} (*red*) tumors. **d** Relative abundance of each CD8⁺ T-cell subsets in WT and *Vsir*^{-/-} tumors. **e** Scatter plot of log₂ fold changes: CD8⁺_cluster1 (*x-axis*) versus CD8⁺_cluster2 (*y-axis*), each compared with *Vsir*-deficient clusters (CD8⁺_cluster0 and cluster3). Genes upregulated only in CD8⁺_cluster1 (*gold*), only in CD8⁺_cluster2 (*green*) or in both WT-specific clusters (*blue*) are highlighted. **f** Flow cytometry plots (left) and quantification (right) of CXCR3⁺CX₃CR1⁺ CD8⁺ T cells in WT (*n* = 8) and *Vsir*^{-/-} (*n* = 7) tumors. **g** Flow cytometry plots (left) and quantification (right) of TCF-1⁺PD-1⁻ CD8⁺ T cells in WT and *Vsir*^{-/-} tumors (*n* = 7 per group). **h** Flow cytometry plots (left) and quantification (right) of terminally exhausted TCF-1⁻PD-1⁺CXCR3⁺CX₃CR1⁻ CD8⁺ T cells (Tex^{term}) in WT and *Vsir*^{-/-} tumors (*n* = 7 per group). Data are mean ± SEM and unpaired two-tailed Student's *t*-tests were applied for panels **f–h**. *n.s.*, not significant. Experiments were independently repeated at least twice; representative results are shown.

These revisions have been thoroughly reviewed and confirmed by Suk-Kyung Shin, Hang-Rae Kim, and myself to ensure accuracy and transparency. I sincerely apologize again for the error and thank you for bringing this to our attention.

2. *The findings of this paper are highly reliant on a relatively immunogenic and genetically irrelevant PancO2 PDAC model. Numerous, more genetically relevant and poorly immunogenic PDAC models exist at this time; can the key results of this study be replicated in a more rigorous KPC-type model?*

(Author's Response/Action)

We appreciate this critical observation and fully acknowledge the limitations of the Pan02 model in terms of both immunogenicity and genetic relevance. In response, we performed additional experiments using the KPC001 tumor model, which is derived from the *Kras*^{G12D/+}; *Trp53*^{R172H/+}; *Pdx1-Cre* (KPC) genetically engineered mouse model and recapitulates the mutational landscape and immunosuppressive tumor microenvironment of human PDAC better than Pan02 model.

Consistent with our findings in the Pan02 model, VISTA deficiency significantly impaired tumor growth in KPC001-bearing mice, albeit to a lesser extent (Supplementary Fig. S1a). Immune profiling at multiple timepoints confirmed that KPC001 tumors harbored lower baseline CD3⁺ T cell infiltration than Pan02, as expected for a poorly immunogenic model. Nevertheless, VISTA deficiency in KPC001 tumors was associated with increased infiltration of CD8⁺ T cells and F4/80⁺ TAMs (Supplementary Fig. S1b and c), consistent with our observations in the Pan02 model. Furthermore, TAMs in *Vsir*^{-/-} KPC001 tumors exhibited increased I-A/I-E expression (Supplementary Fig. S1d), indicative of a more pro-inflammatory phenotype. Although we did not observe a reproducible increase in iNOS⁺ macrophages in this model, the shift in antigen presentation phenotype supports the overall conclusion that VISTA regulates macrophage polarization.

New Supplementary Figure S1

Supplementary Figure S1. VISTA deficiency reduces tumor growth in KPC001 Model. a Tumor growth curve of WT and *Vsir*^{-/-} mice orthotopically implanted with KPC001 cells (WT: *n* = 42, *Vsir*^{-/-}: *n* = 42). Data are mean ± SEM. Significance was determined using multiple *t*-tests, with *p*-values corrected for multiple comparisons using the Holm-Sidak method. **b** Flow cytometric quantification of F4/80⁺CD11b⁺ tumor-associated macrophages (TAMs) in WT (*n* = 19) and *Vsir*^{-/-} (*n* = 17) mice at days 10, 15, and 21, shown as a percentage of CD45⁺ cells. **c** Flow cytometric quantification of CD8⁺ T cells in WT (*n* = 19) and *Vsir*^{-/-} (*n* = 16) mice at days 10, 15, and 21, shown as a percentage of CD45⁺ cells. One-way ANOVA with Sidak's *post hoc* test was applied for panels **b** and **c**. **d** Flow cytometric analysis of I-A/I-E⁺ CD206⁻ TAMs in tumor, expressed as a percentage of CD45⁺ cells (*n* = 5). Data are mean ± SEM. Statistical analysis panels for **b–d** were performed using an unpaired two-tailed Student's *t*-test. *n.s.*, not significant. Data represent at least three independent experiments.

In **Results** (Page 5, Lines 102-108)

Original:

To investigate how VISTA deficiency impacts the immune-oncologic landscape of PDAC, we utilized a syngeneic orthotopic model comparing tumor growth between wild-type (WT) and VISTA-deficient (*Vsir*^{-/-}) mice. Pan02 cells (5×10^5) were orthotopically inoculated into the pancreas tail of both groups. VISTA deficiency significantly impaired pancreatic tumor growth, with tumor volume notably reduced by day 21 post-inoculation, becoming even more pronounced by day 28 (Fig. 1a). By day 28, tumor volumes in *Vsir*^{-/-} mice were approximately threefold smaller than those of WT control.

Revised:

→ “To investigate how VISTA deficiency impacts the immune-oncologic landscape of PDAC, we utilized syngeneic orthotopic models to compare tumor growth between wild-type (WT) and VISTA-deficient (*Vsir*^{-/-}) mice. Pan02 and KPC tumor cells (5×10^5) were orthotopically inoculated into the pancreatic tail of each group. VISTA deficiency significantly impaired tumor growth in both models, resulting in a 2.5-fold reduction in tumor size in the

immunogenic Pan02 model and a 1.8-fold reduction in the less immunogenic KPC model at endpoint (Fig. 1a, Supplementary Fig. S1a). These findings were further supported by pharmacological blockade of VISTA in WT mice, where treatment with an anti-VISTA antibody (α VISTA) similarly suppressed tumor growth in the Pan02 model (Fig. 1b, Supplementary Fig. S2).”

In **Results** (Pages 5-6, Lines 120-124)

Original:

To identify the underlying mechanisms driving these differences in tumor growth and survival, we conducted flow cytometry and immunohistochemistry analyses. VISTA deficiency selectively promoted the infiltration of both tumor-associated macrophages (TAMs; F4/80⁺CD11b⁺) and CD8⁺ T cells into TME, peaking at day 28 (Fig. 1e, f, Supplementary Fig. S2a).

Revised:

→ “To characterize the immunological consequences of VISTA deficiency, we conducted flow cytometry and immunohistochemistry analyses in orthotopic tumor models. *Vsir*^{-/-} mice exhibited a 1.5-fold increase in F4/80⁺CD11b⁺ TAMs and a 1.8-fold (Pan02) to 2.0-fold (KPC) increase in CD8⁺ T cells compared to WT controls (Fig. 1d, Supplementary Fig. S1b, c, and S4a).”

In **Results** (Page 6, Lines 129-132)

Original:

Importantly, TAMs in *Vsir*^{-/-} mice exhibited a pro-inflammatory phenotype with increased iNOS expression (Fig. 1g) and decreased Arginase 1 expression (Fig. 1h), suggesting a shift towards a more tumor-suppressive phenotype.

Revised:

→ “Critically, TAMs in *Vsir*^{-/-} tumors acquired a pro-inflammatory phenotype, characterized by increased iNOS⁺ macrophages (Pan02) and I-A/I-E⁺CD206⁻ macrophages (KPC) (Fig. 1f, Supplementary Fig. S1d), accompanied by reduced Arginase-1⁺ macrophages (Fig. 1g).”

Additionally, we employed the KPC001 model to evaluate the therapeutic efficacy of VISTA targeting in combination with gemcitabine (Fig. 7a). Tumor reduction was observed following anti-VISTA monotherapy, with a further synergistic or additive effect seen in the combination treatment with gemcitabine (Fig. 7b). Key mechanistic findings were recapitulated in this model: CXCL9 and SPP1 levels were elevated in the anti-VISTA group compared to PBS controls (Supplementary Fig. S18c), and an increase in CXCR3⁺ CD8⁺ T cells was observed in both the anti-VISTA and combination therapy groups (Supplementary Fig. S18d). These findings reinforce the mechanistic conclusions drawn from the Pan02 model and independently validate VISTA’s immunoregulatory role in a genetically and immunologically relevant PDAC model, thereby strengthening the translational relevance of our findings. To further deepen the mechanistic insight in this clinically relevant setting, we plan to perform single-cell RNA sequencing of KPC001 tumors in future studies.

New Figure 7

Figure 7. Effect of anti-VISTA antibody in combination with gemcitabine therapy. a Experimental scheme for combination therapy. **b** Tumor volumes in mice treated with combination therapy ($n = 10$ per group). Data are mean \pm SEM. Significance was determined using multiple t -tests, with p -values corrected for multiple comparisons using the Sidak method. *n.s.*, not significant. Experiments were repeated at least three times independently. **c** Schematic summarizing key intercellular communication pathways in wild-type *versus* VISTA-deficient tumors.

In **Results** (Page 17, Lines 400-402)

Original:

“These findings reveal how VISTA deficiency reprograms the TME from an immunosuppressive to an immune-stimulatory state, highlighting the potential for VISTA as a therapeutic target to boost immune responses against pancreatic cancer.”

Revised:**→ “VISTA Blockade Synergizes with Gemcitabine to Remodel the Pancreatic Tumor Immune Landscape**

To determine whether these VISTA-dependent signaling network we identified is therapeutically exploitable, we turned to the KPC orthotopic model and compared anti-VISTA monotherapy, gemcitabine, and their combination (Fig. 7a). Both monotherapies produced a modest but significant reduction in tumor burden; however, the combination elicited a markedly greater suppression, indicating at least an additive—and potentially synergistic—anti-tumor effect (Fig. 7b).

Immune profiling revealed that anti-VISTA monotherapy alone increased total TAMs and increased CD8⁺ T cells (Supplementary Fig. S18a). Importantly, the antibody skewed TAMs toward an iNOS⁺ phenotype while producing only minor changes in Arg1⁺ TAMs (Supplementary Fig. S18b). Consistent with our CellChat predictions, CXCL9⁺/SPP1⁺ TAMs were enriched in the anti-VISTA treatment group (Supplementary Fig. S18c), both anti-VISTA and the combination regimen boosted infiltration of CXCR3⁺ CD8⁺ T cells (Supplementary Fig. S18d)—hallmarks of the CXCL9–CXCR3 chemotactic axis we mapped in *Vsir*^{-/-} tumors. These findings validate our mechanistic model and demonstrate that pharmacologic VISTA blockade can re-program the myeloid–T-cell circuit even in the presence of cytotoxic chemotherapy.”

In **Methods** (Page 21, Line 477)

Original:

“The Pan02 cell line, derived from a methylcholanthrene-induced PDAC, was a generous gift from Dr. Keehoon Jung (Seoul National University)⁶⁵.”

Revised:

→ “The Pan02 cell line from National Cancer Institute (NCI, 0509770), and KPC001 cell line isolated from the tumor of a 5– to 6–month-old genetically engineered mouse model of *Kras*^{G12D}*p53*^{R172H/+}*Pdx-1-Cre* mice were used^{66,67}.”

In **Methods** (Pages 21-22, Lines 489-490)

Original:

“After upper left abdominal incision, pancreatic tails were exposed and injected with 5×10^5 Pan02 cells resuspended in cold...”

Revised:

→ “After upper left abdominal incision, pancreatic tails were exposed and injected with **either** 5×10^5 Pan02 **or KPC001** cells...”

In **Methods** (Page 22, Line 503-504)

Original:

“or Rat IgG2 α , κ isotype control (Clone 2A3, *InVivoMab*, BioXCell) 13 days prior to tumor inoculation, followed by three consecutive daily injections of 0.5 mg.”

Revised:

→ “...0.5 mg. **For combination therapy, 50 mg/kg gemcitabine (Selleckchem, Houston, TX, USA) and/or anti-VISTA antibody (Clone 13F3, *InVivoMab*, BioXCell, Lebanon, NH, USA) were administered intraperitoneally, starting from day 6 after cancer cell injection. Treatments were given every 3 days.**”

*3. It remains unclear whether (i) VISTA deficiency primarily affects **macrophage polarization leading to secondary effects on T cell differentiation**; (ii) VISTA deficiency **primarily affects T cell differentiation/exhaustion leading to secondary effects on macrophage polarization**; or (iii) these effects occur **independently**. Can the authors provide any data that indicates whether VISTA deficiency on macrophages or T cells alone is necessary or sufficient for their phenotypes?*

(Author’s Response/Action)

We thank you for this insightful mechanistic question. While we have not yet performed experiments involving cell type-specific deletion of VISTA or adoptive transfer models, our

current data supports a model in which VISTA primarily regulates macrophage polarization, which then influences CD8⁺ T cell infiltration and activation.

(i) Macrophage-centric mechanism:

In our *in vitro* co-culture system, M0 WT and VISTA KO BMDMs showed no differences in OT-I CD8⁺ T cell proliferation. However, when BMDMs were polarized with LPS and IFN- γ , VISTA KO BMDMs combined with OVA proteins induced significantly greater OT-I T cell proliferation compared to unpolarized BMDMs.

In our *in vivo* system, when OT-I CD8⁺ T cells were co-injected with VISTA KO BMDMs, but not with WT BMDMs, tumor regression was observed in Rag1 KO mice, indicating a macrophage-centric mechanism (Relocated Figure 4i).

Proteome profiling (next page) further revealed decreased phospho-STAT3 in VISTA KO BMDMs even in the absence of T cells, indicating that the shift in macrophage phenotype begins prior to T cell interaction.

In vivo, VISTA deficiency led to increased expression of chemokines, such as CXCL9 in TAMs, which are known to recruit and activate CD8⁺ T cells, further supporting a model where macrophage shift is the initiating event.

- (ii) Direct effects on T cells cannot be fully excluded, but our data do not support a primary role for VISTA in directly regulating T cell differentiation or exhaustion in this context. Instead, the observed changes in T cell phenotype and function appear secondary to alterations in macrophage polarization and cytokine milieu.

Relocated Figure 4i:

In summary, while further investigation is needed to definitively dissect cell-intrinsic roles, our current findings indicate that VISTA deficiency predominantly affects macrophage polarization, which in turn shapes CD8⁺ T cell responses in the tumor microenvironment.

4. Can the authors explain why VISTA-KO BMDMs show a functional phenotype as compared to VISTA WT BMDMs in vitro presumably in the absence of VISTA ligand? What is acting as the VISTA ligand in the WT BMDMs?

(Author's Response/Action)

We appreciate your thoughtful question regarding the ligand-receptor dynamics of VISTA in our *in vitro* system. While endogenous ligand(s) for VISTA remains incompletely characterized, several mechanisms may explain the functional differences between WT and VISTA-KO BMDMs:

- (i) Autocrine/Paracrine VISTA signaling: In WT BMDMs, VISTA may act as both a receptor and a ligand in homotypic interactions (Yoon *et al.*, *Science*, 2015). For example, VISTA on WT macrophages could engage with VISTA on neighboring macrophages, creating an autocrine/paracrine immunosuppressive loop. This signaling would be absent in VISTA-KO macrophages, leading to enhanced pro-inflammatory responses.
- (ii) Known ligands in myeloid cells: PSGL-1, expressed majorly on T cells, binds to VISTA and suppresses T-cell activation (Johnston *et al.*, *Nature*, 2019). In our system, WT BMDMs may engage PSGL-1 on neighboring CD8⁺ T cells, while VISTA-KO BMDMs lack this interaction. VSIG-3, a proposed ligand for VISTA (Wang *et al.*, *Immunology*, 2018), could mediate inhibitory signaling in WT BMDMs.
- (iii) MHC-I interaction: Recent studies suggest VISTA interacts with MHC-I on macrophages, modulating antigen presentation and T cell activation. Loss of VISTA in KO BMDMs may disrupt this interaction, enhancing MHC-I-mediated antigen presentation.

We believe this phenotype may persist *in vitro* without exogenous ligand by either tonic signaling, where VISTA could constitutively suppress pro-inflammatory pathways (*e.g.*, NF- κ B) in WT macrophages, even in ligand-free conditions); or by cis-interactions, where VISTA interacts with co-receptors (*e.g.*, MHC-I) on the same membrane. While defining VISTA's ligand(s) *in vitro* requires further study, which we intend to address in the future, we believe our findings highlight its non-redundant role in myeloid cell self-regulation, independent of exogenous ligands.

5. *Key points of the proposed mechanism could benefit from in vivo validation. Is the CXCL9/CXCR3 axis required for the robust antitumor phenotype in the VISTA-KO mice? Are*

macrophages derived from tumors grown in VISTA-KO versus WT mice differentially able to mediate T cell chemotaxis, priming, and/or function? The validation only in BMDCs does not recapitulate the key, complex biology of the TME.

(Author's Response/Action)

We thank you for this important comment. We agree that *in vivo* validation of CXCL9/CXCR3 axis and macrophage-mediated T cell regulation would provide stronger mechanistic insight into the role of VISTA. However, due to the extensive time required to develop and *execute in vivo* blockade studies (CXCR3 neutralization or macrophage adoptive transfer), we were unable to complete these experiments within the timeframe of our current study.

We are planning on designing follow-up studies to functionally interrogate the CXCL9/CXCR3 axis and the role of VISTA-deficient TAMs in T cell recruitment using *in vivo* models. We have added this limitation and future direction to the revised *Discussion* section. We appreciate your suggestion in enlightening us with what could be done next on our ongoing research.

In **Discussion** (Page 19, Lines 439-444)

Original:

Macrophage secreting CXCL9/10 are thought to support anti-tumor effects of checkpoint inhibitors like anti-PD-1 and anti-CTLA-4 antibodies by recruiting stem-like CD8⁺ T cells ⁵⁵. ⁵⁶. The elevated levels of IFN- γ -producing CD8⁺ T cells observed in VISTA-deficient mice indicate enhanced immune activation. Therefore, targeting VISTA may delay CD8⁺ T cell exhaustion and promote durable anti-tumor responses in PDAC.

Revised:

→ “The elevated levels of IFN- γ -producing CD8⁺ T cells observed in VISTA-deficient mice indicate enhanced immune activation. **While these findings indicate that VISTA may influence CD8⁺ T cell recruitment and activation via the CXCL9/CXCR3 axis, further studies are required to establish whether this pathway is essential for these observed immune effects.**”

Minor comments:

1. Fig 2B: What is the rationale for your anti-CSF1R regimen starting 2 weeks before tumor implantation? Why not apply the antibody during/after implantation, in context of tumor progression? What macrophages are you depleting with the D-14 anti-CSF1R regimen?

(Author's Response/Action)

We thank you for this important question. Our rationale for initiating anti-CSF1R treatment at day -14 relative to tumor implantation was to systemically deplete CSF1R⁺ tissue-resident macrophages and circulating monocytes prior to tumor implantation, thereby isolating their role in early tumor seeding and immune priming. Though we are aware that temporal modulation of CSF1R⁺ cells at later stages would provide critical insights into TAM reprogramming, day -14 timing targets embryonically derived, self-renewing macrophages and bone marrow-derived monocytes, which are critical for initial immune surveillance. By depleting macrophages before TME establishment, we aimed to assess how their absence during the initiation phase alters CD8⁺ T cell infiltration and spatial organization.

2. I would recommend removing the artificial "gates" on the scRNA UMAP plots; let the clusters speak for themselves. Lines drawn in UMAP space have no inherent meaning

(Author's Response/Action)

We appreciate your thoughtful suggestion regarding the artificial contour lines surrounding UMAP clusters (Fig. 3b and c and Fig. 5b and c). We fully agree that such lines may impose misleading boundaries that do not reflect the underlying structure of the high-dimensional data projected in UMAP space. Accordingly, we have removed these gates from the figures to avoid the implication of predefined cluster separations and to better represent the continuity and intrinsic relationships between clusters. We believe this revision allows the data-driven clustering results to be interpreted more objectively and aligns better with standard scRNA-seq visualization practices.

In Figure 3b, 3c (Page 43)

Original:

Revised:

In Figure 5b, c (Page 46)

Original:

Revised:

3. Figure 3F: specifically what do you mean by cumulative frequency at the far right? Cluster 2 shows 100% frequency in both genotype conditions? What does this mean? Also, how did you determine the cutoff for *CXCL9* positivity versus *Spp1* positivity at the gene expression level?

(Author's Response/Action)

We appreciate your thoughtful comment. As noted, the original Figure 3F was based on cumulative cell distributions along a single pseudotime trajectory line. This approach inadvertently resulted in misleading representations—such as cluster 2 appearing to reach 100% frequency in both WT and *Vsir*^{-/-} genotypes—due to the relative nature of cumulative plots. In light of Figure 3c, where the trajectories for WT and *Vsir*^{-/-} are clearly separated, the cumulative frequency plot in **Figure 3f** failed to accurately reflect the actual genotype composition of individual clusters.

To address this issue, we revised **Figure 3f** to display the proportional distribution of WT and *Vsir*^{-/-} cells within each cluster, rather than their cumulative progression along pseudotime. This revised visualization more directly represents the genotype-specific distribution of macrophage clusters and enables clearer interpretation of genotype-driven differences.

In addition, *Cxcl9*⁺ and *Spp1*⁺ cells were defined based on whether the expression level of each gene exceeded its average expression across the entire macrophage population. We chose this gene-specific threshold because global expression levels vary widely across genes, and applying a uniform absolute cutoff could lead to biased or inaccurate classification of gene-positive cells. By using each gene's average expression as a relative cutoff, we aimed to preserve interpretability while accounting for global expression differences across genes.

Finally, to complement the bar graph originally used to illustrate the frequency of *Cxcl9*⁺ and *Spp1*⁺ cells across clusters, we have added a contour plot in Supplementary Figure S9d. This revised figure offers a more intuitive and continuous visualization of the relative expression patterns across clusters, facilitating clearer interpretation of *Cxcl9* and *Spp1* distributions.

In **Figure 3f** (Page 43)

Original:

Revised:

f Group frequencies and cell numbers of macrophage subtypes in WT and *Vsir*^{-/-} tumors. The *Cxcl9*:*Spp1* ratio for each Mono/Mac cluster is calculated, with the highest ratio highlighted in *red* and the lowest in *blue*. Cumulative frequency of cells is compared between WT and *Vsir*^{-/-} tumors.

New Supplementary Figure S9

Supplementary Figure S9. Gene expression profiles from Mono/Macrophage subclusters.

d Density contour plots of macrophage clusters based on *Cxcl9* (*x*-axis) and *Spp1* (*y*-axis). Each plot depicts a two-dimensional kernel density estimation of single macrophages within a cluster. Red dashed lines denote the average expression levels of each gene across all macrophages, defining quadrant thresholds. Cell frequencies within each quadrant are reported based on relative *Cxcl9* and *Spp1* expression.

In **Results** (Page 10, Lines 234-236)

Original:

Further, MΦ_cluster2 had a *Cxcl9*:*Spp1* ratio of 2.51, compared to a ratio of 0.69 in MΦ_cluster1, indicating a higher *Cxcl9*:*Spp1* ratio in *Vsir*^{-/-} TAMs could enhance anti-tumor effect (Fig. 3f).

Revised:

→ “The *Cxcl9:Spp1* ratio was 2.51 in MΦ_cluster2 versus 0.69 in MΦ_cluster1 (Fig. 3f, Supplementary Fig. S9d), indicating a shift toward a more chemotactic, immunostimulatory TAM phenotype in *Vsir*^{-/-} tumors.”

4. Figure 4i: the % positive gate seems to bias the results because it excludes non-proliferating T cells, of which there are more in the WT BMDM group. If you only assess only the proliferating cells in both groups, is there a difference in cytokine production btw groups?

(Author’s Response/Action)

Thank you for pointing this out. We sincerely apologize for this oversight and appreciate your careful review. We acknowledge that including non-proliferating T cells in the calculation of cytotoxic molecule frequencies may have introduced bias, particularly given the higher proportion of non-proliferating cells in WT BMDM group. To address this, we re-gated our flow cytometric data to focus exclusively on proliferating CD8⁺ T cells.

The revised analysis, presented in the update Figure 4f, demonstrates that VISTA KO BMDMs still elicit significantly higher frequencies of IFN- γ , TNF- α , and perforin-producing CD8⁺ T cells compared to WT BMDM + CD8⁺ T cells, even when restricted to proliferating populations. This correction hopefully ensures a more accurate representation of functional differences and confirms that the enhanced cytotoxic activity in VISTA KO BMDM co-cultures is not an artifact of gating strategy. We thank you once again for this critical insight.

In **Figure 4i** (Page 44)

Original(left)/Revised(right):

5. Figure 4H: Is there any difference in costimulatory ligand expression (CD80/CD86, etc.) in BMDMs from WT versus VISTA-KO mice? Or is the difference in cross-presentation capacity solely due to higher levels of pMHC?

(Author's Response/Action)

We thank you for this important mechanistic question. We examined the expression of key costimulatory molecules, including CD80, CD86, and MHC-II (I-A/I-E), on BMDMs from WT and VISTA-KO mice. As shown in Supplementary Figure S13, there was no significant difference in the expression of these surface markers between WT and VISTA-KO BMDMs. These results suggest that the enhanced cross-presentation capacity observed in VISTA-KO macrophages is not due to altered costimulatory ligand expression, but it is likely attributable to increased pMHC-I presentation or antigen-cross processing efficiency. This mechanistic point has been clarified in the revised *Results*.

In **Figure 4h** (Page 44)

Revised:

Supplementary Figure S13. Expression of CD80, CD86, and I-A/I-E in WT versus *Vsir*^{-/-} BMDMs. Flow cytometric analysis of CD80, CD86, and I-A/I-E expression on WT and *Vsir*^{-/-} BMDMs. Quantification is shown as mean fluorescence intensity (MFI) ($n = 11$). Data are mean \pm SEM. Statistical significance was assessed using an unpaired two-tailed Student's *t*-test. *n.s.*, not significant.

In **Results** (Page 12, Lines 276-283)

Original:

“To validate these findings, we performed an antigen cross-presentation assay by measuring SIINFEKL peptide bound to MHC class I (H2-K^b) on WT and *Vsir*^{-/-} BMDMs. After treatment with OVA peptide for 1 h or OVA protein for 24 h, we assessed antigen presentation efficiency. While SIINFEKL-loaded BMDMs showed no difference in H2-K^b SIINFEKL MFI between WT and *Vsir*^{-/-} groups (Supplementary Fig. S8c), *Vsir*^{-/-} BMDMs loaded with OVA protein exhibited significantly higher SIINFEKL peptide bound H2-K^b levels compared to WT BMDMs, especially at higher OVA protein concentrations (Fig. 4g). This suggests an enhanced antigen processing and presentation capacity in the absence of VISTA.”

Revised:

→ “However, *Vsir*^{-/-} BMDMs exposed to full-length OVA protein displayed significantly higher SIINFEKL-H2-K^b levels, especially at higher antigen concentrations (Fig. 4d), indicating enhanced antigen processing and cross-presentation in the absence of VISTA.

Notably, this occurred without changes in CD80, CD86, or I-A/I-E expression (Supplementary Fig. S13), suggesting that VISTA loss selectively enhances antigen-presenting function without broadly activating macrophages.”

6. Figure 6: CXCR3 and CX3CR1 expression are likely insufficient to conclusively infer levels of T cell exhaustion. Could you show a difference in expression of canonical markers such as **PD1/Tim-3, PD-1/CD39, or PD-1/TOX?**

(Author’s Response/Action)

We sincerely appreciate the reviewer’s insightful comment. To more precisely evaluate the exhaustion states of CD8⁺ T cells, we updated Figure 5c to clarify the identities of CD8⁺ T-cell subsets and expanded our analysis to include co-expression of canonical exhaustion marker pairs including *Pdcd1/Havcr2*, *Pdcd1/Entpd1*, *Pdcd1/Tox*, and *Tox/Entpd1*, which have been widely associated with terminal exhaustion (Jin *et al.*, *Proc Natl Acad Sci*, 2010; Canale *et al.*, *Cancer Res*, 2018; Scott *et al.*, *Nature*, 2019). Gene expression patterns revealed that WT-specific cluster 2 and *Vsir*-deficient Cluster 0 exhibited the highest frequencies of co-expressing both markers among the genotype-specific clusters, suggesting that these populations represent exhausted CD8⁺ T cells. Notably, *Tox/Entpd1* expression was highest in Clusters 1 and 2, both enriched in WT CD8⁺ T cells (Figure a below). In contrast, the *Il7r/Cd226* pair showed the highest expression in Cluster 3, which may reflect a less exhausted or progenitor state (Figure b below). These results collectively indicate that Clusters 0, 1, and 2 represent exhausted CD8⁺ T cells, whereas Cluster 3 retains features of an early exhausted phenotype in intratumoral CD8⁺ T cells.

Density contour plots of CD8⁺ T cells using canonical exhaustion and effector-like markers

Each contour plot depicts a two-dimensional kernel density estimation of single CD8⁺ T-cell clusters, based on normalized expression levels of paired exhaustion markers (**a**) and *I17r*/*Cd226* (**b**) derived from CD8⁺ T cells. Red dashed lines represent the average expression level of each gene across all CD8⁺ T cells and serve as quadrant thresholds for calculating the frequency distribution within each cluster.

Reviewer #3's Comments and Responses

1. Deeper Mechanistic Research Needed

•When the study reveals the regulatory effects of *VISTA deficiency on TAM polarization*, it lacks a deeper exploration of *upstream signaling pathways or regulators*. Including more *mechanistic studies* could enhance its scientific impact.

(Author's Response/Action)

We thank you for this valuable comment. In response, we have strengthened the mechanistic aspect of our study in two ways by: (1) expanding the *Discussion* to incorporate established knowledge regarding pathways involved in VISTA regulation, and (2) performing a phospho-protein kinase array in WT and VISTA-deficient macrophages to identify differentially regulated signaling molecules.

Regarding upstream regulators, HIF-1 α has been shown to bind to a conserved hypoxia response element in the *VISTA* promoter, enhancing VISTA expression in myeloid cells and contributing to an immunosuppressive tumor microenvironment (Deng *et al.*, *Cancer Immunol Res*, 2020). While interferons are known to broadly regulate immune checkpoint molecules, direct transcriptional control of *VISTA* by type I/II IFNs remains poorly defined and warrants further investigation.

On the downstream side, recent studies have linked VISTA to ERK and STAT3 signaling cascades in MDSCs (Zhang *et al.*, *Cell Rep*, 2024). This aligns with our observation that VISTA KO TAMs exhibited reduced SPP1 expression (Supplementary Figure S18c), a gene transcriptionally regulated by STAT3 (Yu *et al.*, *Dev Cell*, 2025).

To investigate early signaling dynamics influenced by VISTA, we performed a phospho-proteome array in WT and VISTA KO BMDMs following LPS and IFN- γ stimulation for 15 minutes. This revealed a marked reduction in GSK-3 α/β phosphorylation at serine residues in VISTA KO BMDMs, implicating the GSK-3 pathway in VISTA-mediated macrophage regulation. We also observed decreased phosphorylation of p53 (S15/S45) in VISTA KO BMDMs. As p53 activity has been associated with M2 polarization (Li *et al.*, *Cell Death Differ*, 2014), this result further supports a shift away from an immunosuppressive phenotype.

Additionally, phosphorylation of STAT3 at S727, which contributes to M2-like macrophage function (Zhang *et al.*, *Front Immunol*, 2023), was similarly reduced in VISTA KO BMDMs.

These findings are now contextualized in the revised *Discussion*. We have opted not to include the phospho-proteome array data in the main manuscript, as this analysis remains ongoing. We appreciate your suggestion, which has helped refine and deepen the mechanistic framework of our study. Based on our previous experience studying VISTA in kidney disease model, we plan to pursue these signaling pathways in future mechanistic investigations.

In **Discussion** (Page 18, Line 412)

Original:

“Our study highlights VISTA, a B7 checkpoint molecule, as a pivotal regulator of TAM polarization and the immune landscape in PDAC^{46,47}.”

Revised:

→ “Our study highlights VISTA as a modulator of macrophage polarization and immune dynamics in PDAC, **although further studies are needed to define its mechanistic role** ^{46, 47}.”

•The mechanisms underlying the enhanced CD8⁺ T cell function (e.g., metabolic states or epigenetic regulation) are underexplored. Please adding more data on these aspects would strengthen this part.

(Author’s Response/Action)

Thank you for this comment. To complement our phenotypic analysis of CD8⁺ T cell exhaustion, we examined the frequency of *Eomes*⁺*Cd38*⁺ double-positive cells across clusters. This subset, which has been previously associated with terminal exhaustion under chronic stimulation (Philip *et al.*, *Nature*, 2017, Chen *et al.*, *Sci Adv*, 2022), was most abundant in WT-specific cluster 2 (44.0%) and moderately enriched in cluster 1 (22.6%), but substantially reduced in *Vsir*-deficient clusters 0 (17.6%) and 3 (5.0%) (**Figure 5f; revised to address Reviewer 1’s comment**). These data suggest that VISTA deficiency may attenuate terminal differentiation and promote less exhausted states among CD8⁺ TILs.

In **Figure 5f** (Page 46)

Original (left) /Revised (right):

To further investigate the underlying mechanisms, we evaluated module scores for key metabolic pathways, including glycolysis, OXPHOS, FAO, mTORC1, MYC, and hypoxia, using curated gene sets from the MSigDB Hallmark collection (Liberzon *et al.*, *Cell Syst*, 2015). To focus on biologically meaningful signals, we applied a visualization threshold (col.min = 0.5). WT-specific cluster 2, which harbored the highest proportion of *Eomes*⁺*Cd38*⁺ cells, exhibited strong enrichment of glycolysis, MYC, and hypoxia—consistent with terminal exhaustion under metabolic stress. Cluster 1 showed high mTORC1 activity, supporting an effector-like state. In contrast, *Vsir*-deficient cluster 0 showed relatively low scores except for FAO, suggestive of a memory-like metabolic profile. Cluster 3 displayed elevated OXPHOS and mTORC1 despite low FAO, indicative of a metabolically active yet less terminally exhausted population.

Metabolic module scores across intratumoral CD8⁺ T-cell clusters

Dot plot showing average expression and percent-expressing cells for key metabolic modules—including mTORC1, MYC, hypoxia, glycolysis, FAO, and OXPHOS—across four CD8⁺ T-cell clusters in the tumor microenvironment.

We agree that validating these metabolic profiles through functional assays and complementary epigenetic interrogation would provide a deeper mechanistic insight. However, we believe these experiments are beyond the scope of the current study. We appreciate your comment and hope to pursue them in a future project focused specifically on delineating the metabolic and epigenetic regulation of CD8⁺ T cell differentiation in the context of VISTA signaling.

2.Optimizing Data Integration and Presentation

•The article relies heavily on supplementary materials (e.g., many critical findings are presented in supplementary figures). This might hinder reviewers' ability to grasp the study's significance quickly.

(Author's Response/Action)

We acknowledge this concern and have restructured the manuscript to relocate key supplementary figures to the main figure panels. This adjustment ensures that critical findings are more prominently presented and improves the overall readability and accessibility of the manuscript.

In Figure 2 (Page 41)

Original:

Revised:

In Figure 4 (Page 44)

Original:

Revised:

In Figure 5 (Page 46)

Original:

Revised:

In Figure 6 (Page 48)

Original:

Revised:

•Although the single-cell RNA sequencing analysis is comprehensive, the data could be better integrated to simplify conclusions and improve readability.

(Author's Response/Action)

We have streamlined the single-cell RNA sequencing analysis by consolidating key findings into simplified visualizations (e.g., integrated UMAPs and pathway enrichment summaries). We hope these revisions enhance data integration and make the conclusions more accessible to the readers by removing:

- (1) artificial gating in scRNA-seq data in Figure 3 and 5, and
- (2) Figure 5a, as the text already includes information on CD8⁺ T-cell subsets analyzed in the same manner as macrophages.

In **Figure 3b, 3c** (Page 43)

Original:

Revised:

In Figure 5 (Page 46)

Original:

Revised:

3. Strengthening the Discussion on Clinical Relevance

•The discussion on the **clinical feasibility of VISTA-targeted therapy** is brief. For example, more information on the **development and challenges of existing VISTA inhibitors** would be valuable. High-impact journals often emphasize **translational relevance**.

(Author's Response/Action)

We appreciate this insightful suggestion. In the revised *Discussion*, we have expanded upon the current challenges associated with VISTA inhibitors in clinical development, including

toxicity, limited tumor specificity, and suboptimal therapeutic efficacy. We also emphasized the rationale for renewed focus on VISTA-targeted therapies and discussed potential strategies to address these limitations. These additions were made to better underscore the translational relevance of our findings.

In **Discussion** (Page 20, Lines 458- 464)

Original:

→ “The findings from our study provide a strong rationale for targeting VISTA in PDAC. Inhibiting VISTA has the potential to reprogram TAMs to adopt a pro-inflammatory, antitumor phenotype while enhancing recruitment and activation of stem-like CD8⁺ T cells. Such an approach could synergize with existing immune checkpoint inhibitors to improve their efficacy. By simultaneously reprogramming TAMs and augmenting T cell function, VISTA inhibition represents a promising strategy to overcome the immunosuppressive environment of PDAC and improve patient outcomes^{63, 64}.”

Revised:

→ “ICB therapies targeting PD-1 and CTLA-4 have demonstrated limited efficacy in PDAC, underscoring the need for alternative strategies that address multiple layers of immune suppression. In this context, VISTA inhibition offers a promising therapeutic avenue. Notably, anti-VISTA monotherapy combined with gemcitabine elicited a synergistic anti-tumor response, characterized by an increased CXCL9/SPP1 ratio in TAMs and enhanced infiltration of CXCR3⁺ CD8⁺ T cells. This additive effect is particularly relevant for immune-excluded tumors like PDAC, where conventional ICB therapies have failed to yield substantial clinical benefit. Moreover, considering the well-documented toxicity profile of gemcitabine, our findings suggest that VISTA blockade could enhance therapeutic efficacy while potentially reducing dependence on cytotoxic agents.”

4. Addressing the Lack of Comparative Analysis

• *The study does not compare VISTA’s function and utility with other immune checkpoint molecules (e.g., PD-1, TIM-3). Such comparisons could better highlight VISTA’s uniqueness.*

(Author’s Response/Action)

Thank you for this insightful comment. Rather than positioning VISTA in direct comparison with other immune checkpoint molecules such as PD-1 or TIM-3, we believe its therapeutic value lies in its complementary mechanism of action, which offers potential for combination strategies with existing immune checkpoint blockade therapies.

VISTA is predominantly expressed on myeloid cells, including TAMs, as shown below, and functions early in the immune response by modulating antigen presentation and naïve T cell priming. In contrast, PD-1 and TIM-3 are mainly expressed on T cells and regulate effector functions at later stages (Kumagai *et al.*, *Nat Immunol*, 2020). These non-overlapping roles suggest that VISTA blockade could enhance or restore anti-tumor immunity in settings where traditional ICB shows limited efficacy, particularly in “cold” or myeloid-rich tumors like pancreatic cancer.

We have now added an explanatory statement in the *Discussion* to clarify that macrophages exhibit high levels of VISTA expression.

In **Discussion** (Page 18, Lines 418-420)

Original:

“Cxcl9/10-engineered DCs have previously been shown to suppress tumor growth in NSCLC murine models, emphasizing a crucial role of Cxcl9 in anti-tumor immune responses ⁴⁸.”

Revised:

“Cxcl9/10-engineered DCs have previously been shown to suppress tumor growth in NSCLC murine models, emphasizing a crucial role of Cxcl9 in anti-tumor immune responses ⁴⁹. In PDAC, however, VISTA’s predominant expression in TAMs creates distinct microenvironment dynamics. Our murine model revealed VISTA deficiency drives TAM polarization towards a *Cxcl9*⁺ state while lowering *Spp1*⁺ state.”

Responses to Reviewer's Comments

We are deeply grateful for your thoughtful comments and invaluable suggestions regarding our manuscript. Your feedback has provided us with an excellent opportunity to improve the quality and clarity of our work. We have carefully revised the manuscript in accordance with the reviewers' recommendations and your editorial guidance. Below, we respectfully addressed each of the reviewers' comments point-by-point and described the corresponding revisions made to the manuscript. All changes are clearly **marked in red** for your convenience. Thank you once again for your time and consideration.

Reviewer #1 (Remarks to the Author):

The authors have significantly improved the manuscript by addressing previous critiques and enhancing both translational and mechanistic depth. The inclusion of human pancreatic ductal adenocarcinoma (PDAC) tissue and single cell RNA sequencing data from the HTAN cohort connects findings from mouse models with human data. This demonstrates a consistent relationship between VSIR expression, the phenotype of tumor-associated macrophages (TAMs), and the functionality of CD8⁺ T cells. The addition of rigorous multivariate analyses, single-cell stratification, and other co-expression profiling provides strong support for the manuscript's central hypothesis: that VISTA orchestrates immunosuppression in PDAC by modulating both the TAM and T-cell compartments.

Finally, expanded the discussion of upstream and downstream VISTA signaling, and strengthened the experimental data by including studies on CSF1R inhibition and combination therapy with gemcitabine. The revised manuscript is comprehensive, data-driven, and well-integrated across both murine and human systems.

(Author's Response)

We greatly appreciate your recognition of the improvements to our manuscript. Your constructive feedback in the previous revision, particularly the suggestion to integrate human PDAC tissue data with single-cell RNA sequencing to bridge mouse and human findings, was invaluable in enhancing the clarity and impact of our study.

Reviewer #2 (Remarks to the Author):

Thank you for your detailed responses to my concerns, which have been sufficiently addressed.

(Author's Response/Action)

We sincerely appreciate your thoughtful concerns and are pleased that our revisions have sufficiently addressed your concerns. We are confident that the revised manuscript has been substantially improved.

Reviewer #3 (Remarks to the Author):

The revised manuscript by authors investigates the immunomodulatory role of VISTA in pancreatic ductal adenocarcinoma (PDAC), with a particular focus on tumor-associated macrophage (TAM) polarization and the downstream effects on CD8⁺ T cell function. The study addresses an important and timely topic, given the poor responsiveness of PDAC to current immunotherapies and the increasing interest in checkpoint blockade.

Major Concerns

1. Mechanistic Evidence for VISTA Signaling Remains Incomplete:

Although the authors state that phospho-protein array experiments were performed to explore downstream signaling pathways, they have not presented any of these data. As a result, the claim that VISTA regulates TAM polarization through specific pathways remains speculative.

(Author's Response/Action)

Thank you for this comment. To address your concern, we have carefully clarified the Discussion and toned down statements that could be interpreted as overstatements, ensuring that all conclusions are strictly supported by the data. Specifically, we now present a complete and coherent chain of evidence that links the *in vivo* VISTA KO phenotype to macrophage state changes and a functionally required IFN- γ -CXCL9-CXCR3 axis, and we also provide the phospho-protein array dataset that was previously referenced but not shown.

We first established the *in vivo* phenotype that motivated the mechanistic analyses. Compared with WT, VISTA KO tumors grew more slowly and exhibited a distinct immune composition by flow cytometry, including increased macrophage and CD8⁺ T cell infiltration (**Fig. 1**). To identify the cellular programs and interactions underlying these differences, we performed tumor scRNA-seq. This analysis highlighted macrophage and CD8⁺ T cells as key cellular compartments linked to the VISTA KO immune contexture, and it nominated the CXCL9-CXCR3 ligand-receptor pair as a prominent interaction enriched in the VISTA KO

setting (**Fig. 6a, 6b**). We validated this single-cell inference by flow cytometry, confirming that CXCR3⁺ CD8⁺ T cells were increased in VISTA KO tumors relative to WT (**Fig. 5f**).

In parallel, scRNA-seq-based pathway analyses of macrophage subsets provided a mechanistic direction upstream of the CXCL9 program. When we compared WT and VISTA KO macrophage subsets, gene set enrichment analyses indicated that antigen-presentation related pathways were more prominent in VISTA KO macrophages (**Fig. 4c**), consistent with enhanced T cell stimulatory capacity. We tested this in a controlled antigen-specific system by co-culturing WT or VISTA KO BMDMs loaded with OVA peptide or protein, with OT-I CD8⁺ T cells. OT-I T cells produced higher IFN- γ when cultured with VISTA KO BMDMs than with WT BMDMs (**Fig. 4f**), establishing that VISTA status in macrophages affects the magnitude of antigen-driven IFN- γ producing T cell responses. In the same co-culture setting, CXCL9 output was also higher in VISTA KO BMDM–OT-I co-cultures than in WT co-cultures (**New Supplementary Fig. S15**), aligning the enhanced IFN- γ response with a CXCL9-high macrophage output state.

New Supplementary Figure S15

Gated on F4/80⁺CD11b⁺ Macrophages

Flow cytometric analysis of CXCL9⁺SPP1⁺ expression in WT and *Vsir*^{-/-} BMDMs co-cultured with OT-I CD8⁺ T cells at a 1:1 ratio. Quantification is shown as the percentage of CXCL9⁺SPP1⁺ cells. Data are pooled from two independent experiments.

We then directly tested whether this IFN- γ -CXCL9-CXCR3 axis is required for the tumor control phenotype associated with VISTA deficiency. In longitudinal antibody blockade experiments, neutralization of IFN- γ and blockade of CXCR3 each attenuated the tumor-growth phenotype observed in the VISTA KO setting (**New Figure 6c**). These *in vivo* perturbation experiments functionally validate that the IFN- γ -CXCL9-CXCR3 axis is necessary for the VISTA KO associated anti-tumor effect, consistent with the interaction signal inferred from scRNA-seq and the CXCR3⁺ CD8⁺ T-cell increase confirmed by flow cytometry.

New Figure 6

c Longitudinal tumor growth curves of tumor burden following anti-IFN- γ antibody (left) or anti-CXCR3 (right) treatment in WT and VISTA KO tumor-bearing mice.

To determine whether macrophage-intrinsic differences contribute to this phenotype beyond differences in T cell-derived IFN- γ , we performed macrophage only stimulation experiments. WT and VISTA KO BMDMs were stimulated with identical concentrations of LPS and IFN- γ across a dose range, and CXCL9 secretion was quantified by ELISA. Under these matched macrophage-only stimulation conditions, VISTA KO macrophages secreted higher CXCL9 in

a dose-responsive manner (**Figure 1 for Review**), demonstrating increased CXCL9 output capacity in the absence of VISTA even without T cell co-culture.

Figure 1 for Review. CXCL9 production by WT and VISTA KO BMDMs after 24 h of LPS and IFN- γ and LPS stimulation.

Finally, the phospho-protein array data referenced previously are now presented in full for transparency, including complete membranes, densitometric quantification, and comprehensive target-level tables. These membranes are provided as **Source Data/Figure 2** and **Figure 3 for Review**, and the bar graphs as **New Supplementary Fig. S12**, together with **New Supplementary Table S3** and **S4**. This unbiased screen identifies phosphorylation features associated with VISTA status under defined stimulation conditions, including differences at inhibitory phosphorylation of GSK3 α/β Ser21/Ser9 (**Figure 4 for Review**), and we include the accompanying inhibitor/perturbation datasets as additional supporting material in the Supplementary section (**New Supplementary Fig. S12a**).

Figure 2 for Review. Phospho-kinase array analysis showing differential phosphorylation patterns between experimental conditions in WT and *Vsir* KO BMDM groups.

New Supplementary Figures S12

a. Integrated pixel density of phosphoproteins in WT and VISTA KO BMDMs following LPS and IFN- γ stimulation. Quantification is shown alongside phospho-protein array membranes.

LPS+IFN- γ	WT		Vsir KO	
	Integrated pixel density (x1000)			
CREB (S133)	37.747	30.245	52.046	49.6
EGFR (Y10B6)	15.919	15.802	13.796	13.036
eNOS (S1177)	44.735	48.471	32.583	33.073
ERK1/2 (T202/Y204, T185/Y187)	41.247	41.886	32.467	32.024
Chk-2 (T68)	47.131	47.326	42.8	43.779
c-Jun (S63)	27.58	23.61	4.881	14.688
Fgr (Y412)	40.942	41.72	32.397	32.197
GSK-3 α/β (S21/S9)	73	71.855	27.38	27.795
GSK-3 β (S9)	85.278	84.551	53.053	49.035
HSP27 (S78/S82)	2.613	1.86	1.997	1.442
p53 (S15)	18.663	19.937	8.995	9.145
p53 (S46)	24.422	21.953	8.765	8.506
p53 (S392)	12.727	13.794	9.829	9.774
JNK 1/2/3 (T183/Y185, T221/Y223)	36.533	39.882	32.971	34.231
Lck (Y394)	11.313	10.261	9.876	9.752
Lyn (Y397)	33.393	35.721	27.944	34.68
MSK1/2 (S376/S360)	44.544	40.952	37.708	36.402
p70 S6 Kinase (T389)	18.496	19.772	10.084	10.317
p70 S6 Kinase (T421/S424)	8.59	7.823	3.274	2.904
PRAS30 (T246)	36.25	36.494	26.391	28.537
p38 α (T180/Y182)	32.079	34.436	28.833	28.398
PDGF R β (Y751)	43.253	35.853	38.242	39.227
PLC- γ 1 (Y751)	44.833	44.833	41.323	45.175
Src (Y419)	72.343	69.933	79.469	78.581
PYK2 (Y402)	45.148	46.2	24.768	24.859
RSK1/2 (S221/S227)	53.147	51.238	30.093	27.993
RSK1/2/3 (S380/S386/S377)	35.474	31.709	27.845	26.945
STAT2 (pY690)	46.826	44.661	40.411	38.97
STAT5a/b (Y694/Y699)	63.299	62.404	50.892	53.375
WNK1 (T60)	91.902	93.885	103.311	97.51
Yes (Y426)	42.941	37.474	38.968	29.426
STAT1 (Y701)	21.731	22.429	9.551	9.577
STAT3 (Y705)	15.467	14.417	5.882	7.373
STAT3 (S727)	163.35	149.139	114.539	123.152
β -catenin	54.361	39.305	54.789	47.764
STAT36 (Y641)	9.576	9.856	6.793	6.513
HSP60	8.986	8.985	6.615	6.89

New Supplementary Table S3 Quantification of phospho-protein array results in WT and *Vsir* KO BMDMs under LPS and IFN- γ stimulation.

Figure 3 for Review. Phospho-kinase array analysis showing differential phosphorylation patterns between experimental conditions in WT and *Vsir* KO BMDM groups.

Gated on F4/80⁺CD11b⁺Ly6C⁺ cells

Figure 4 for Review. Flow cytometric analysis (left) and bar plot (right) of CD86⁺iNOS⁺ cells gated on F4/80⁺CD11b⁺ Ly6C⁺ macrophages in LPS and IFN- γ treated BMDMs. TWS119 was treated 1 h prior to LPS+IFN- γ stimulation.

New Supplementary Figure S12

b. Integrated pixel density of phosphoproteins in WT and VISTA KO BMDMs following TGF- β 1 and IL-10 stimulation.

TGF- β 1+IL-10	WT		Vsir KO	
	Integrated pixel density			
Chk-2 (T68)	13,593	14,372	14,512	14,065
c-Jun (S63)	7,901	8,067	6,771	7,323
Fgr (Y412)	10,487	9,781	19,488	16,305
GSK-3 α/β (S21/S9)	42,564	43,982	50,767	51,219
GSK-3 β (S9)	30,307	32,938	30,217	30,662
HSP27 (S78/S82)	3,952	3,215	3,957	4,214
p53 (S15)	7,516	7,772	7,413	7,211
p53 (S46)	10,599	11,403	9,775	10,650
p53 (S392)	10,448	11,240	9,490	9,816
JNK 1/2/3 (T183/Y185, T221/Y223)	11,109	12,048	23,483	20,880
Lck (Y394)	9,178	10,342	14,076	14,022
Lyn (Y397)	18,419	17,859	27,006	26,887
MSK1/2 (S376/S360)	11,760	12,888	14,200	15,575
p70 S6 Kinase (T389)	8,425	8,420	8,718	8,573
p70 S6 Kinase (T421/S424)	5,456	6,257	5,359	5,661
PRAS30 (T246)	14,699	15,629	13,602	13,903
p38 α (T180/Y182)	10,648	10,512	19,728	18,015
PDGF R β (Y751)	16,650	16,995	24,575	23,813
PLC- γ 1 (Y751)	11,384	12,092	16,866	17,791
Src (Y419)	21,501	23,207	28,560	28,186
PYK2 (Y402)	16,957	17,753	14,234	14,208
RSK1/2 (S221/S227)	10,722	11,428	10,420	10,493
RSK1/2/3 (S380/S386/S377)	11,319	11,116	10,332	10,983
STAT2 (pY690)	12,170	11,544	17,534	17,057
STAT5a/b (Y694/Y699)	25,163	25,454	35,065	38,943
WNK1 (T60)	15,655	15,001	21,174	21,461
Yes (Y426)	12,202	13,102	17,258	17,340
STAT1 (Y701)	6,778	6,616	5,697	5,553
STAT3 (Y705)	6,987	7,594	8,037	7,688
STAT3 (S727)	29,383	29,320	27,556	28,579
β -catenin	25,141	26,375	36,591	37,448
STAT36 (Y641)	6,848	7,148	7,352	7,529
HSP60	6,612	6,447	6,904	6,879

Supplementary Table S4 Quantification of phospho-protein array results in WT and *Vsir* KO BMDMs under TGF- β 1 and IL-10 stimulation.

Overall, the revised submission now provides direct, multi-level mechanistic support, spanning *in vivo* phenotype, single-cell interaction inference with orthogonal flow validation, antigen-specific macrophage-T cell functional assays, *in vivo* necessity tests by blockade, and

macrophage-only intrinsic chemokine output, together with the complete phospho-protein array dataset that was previously referenced but not shown. We have refined the manuscript to improve clarity and ensure closer alignment between the data and the conclusions, as detailed below.

In **Abstract** (Page 3, Lines 7-10)

Original:

“Mechanistically, VISTA deficiency is linked to a shift in tumor-associated macrophages (TAMs) from an immunosuppressive phenotype marked by secreted phosphoprotein 1 (SPP1) to one enriched for C-X-C motif chemokine ligand 9 (CXCL9), indicative of a pro-inflammatory state.”

Revised:

→ “**Functionally**, VISTA deficiency is linked to a shift in tumor-associated macrophages (TAMs) from an immunosuppressive phenotype marked by secreted phosphoprotein 1 (SPP1) to one enriched for C-X-C motif chemokine ligand 9 (CXCL9), indicative of a pro-inflammatory state.”

In **Results** (Page 8, Line 118-122)

Original:

“Collectively, these findings identify VISTA as a central regulator of immune suppression in PDAC, orchestrating TAM-mediated immunosuppression and dampening CD8⁺ T cell responses. Targeting VISTA represents a promising immunotherapeutic strategy to overcome the immunosuppressive TME and enhance anti-tumor immunity in PDAC.”

Revised:

→ “Collectively, these findings **position** VISTA as a **key immunoregulatory checkpoint associated with immunosuppressive macrophage states and impaired CD8⁺ T cell function** in PDAC. Targeting VISTA represents a promising immunotherapeutic strategy to overcome the

immunosuppressive TME and enhance anti-tumor immunity in PDAC.”

In **Results** (Page 11-12, Line 208-212)

Original:

“These findings suggest that VISTA shapes the macrophage landscape to favor immune suppression, and that its inhibition reprograms TAMs toward a phenotype that promotes anti-tumor immunity.

Revised:

→ “These findings indicate that *VISTA* expression in human PDAC is associated with macrophage states enriched for immunosuppressive gene programs, consistent with the immune regulatory patterns observed in murine models.”

In **Results** (Page 12, Lines 217-225)

Original:

“To assess the functional impact of VISTA deficiency on macrophages, we analyzed several characteristics. While WT and *Vista*^{-/-} BMDMs showed comparable phagocytic ability (Supplementary Fig. S11a), transcriptomic and pathway analyses revealed notable changes in immunomodulatory capacity. Principal Component Analysis (PCA) highlighted distinct transcriptional profiles among MΦ_clusters 6, 0, and 4, while MΦ_clusters 1, 2 and 3 were closely related, with MΦ_clusters 2 and 3 particularly similar (Supplementary Fig. S12a). The largest variance (PC1) separated Mono4 and MΦ_cluster6 from other clusters. Given the distinct profiles between MΦ_clusters 2 and 3 in *Vista*^{-/-} tumor *versus* WT control, we focused on their functional characteristics and spatial proximity to CD8⁺ T cells.”

Revised:

→ “While WT and *Vista*^{-/-} BMDMs showed comparable phagocytic ability (Supplementary Fig. S11a), transcriptomic and pathway analyses revealed notable changes in immunomodulatory capacity. Consistent with these transcriptional differences, phospho-protein profiling of WT

and *Vsir*^{-/-} BMDMs under matched stimulation conditions revealed distinct phosphorylation patterns across multiple signaling nodes (Supplementary Fig. S), indicating altered intracellular signaling states in the absence of VISTA. Principal Component Analysis (PCA) highlighted distinct transcriptional profiles among MΦ_clusters 6, 0, and 4, while MΦ_clusters 1, 2, and 3 were closely related, with MΦ_clusters 2 and 3 particularly similar (Supplementary Fig. S12a).”

In **Results** (Page 14, Lines 258-261)

Original:

“These co-cultures also showed higher IFN- γ , TNF- α , and perforin production in OT-I T cells (Fig. 4f). In contrast, SIINFEKL peptide stimulation produced similar CD8⁺ T cell proliferation regardless of macrophage genotype (Fig. 4e, bottom), ...”

Revised:

→ “...(Fig. 4f). Notably, enhanced IFN- γ production in the *Vsir*^{-/-} BMDM co-cultures was accompanied by increased CXCL9 output (Supplementary Fig. S15), consistent with activation of an IFN- γ -CXCL9 chemokine axis in this setting. In contrast, SIINFEKL peptide stimulation produced comparable CD8⁺ T cell proliferation regardless of macrophage genotype (Fig. 4e, bottom), ...”

In **Discussion** (Page 21, Lines 434-438)

Original:

“Our study highlights VISTA as a modulator of macrophage polarization and immune dynamics in PDAC, although further studies are needed to define its mechanistic role^{47, 48}.”

Revised:

“Our study highlights a close association between VISTA expression and macrophage state composition in PDAC; how this relationship is established within the tumor microenvironment warrants further investigation.”

In **Discussion** (Page 23, Lines 479-483)

Original:

“...Our single-cell analysis of the PDAC TME suggests that modulating the *Cxcl9:Spp1* ratio via VISTA could enhance anti-tumor responses by shifting the balance toward pro-inflammatory immune responses.”

Revised:

→ “...Our single-cell analysis of the PDAC TME suggests that modulating the *Cxcl9:Spp1* ratio via VISTA could enhance anti-tumor responses by shifting the balance toward pro-inflammatory immune responses. **The precise mechanism by which VISTA drives this shift in macrophages remains to be determined.**”

2. Lack of Functional Validation for CXCL9/CXCR3 Axis:

While the manuscript shows increased CXCL9 expression and CXCR3⁺ CD8⁺ T cell infiltration in VISTA-deficient tumors, it does not directly test whether this axis is necessary for the anti-tumor effects. Blocking experiments would be critical to establish causality. If not feasible, this limitation should be explicitly stated in the Discussion.

(Author’s Response/Action)

Thank you for this important comment regarding functional validation of the CXCL9/CXCR3 axis. In response, we performed *in vivo* antibody blockade experiments and expanded our presentation to include longitudinal tumor growth trajectories, thereby directly testing whether this axis is required for the VISTA KO associated tumor-control phenotype.

We treated orthotopic KPC001 tumor bearing mice with either anti-IFN- γ or anti-CXCR3 antibodies (200 μ g, i.p.) using a tumor burden matched initiation design. Specifically, the first dose was administered when tumors reached approximately 50–80 mm³, followed by dosing every three days until day 20, and tumor volume was monitored serially over time. The complete longitudinal growth curves are provided (**New Fig. 6c**). Under these conditions, IFN- γ neutralization markedly attenuated the tumor control phenotype observed in the VISTA KO setting, and CXCR3 blockade similarly reduced the phenotype. These results provide direct functional evidence that the IFN- γ –CXCL9–CXCR3 axis is required *in vivo* for the anti-tumor effect associated with VISTA deficiency.

New Figure 6

c Longitudinal tumor growth curves of tumor burden following anti-IFN- γ antibody (left) or anti-CXCR3 (right) treatment in WT and VISTA KO tumor-bearing mice.

To evaluate reproducibility across orthotopic PDAC models, we performed additional CXCR3 blockade experiments in a second orthotopic model (Pan02). In this independent system, anti-CXCR3 treatment again resulted in reduced intratumoral CD8⁺ T cell accumulation (normalized to tumor volume) in VISTA KO mice (**New Supplementary Fig. S21a and b**), consistent with CXCR3 blockade impairing CD8⁺ T-cell recruitment/retention in the tumor microenvironment. Together, these data demonstrate that CXCR3-dependent CD8⁺ T-cell recruitment is reproducibly linked to the VISTA KO anti-tumor phenotype across two orthotopic PDAC models.

New Supplementary Figure 21

a Tumor volume of WT and *Vsir* KO mice bearing Pan02 tumors on day 20 and 27 with or without anti-CXCR3 antibody administration. **b** Flow cytometric quantification of CD8 T cells normalized to tumor volume ($n = 4$ per group) at day 21 following anti-CXCR3 antibody administration.

Taken together, these blockade experiments, supported by longitudinal growth analysis and replication in an independent model, provide direct functional validation that the IFN- γ -CXCL9-CXCR3 axis is necessary for the VISTA KO associated anti-tumor immune phenotype. We have revised the Results section accordingly to reflect this strengthened causal evidence.

In **Results** (Page 18, Lines 358-370)

Original:

“This interaction suggests that M Φ _cluster2 macrophages in *Vsir*^{-/-} tumor create an inflammatory environment, enhancing CD8⁺ T cell recruitment and activation, which strengthens the anti-tumor response. Specifically, the *Cxcl9-Cxcr3* axis facilitates CD8⁺ T cell infiltration and activation, supporting efficient tumor clearance³⁴. Both WT and *Vsir*^{-/-} tumors exhibited *Cxcl16-Cxcr6* and *Ccl8-Ccr5* interactions, supporting tissue repair and immune regulation, respectively^{35, 36}.”

Revised:

→ “This interaction suggests that M Φ _cluster2 macrophages in *Vsir*^{-/-} tumor create an

inflammatory environment, enhancing CD8⁺ T cell recruitment and activation, which strengthens the anti-tumor response. Specifically, the *Cxcl9-Cxcr3* axis mediates CD8⁺ T cell infiltration and activation, facilitating efficient tumor clearance³⁴.

We evaluated whether the anti-tumor effect of *Vsir* deletion was mediated by IFN- γ –CXCL9–CXCR3 axis by administering a neutralizing anti-IFN- γ antibody or anti-CXCR3 antibody to VISTA-KO mice. Anti-IFN- γ treatment abolished the reduced tumor growth seen in *Vsir*^{-/-} mice. Likewise, blockade of CXCR3 also reversed the enhanced tumor control in *Vsir*^{-/-} hosts. These results indicate that the protective effect of VISTA loss depends on the IFN- γ –CXCL9–CXCR3 pathway (Fig. 6c, Supplementary Fig. S21). In addition, both WT and *Vsir*^{-/-} tumors exhibited *Cxcl16-Cxcr6* and *Ccl8-Ccr5* interactions, supporting tissue repair and immune regulation, respectively (Fig. 6b)^{35, 36}.”

In **Methods** (Page 25-26, Lines 533-536)

“For IFN- γ and CXCR3 neutralization, 200 μ g of anti-IFN- γ (Clone XMG1.2, *InVivoMab*, BioXCell) or anti-CXCR3 neutralizing antibody (Clone CXCR3-173, *InVivoMab*, BioXCell) were administered intraperitoneally 5 days post KPC001 cell inoculation every 3 days until tumor harvest.”

3. CD8⁺ T Cell Exhaustion Analysis Requires Further Support:

The conclusions regarding reduced T cell exhaustion rely mainly on module scoring from scRNA-seq. Without additional validation—such as flow cytometric co-expression of exhaustion markers or functional assays. The current interpretation may overstate the data. The language should be moderated to reflect the correlative nature of the evidence.

(Author’s Response/Action)

We appreciate this important point. We agree that our conclusion regarding reduced CD8 T cell exhaustion relied heavily on scRNA-seq module scoring and could be interpreted as overstating the evidence. To address this concern, we now added directed flow-cytometric validation in tumor-infiltrating CD8 T cells and revised the manuscript wording to reflect the correlative nature of the data.

Specifically, we performed multiparameter flow cytometry to quantify co-expression of canonical exhaustion-associated inhibitory receptors (PD-1, TIM-3, LAG-3, and TIGIT) on

intratumoral CD8 T cells from WT and VISTA KO tumors (**New Supplementary Figure S17d**). In this analysis, the frequency of TIM-3⁺PD-1⁺ CD8 T cells alone did not differ substantially between WT and VISTA KO tumors (**Figure 5 for Review**). We explicitly report this result because it underscores that single-marker or two-marker comparisons may not capture differences along the exhaustion continuum, particularly when both groups contain activated/exhaustion-prone T cells in an inflamed tumor microenvironment.

Figure 5 for Review. Flow cytometric analysis of TIM⁺PD-1⁺ (top) and LAG3⁺PD-1⁺ (bottom) expression gated on CD8⁺ T cells from WT and *Vsir* KO samples.

Motivated by our scRNA-seq observation that multiple exhaustion-associated transcripts, including markers linked to more advanced or terminal exhaustion, were relatively enriched in WT tumors, we therefore stratified CD8 T cells along the exhaustion spectrum using a cumulative marker-burden approach. Rather than interpreting exhaustion from any single receptor, we quantified the fraction of CD8 T cells expressing increasing numbers of inhibitor receptors (0, 1, 2, and ≥ 3 markers; **New Supplementary Figure S17d**), which is conceptually consistent with the view that more advanced exhaustion states are characterized by broader and

more stable expression of multiple inhibitor receptors and reduced effector capacity (Wherry and Kurachi, Nat Rev Immunol, 2015; McLane *et al.*, Nat Immunol, 2019).

New Supplementary Figure S17

d Frequency distribution of intratumoral CD8⁺ T cells expressing 0, 1, 2, or ≥ 3 exhaustion markers (PD-1, TIM-3, LAG-3, and TIGIT) in KPC001 tumors.

Using this approach, we observed that WT tumors contained a higher proportion of CD8 T cells with high cumulative exhaustion marker co-expression (cells expressing ≥ 3 inhibitory receptors), compared with VISTA KO tumors at day 32 post-KPC001 injection (WT vs KO: 1.9 fold difference, **New Supplementary Figure S17d**). We interpret this as supportive evidence that, even if PD-1/TIM-3 alone does not differ, WT tumors may harbor a larger fraction of CD8 T cells occupying a more advanced or terminally exhausted state, considering the overall inhibitory receptor burden. Importantly, we have revised the Results text to avoid claiming definitive “reduced exhaustion” and instead describe this finding as evidence compatible with a shift in exhaustion-state distribution.

Consistent with this interpretation, our intracellular cytokine staining data (Supplementary Figure S6b) show that IFN- γ ⁺TNF- α ⁺ CD8 T cells were more abundant in VISTA KO tumors than in WT tumors. Because more exhausted CD8 T cells typically exhibit impaired polyfunctional cytokine production and reduced effector programs (Ahmed *et al.*, Immunity, 2007), this functional readout is compatible with (but does not on its own prove) a relative enrichment of better preserved effector-like CD8 T cells in the absence of VISTA.

Finally, in line with your recommendation, we have now moderated the language through the manuscript to emphasize that our exhaustion related conclusions are supported by a combination of (i) scRNA-seq signatures and (ii) flow-cytometric marker burden patterns and cytokine production, and should be interpreted as correlative rather than definitive mechanistic proof of altered exhaustion programming.

In **Abstract** (Page 3, Lines 10-13)

Original:

“This shift was accompanied by enhanced recruitment of CXCR3⁺ CD8⁺ T cells, which showed reduced exhaustion-and sustained cytotoxic potential.”

Revised:

→ “This shift was accompanied by enhanced recruitment of CXCR3⁺ CD8⁺ T cells, which showed a lower prevalence of terminal exhaustion-like CD8 T cell states and sustained cytotoxic potential.”

In **Results** (Pages 15-16, Lines 300-309)

Original:

“CD8⁺ T_clusters 0 and 3 were predominantly derived from *Vsir*^{-/-} tumor, whereas clusters 1 and 2 were enriched in CD8⁺ T cells from WT tumor (Fig. 5c and d). To further characterize their functional states, we quantified exhaustion and effector phenotypes using established gene signatures. CD8⁺ T_clusters 1 and 2 contained higher proportions of *Cd38*⁺*Eomes*⁺ cells (22.5% and 44.0%) and *Cd226*⁻*Pdcd1*⁺ cells³⁰ (41.0% and 44.3%), respectively, compared to CD8⁺ T_clusters 0 and 3 (*Cd38*⁺*Eomes*⁺: 14.0% and 5.0%; *Cd226*⁻*Pdcd1*⁺: 31.5% and 25.3%) (Fig. 5e, Supplementary Fig. S15a and b). These data indicate that *Vsir*^{-/-} tumors harbor CD8⁺ T cells with reduced exhaustion and enhanced cytotoxic potential compared to their WT counterparts.”

Revised:

→ “...CD8⁺ T_clusters 1 and 2 contained higher proportions of *Cd38*⁺*Eomes*⁺ cells (22.5% and 44.0%) and *Cd226*⁻*Pdcd1*⁺ cells³⁰ (41.0% and 44.3%), respectively, compared to CD8⁺

T_clusters 0 and 3 ($Cd38^+Eomes^+$: 14.0% and 5.0%; $Cd226^-Pdc1^+$: 31.5% and 25.3%) (Fig. 5e, Supplementary Fig. S15a and b). Flow cytometric analysis demonstrated that WT tumors contained a higher cumulative burden of exhaustion-associated inhibitory receptors on CD8⁺ T cells, as assessed by the co-expression of PD-1, TIM-3, LAG-3, and TIGIT (Supplementary Fig. S17d). In particular, CD8⁺ T cells expressing three or more exhaustion markers were enriched in WT tumors compared with *Vsir*^{-/-} tumors, consistent with a greater prevalence of terminal exhaustion-like CD8⁺ T cell states. In contrast, *Vsir*^{-/-} tumors exhibited a relative reduction in these highly exhausted populations, accompanied by increased TNF- α and IFN- γ production (Supplementary Fig. S6b).”

In **Results** (Pages 16, Lines 310-314)

Original:

“Consistently, *Vsir* expression was significantly correlated with key exhaustion markers (*Havcr2*, *Tigit*, *Lag3*, *Pdc1*, *Cxcl13*, and *Layn*) in both TCGA-PAAD tumor/normal samples and GTEx normal pancreas samples (Supplementary Fig. S14b), highlighting VISTA’s role in promoting T cell exhaustion within the TME.”

Revised:

→ “Consistently, *Vsir* expression was significantly correlated with key exhaustion markers (*Havcr2*, *Tigit*, *Lag3*, *Pdc1*, *Cxcl13*, and *Layn*) in both TCGA-PAAD tumor/normal samples and GTEx normal pancreas samples (Supplementary Fig. S16b), **suggesting a potential association between VISTA and T cell exhaustion within the TME.**”

4. Insufficient Human Tissue Validation:

Although the authors perform database analyses, they have not provided direct experimental data on VISTA expression or macrophage phenotype in human PDAC tissue. Immunohistochemistry or multiplex immunofluorescence staining on a small patient cohort could help validate the translational significance of the findings.

(Author’s Response/Action)

We appreciate your suggestion to strengthen the translational relevance of our findings by providing direct experimental validation in human PDAC tissue. We agree that database-based analyses alone are not sufficient. To address this concern, we performed multiplex immunofluorescence (mIF) on human pancreatic cancer tissue microarrays (TMAs) to directly evaluate VISTA expression in the myeloid compartment and its immune spatial context.

Specifically, we conducted mIF staining for VISTA, CD68, CD8, and CXCR3 using four commercially available PDAC TMAs and quantified both absolute cell counts and cell density per tissue core, alongside spatial proximity metrics between VISTA⁺ macrophages and CD8⁺ T cells (**New Supplementary Figure S19c**). Across tumor grades, we observed an increase in VISTA⁺ macrophages and CD8⁺ T cells (**New Supplementary Figure S19b**). Across tumor grades, we observed a clear increase in VISTA⁺CD68⁺ macrophages, both in absolute number and in density per core (**New Supplementary Figure S19c**). This direct human-tissue result supports the translational consistency of our murine observation that a VISTA-associated macrophage compartment expands with disease progression. Importantly, this pattern is aligned with prior work reporting that VISTA is prominently expressed on CD68⁺ macrophages in human PDAC and enriched in the pancreatic tumor stroma (Blando *et al.*, PNAS, 2019).

New Supplementary Figure S3

f Percentage and cell density of VISTA⁺CD68⁺ co-expressing macrophages (left and middle), and the proportion of VISTA⁺ cells within the CD68⁺ macrophage population (right), across tumor grades in tissue microarrays.

In parallel, we quantified CD8 T cell infiltration and CXCR3 expression in the same TMAs. While CD8 T cell infiltration showed an overall increasing trend across grades, the fraction of

CXCR3⁺ cells within the CD8⁺ compartment decreased when comparing grade 1 versus grade 2 and grade 1 versus grade 3 tumors (**New Supplementary Figure S19a and b**). Although we interpret this finding cautiously given cohort size and the limitations of TMA-based snapshots, an inverse relationship between tumor grade and CXCR3 expression is biologically plausible in PDAC, where progressive immunosuppression can be accompanied by impaired effector trafficking programs.

New Supplementary Figure S19

a Representative images from grade 1 and grade 3 patient samples (20×; left) and bar graph showing the density of CD8⁺ T cells (cells/mm²; right). **b** Representative images from grade 1 and grade 3 patient samples (20×; left) and bar graph showing the density of CXCR3⁺ cells (cells/mm²; right). CD68, Opal 620; VISTA, Opal 690; CXCR3, Opal 520; and CD8a, Opal 480.

Notably, we observed that the nearest-neighbor distance between VISTA⁺CD68⁺ macrophages and CD8⁺ T cells decreased from grade 2 and grade 3 tumors (**New Supplementary Figures S19c**), indicating increased spatial proximity in higher-grade disease.

Together, these findings provide a cellular context consistent with a macrophage-T cell interface in which VISTA expressing macrophages may influence CD8 T cell positioning and functional state in advanced PDAC.

New Supplementary Figure S19

c Representative images from grade 2 and grade 3 patient samples (20 \times ; left) and bar graph showing the nearest-neighbor distance between CD68⁺VISTA⁺ macrophages and CD8⁺ T cells (right). CD68, Opal 620; VISTA, Opal 690; CXCR3, Opal 520; and CD8a, Opal 480.

We emphasize that the purpose of this human mIF experiment is to provide direct tissue-level validation of (i) VISTA localization to the macrophage compartment in PDAC and (ii) the abundance and spatial context of VISTA⁺CD68⁺ macrophages is associated with tumor grade. Accordingly, we present these data as translational validation of a grade-associated VISTA⁺ macrophage compartment and its immune spatial context. Collectively, these new patient tissue mIF data move our study beyond database inference and directly address your request for experimental validation in human samples.

In **Results** (Page 6-7, Lines 89-94)

Original:

“Interestingly, *VISTA* expression increased with advancing histological grade (Supplementary Fig. S3c), a trend also confirmed at the protein level by immunohistochemistry using a PDAC tissue microarray (Supplementary Fig. S3d and e).”

Revised:

→ “Interestingly, *VISTA* expression increased with advancing histological grade (Supplementary Fig. S3c), a trend that was independently validated at the protein level by immunohistochemistry and immunofluorescence analysis of a PDAC tissue microarray (Supplementary Fig. S3d–f). Together, these clinical observations support the relevance of *VISTA* as a therapeutic target in PDAC.”

In **Results** (Page 16-17, Lines 321-336)

Original:

“Flow cytometric analysis further characterized the altered state of CD8⁺ T cells in *Vista*^{-/-} tumor. *Vista*^{-/-} tumor had significantly increased expression of CXCR3, with 28.3% of CD8⁺ T cells co-expressing CXCR3 and CX3CR1 compared to 8.07% in WT tumor (Fig. 5f). Additionally, *Vista*^{-/-} tumor had a higher proportion of TCF-1⁺PD-1⁺ CD8⁺ T cells and fewer terminally exhausted CXCR3⁺CX3CR1⁻ CD8⁺ (Tex^{term}) cells compared to WT control (Fig. 5g and h). These findings suggest that *VISTA* deficiency promotes a CD8⁺ T cell phenotype with reduced exhaustion and improved effector function within the TME.

Finally, we validated these observations in the HTAN WUSTL...”

Revised:

→ “...within the TME. To extend these findings to human disease, multiplex immunofluorescence analysis of a limited number of PDAC specimens revealed that that, although overall CD8⁺ T cell infiltration showed an increasing trend with advancing tumor grade, the fraction of CXCR3⁺ cells within the CD8⁺ T cell compartment was significantly reduced in grade 2 and grade 3 tumors compared with grade 1 tumors (Supplementary Fig. S19a and b), demonstrating an inverse relationship between tumor grade and CXCR3 expression. Notably, spatial analysis further showed a decreased nearest-neighbor distance between *VISTA*⁺CD68⁺ macrophages and CD8⁺ T cells in grade 2 and grade 3 tumors, indicating increased spatial proximity in higher-grade disease (Supplementary Fig. S19c).”

In **Methods** (Page 29, Lines 623-626)

Revised:

→ “Opal multiplex immunofluorescence staining was performed on pancreatic cancer tissue microarrays (PA2082b, PA2081-L64, PA961-L87, and PA483-L97). Sections were stained with antibodies against CD68, VISTA, CXCR3, and CD8 α , which were visualized using Opal 620, Opal 690, Opal 520, and Opal 480 fluorophores, respectively.

Responses to Reviewer's Comments

We are deeply grateful for your thoughtful comments and invaluable suggestions regarding our manuscript. Your feedback has provided us with an excellent opportunity to improve the quality and clarity of our work. We have carefully revised the manuscript in accordance with the reviewers' recommendations and your editorial guidance. Below, we respectfully addressed each of the reviewers' comments point-by-point and described the corresponding revisions made to the manuscript. All changes are clearly **marked in red** for your convenience. Thank you once again for your time and consideration.

Reviewer #1 (Remarks to the Author):

The authors have significantly improved the manuscript by addressing previous critiques and enhancing both translational and mechanistic depth. The inclusion of human pancreatic ductal adenocarcinoma (PDAC) tissue and single cell RNA sequencing data from the HTAN cohort connects findings from mouse models with human data. This demonstrates a consistent relationship between VSIR expression, the phenotype of tumor-associated macrophages (TAMs), and the functionality of CD8⁺ T cells. The addition of rigorous multivariate analyses, single-cell stratification, and other co-expression profiling provides strong support for the manuscript's central hypothesis: that VISTA orchestrates immunosuppression in PDAC by modulating both the TAM and T-cell compartments.

Finally, expanded the discussion of upstream and downstream VISTA signaling, and strengthened the experimental data by including studies on CSF1R inhibition and combination therapy with gemcitabine. The revised manuscript is comprehensive, data-driven, and well-integrated across both murine and human systems.

(Author's Response)

We greatly appreciate your recognition of the improvements to our manuscript. Your constructive feedback in the previous revision, particularly the suggestion to integrate human PDAC tissue data with single-cell RNA sequencing to bridge mouse and human findings, was invaluable in enhancing the clarity and impact of our study.

Reviewer #2 (Remarks to the Author):

Thank you for your detailed responses to my concerns, which have been sufficiently addressed.

(Author's Response/Action)

We sincerely appreciate your thoughtful concerns and are pleased that our revisions have sufficiently addressed your concerns. We are confident that the revised manuscript has been substantially improved.

Reviewer #3 (Remarks to the Author):

The revised manuscript incorporates substantial improvements in response to peer review, transforming the study from a phenotypic analysis into a rigorous, mechanistically detailed investigation with clear clinical relevance. Using a multi-tiered experimental approach, the authors establish a comprehensive evidence chain and confirm VISTA as a pivotal regulator of immune evasion in pancreatic ductal adenocarcinoma (PDAC). Thank you for your detailed responses to my concerns, which have been sufficiently addressed.

(Author's Response/Action)

Thank you for your thorough evaluation and generous comments on the revised manuscript. We appreciate the acknowledgement of the improved mechanistic rigor and clinical relevance, and we are pleased that our responses have sufficiently addressed all of your concerns.